# Evidence of shared and distinct functional and structural brain signatures in schizophrenia and autism spectrum disorder

Yuhui Du [1,2,5✉], Zening Fu[2,5], Ying Xing[1], Dongdong Lin[2], Godfrey Pearlson[3], Peter Kochunov[4], L. Elliot Hong[4], Shile Qi[2], Mustafa Salman[2], Anees Abrol [2] & Vince D. Calhoun [2]

Schizophrenia (SZ) and autism spectrum disorder (ASD) share considerable clinical features and intertwined historical roots. It is greatly needed to explore their similarities and differences in pathophysiologic mechanisms. We assembled a large sample size of neuroimaging data (about 600 SZ patients, 1000 ASD patients, and 1700 healthy controls) to study the shared and unique brain abnormality of the two illnesses. We analyzed multi-scale brain functional connectivity among functional networks and brain regions, intra-network connectivity, and cerebral gray matter density and volume. Both SZ and ASD showed lower functional integration within default mode and sensorimotor domains, but increased interaction between cognitive control and default mode domains. The shared abnormalties in intra-network connectivity involved default mode, sensorimotor, and cognitive control networks. Reduced gray matter volume and density in the occipital gyrus and cerebellum were observed in both illnesses. Interestingly, ASD had overall weaker changes than SZ in the shared abnormalities. Interaction between visual and cognitive regions showed disorder-unique deficits. In summary, we provide strong neuroimaging evidence of the convergent and divergent changes in SZ and ASD that correlated with clinical features.

[1] School of Computer and Information Technology, Shanxi University, Taiyuan, China. [2] Tri-Institutional Center for Translational Research in Neuroimaging and Data Science (TReNDS), Georgia State University, Georgia Institute of Technology, Emory University, Atlanta, GA, USA. [3] Department of Psychiatry, Yale University, New Haven, CT, USA. [4] Maryland Psychiatric Research Center, Department of Psychiatry, University of Maryland, School of Medicine, Baltimore, MD, USA. [5] These authors contributed equally: Yuhui Du, Zening Fu. ✉email: duyuhui@sxu.edu.cn

Schizophrenia (SZ) and autism spectrum disorder (ASD) have both been characterized as disorders of altered neurodevelopment that lead to lifelong disability. ASD is characterized by the onset during early childhood with a spectrum of abnormalities including deficits in social communication and interaction, and restricted and repetitive patterns of behaviors, interests, or activities. The onset of SZ often occurs in late adolescence life and is associated with the development of psychosis and cognitive and social deficits[1]. There is a considerable overlap in the clinical presentation between SZ and ASD, leading to an initial classification of ASD as "childhood-onset SZ"[2]. The historical parallels and sharing of symptom patterns may be caused in part by the sharing of deficits in the underlying neural circuits. However, few studies have directly compared brain function and structure between the two disorders. In this study, we propose to use a multi-modality data-informed method to comprehensively study the similarity and differences between two disorders, aiming to provide statistically powerful evidence to address the following questions: (1) what are the common abnormalities between SZ and ASD in brain functional networks, functional connectivity (FC), and gray matter volume and density? (2) What are the disorder-unique changes of the two disorders in these neuroimaging measures? (3) Measured by these biologically meaningful measures, to what extent ASD resembles SZ?

The relationship between SZ and ASD is complex[3,4]. Although there has been a separation between the two illnesses since the International Classification of Diseases, Ninth Revision (ICD-9) (1977) and the Diagnostic and Statistical Manual of Mental Disorders, Third Edition (DSM-III) (1980), historically, SZ and ASD were considered to be a developmental continuum of the same disorder and both disorders were assumed to share the underlying etiology. Recently, increasing evidence has supported their overlap in social withdrawal, cognitive deficits, and communication impairment[2,5–7]. Remarkably, Eack et al.[8] found a high degree of shared impairments between the two disorders in both social and non-social cognitive domains, especially the slow processing speed and inability to understand emotion. All the findings suggest the possibility of common underlying neurological mechanisms in SZ and ASD. In addition, both disorders may share copy number variants[9,10] and gene expression patterns[11]. Other evidence included higher incidents of children with ASD in the families with a history of SZ[12] and higher rates of developing psychosis during adolescence in children with ASD[13–15]. However, there are also notable differences between populations afflicted with the two disorders, especially the presence of different atypical behaviors in them[16]. Patients with SZ often experience hallucinations and delusional thoughts that are uncommon in ASD. There is a greater likelihood of restricted and repetitive behaviors[17], stereotyped language, and seizures[18] in ASD than in SZ. Patients with SZ often show progressive loss of contact but children with ASD lack contact from the start. Therefore, the unique mechanism of the two disorders also needs to be further explored.

The long-standing debate in terms of their relationship[3,4,19,20] has led to neuroimaging-based attempts to provide an unbiased quantification of similarity and uniqueness between SZ and ASD. There have been studies that evaluated the two disorders using a meta-analysis and revealed overlapping reductions in limbic-striatothalamic gray matter volume[21–23]. Whole-brain FC analysis using resting-state functional magnetic resonance imaging (fMRI) data has reported common and divergent connectivity impairments largely in regions of the default mode (DM) and salience networks[24]. Mastrovito et al.[25] used a classification strategy on resting-state effective connectivity to explore the two disorders, showing their common impairments in connectivity within the DM and salience networks, and the ASD-unique increases within visual (VI)

processing and DM networks. With a task design, seed-based FC analysis supported that connectivity alterations are significantly different between SZ and ASD[26]. Park et al.[27] found significant cortical thickness changes in both ASD and SZ in brain regions relating to the frontoparietal and limbic networks; however, SZ was found to show decreased cortical thickness, while ASD presented increases in cortical thickness[27]. Haigh et al.[28] used diffusion data to study the two disorders, revealing SZ-specific changes in its greater mean diffusivity than both ASD and healthy controls (HCs)[28]. To sum up, both similarity and uniqueness have been disclosed to some extent. However, studies comparing the two disorders using multimodal neuroimaging measures and large-sample data are still very limited, and no existing study has quantified their overlap and uniqueness yet.

In this study, we comprehensively investigate SZ and ASD using large-sample brain functional and structural magnetic resonance imaging data to explore their similarities and differences in the neural substrates. We evaluate large-scale brain spatial functional networks estimated from data-driven independent component analysis (ICA), functional network connectivity (FNC) obtained from ICA, and FC between brain regions of interest (ROIs) derived from different brain atlases. In parallel, we perform the voxel-wise analyses of gray matter volume and density. Our study's contributions are at least twofold. (1) Our results show strong neuroimaging evidence in terms of the convergent and divergent changing patterns between SZ and ASD (relative to HCs) using multiple functional and structural measures. (2) Our neuroimaging evidence provides insights about the relationship between SZ and ASD, which have intertwined historical roots.

## Results

After data preprocessing and quality control, we estimated brain functional measures from resting-state functional magnetic resonance imaging (fMRI) data of 2980 subjects (1665 HCs, 537 SZs, and 778 ASDs), gray matter volume from structural magnetic resonance imaging (sMRI) data of 3148 subjects (1661 HCs, 517 SZs, and 970 ASDs), and gray matter density from sMRI data of 3374 subjects (1789 HCs, 555 SZs, and 1030 ASDs). The Supplementary Tables S1–S4 summarize the sample size, demographic information, and head motion measures of the selected subjects. As outlined in Fig. 1, we then identified the disorder-common and disorder-unique abnormalities for SZ and ASD by investigating their changes relative to the HC group and their differences from each other using multiple neuroimaging measures that reflect brain function and structure.

**SZ and ASD show shared and distinct changes in brain functional networks.** The large-scale brain functional networks were estimated from fMRI data via our recently proposed NeuroMark pipeline[29], which automates the estimation of subject-specific functional networks by leveraging group information-guided ICA (GIG-ICA)[30,31] with the reliable network templates obtained from fMRI data of independent large-sample HCs as priors. The detailed information of the network templates can be found in Supplementary Table S5. As displayed in Supplementary Fig. S1, the resulting functional networks were assigned into seven functional domains, including five sub-cortical (SC), two auditory (AU), nine sensorimotor (SM), nine VI, 17 cognitive control (CC), seven DM, and four cerebellar (CB) networks. We evaluated the spatial similarity of subject-specific functional networks and show the results in the Supplementary Fig. S2, supporting that each functional network (estimated from one same network template) was comparable across different subjects and feasible for further statistical analyses among groups. In addition, the network correspondence was relatively stable across different datasets and groups.

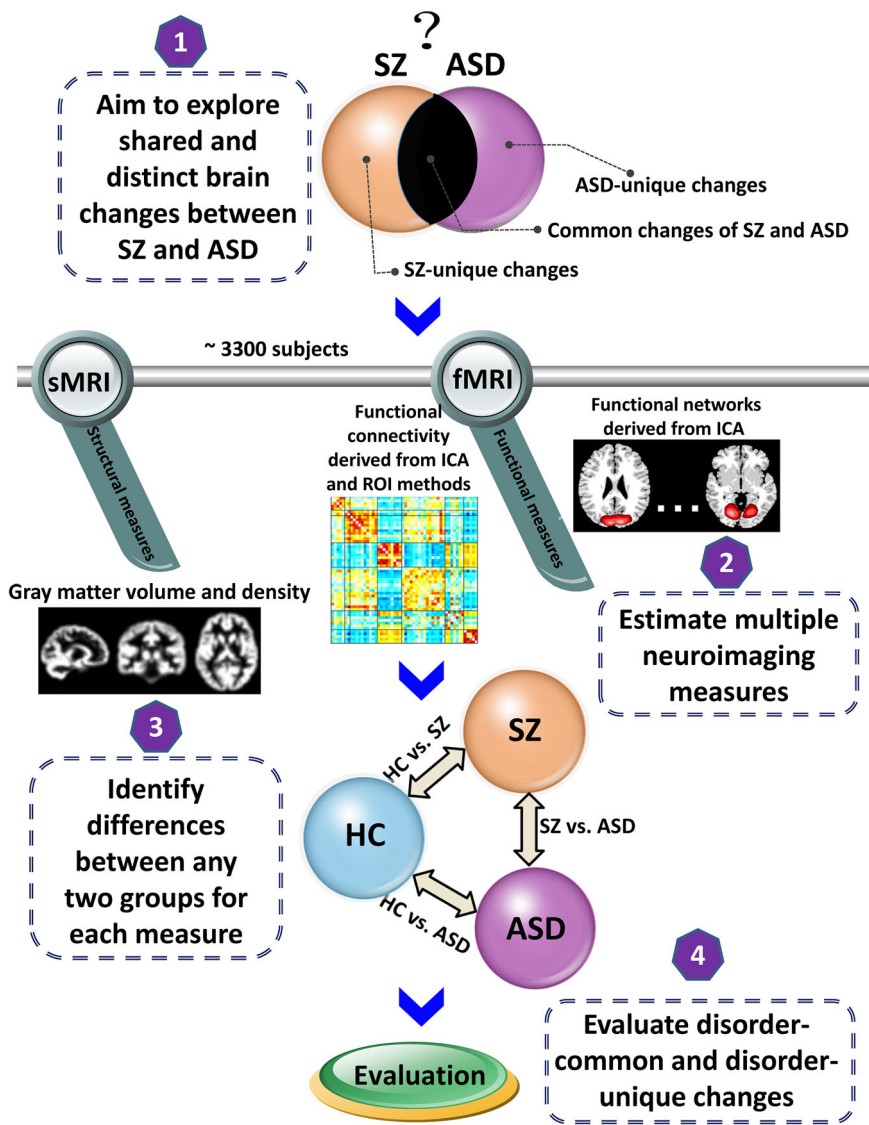

**Fig. 1 Analysis pipeline for identifying the common and unique brain changes of SZ and ASD (relative to HC).** Multiple neuroimaging measures were compared among the HC, SZ, and ASD groups. Brain functional measures included the spatial functional networks and functional network connectivity (FNC) derived from independent component analysis (ICA), and also included the whole-brain functional connectivity estimated by region of interest (ROI) based method using different brain atlases. Brain structural measures involved the gray matter volume and gray matter density. Group differences were investigated between any paired groups (HC vs. SZ, HC vs. ASD, and SZ vs. ASD) to identify the disorder-common and disorder-unique abnormality for SZ and ASD.

By using NeuroMark, each brain functional network is represented by one spatial independent component (IC) consisting of Z-score of each brain voxel, in which the voxels with greater Z-scores have relatively higher intra-connectivity in the network. Many studies have employed the voxel-wise Z-scores as intra-network connectivity measures to study brain abnormality in patients relative to HCs[32,33] and distinguish patients with different disorders[30,34,35]. In the work, consistent statistical analyses were conducted on the neuroimaging measures (including the Z-score in the brain functional network here), to identify the common and unique brain abnormalities of SZ and ASD (relative to HC). The analysis framework is shown in Fig. 2 and more details can be found in the "Methods" section.

**Disorder-common changes of SZ and ASD (relative to HC) in brain functional networks.** Our results (Table 1) suggest that overall SZ and ASD showed more common changes (relative to HC) than unique changes (relative to HC) in brain functional

networks. Among all 53 networks, 35.5% and 39.9% of brain voxels that passed the analysis of variance (ANOVA) showed common decreases and increases in the two disorders (relative to HC), respectively. As shown in Fig. 3a, the common changes were primarily found in the DM, SM, and CC networks, including the common decreases in the posterior cingulate cortex (PCC; IC 94, DM), inferior parietal lobule (IC 68, CC), precentral gyrus (IC 66, SM), and subthalamus/hypothalamus (IC 53, SC), and the common increases in the middle frontal gyrus (IC 38, CC), superior parietal lobule (IC 80, SM), hippocampus (IC 48, CC), and precuneus (IC 51, DM). Several CB regions (e.g., IC 4 and IC 7) also showed a common decrease. Notably, as shown in Fig. 3b and Table 1, for the voxels with common decreases in SZ and ASD (relative to HC), 85.3% voxels showed weaker (smaller) changes in ASD than SZ; for the voxels with common increases in SZ and ASD (relative to HC), 94.4% voxels had weaker (smaller) changes in ASD than SZ, supporting that in general ASD presented weaker changes than SZ for the shared abnormalities.

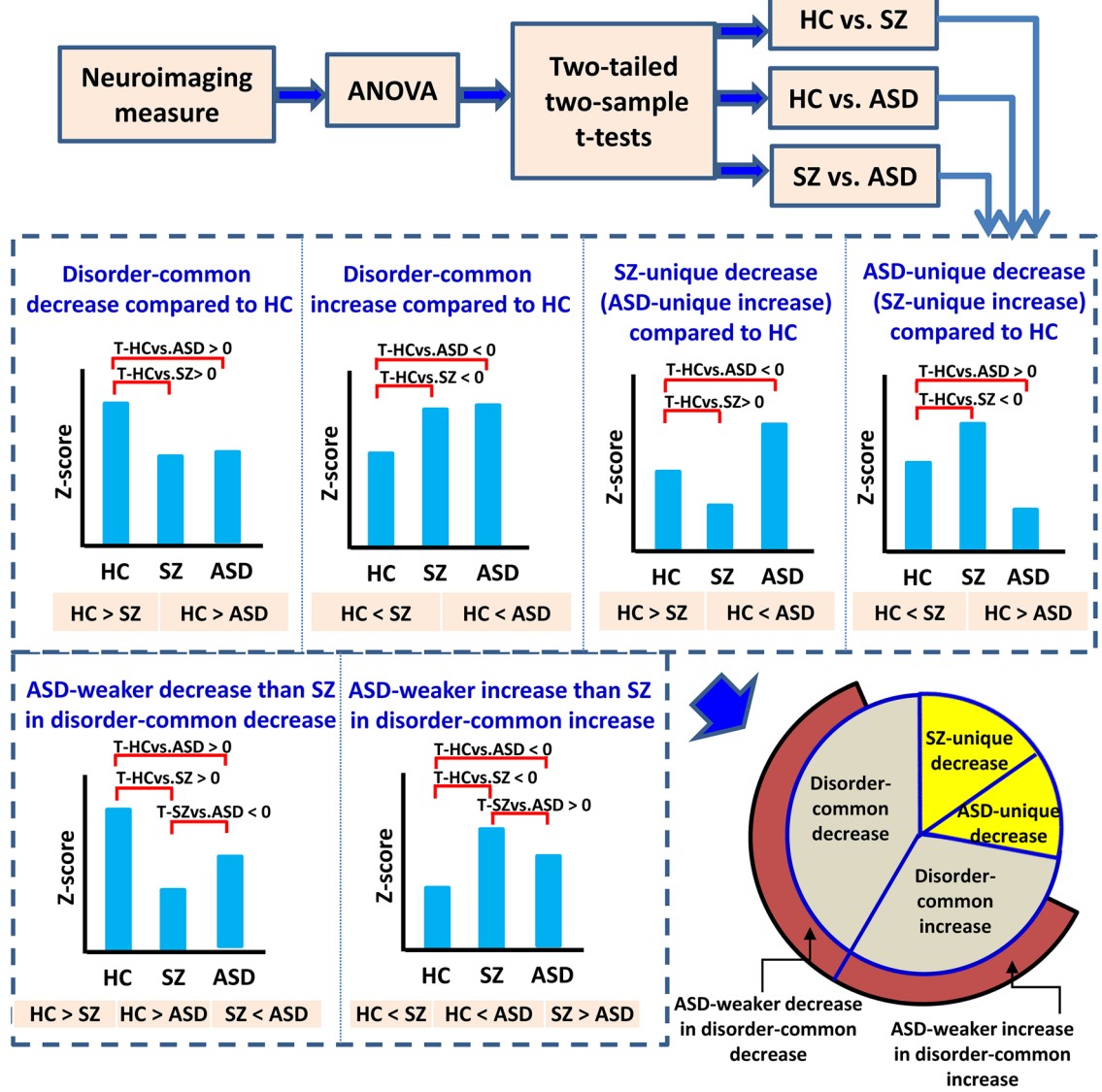

**Fig. 2 Statistical analysis outline for identifying and summarizing the common and unique brain abnormalities of SZ and ASD.** The statistical analysis procedure is consistent across all neuroimaging measures mentioned in Fig. 1. Regarding each measure, analysis of variance (ANOVA) and two-tailed two-sample t-tests were first performed, resulting in group differences for HC vs. SZ, HC vs. ASD, and SZ vs. ASD. For the measures passing ANOVA, different types of changes were then summarized, including the disorder-common decrease compared to HC (the voxels with $T$-values > 0 in both HC vs. SZ and HC vs. ASD), the disorder-common increase compared to HC (the voxels with $T$-values < 0 in both HC vs. SZ and HC vs. ASD), the SZ-unique decrease (the voxels with both $T$-values < 0 in HC vs. ASD, and $T$-values > 0 in HC vs. SZ), and the ASD-unique decrease (the voxels with both $T$-values < 0 in HC vs. SZ and $T$-values > 0 in HC vs. ASD). For each of the four types of changes, the percentage was calculated as the number of neuroimaging measures relating to the change divided by the number of measures passing ANOVA. Among the measures with the disorder-common decrease (or increase), the percentage of ASD-weaker decrease (or increase) than SZ, which showed $T$-values < 0 in SZ vs. ASD in the disorder-common decrease (or showed $T$-values > 0 in SZ vs. ASD in the disorder-common increase), were further summarized.

Interestingly, the inferior frontal gyrus (IC 67), CB region (IC 4), and PCC (IC 94) showed common alterations between SZ and ASD (relative to HC) in most voxels, with the decreases dominating the impairments[36]. Moreover, ASD also showed weaker abnormalities than SZ (relative to HC) in the most commonly changed voxels for them.

**Disorder-unique changes of SZ and ASD (relative to HC) in brain functional networks.** Our results revealed the uniqueness of network abnormalities for SZ and ASD. SZ-unique decreases (in which SZ decreased but ASD increased, relative to HC) involved the middle occipital gyrus (IC 5, VI) and inferior parietal lobule (IC 63, CC). Regions showing ASD-unique decreases (in

which ASD decreased but SZ increased, relative to HC) were primarily located at the superior frontal gyrus (IC 96, CC), hippocampus (IC 83, CC), and middle cingulate cortex (IC 37, SM). The percentage of disorder-unique impairments (totaling < 25%), with slightly more SZ-unique decreases, was much lower than the percentage of disorder-common changes (Table 1).

**SZ and ASD show shared and distinct changes in FNC**
*Disorder-common changes of SZ and ASD (relative to HC) in FNC.* We also evaluated group differences in the FNC using the same strategy as the above functional network analyses (see Fig. 2), aiming to investigate the shared and distinct changes of SZ and ASD in the network interaction. The results of statistical analyses and their

**Table 1 Summary of the disorder-common and disorder-unique changes evaluated using functional networks, functional network connectivity (FNC), functional connectivity (FC), and gray matter volume and density.**

| | Common decrease [percentage, number] | Common increase [percentage, number] | SZ-unique decrease (ASD-unique increase) [percentage, number] | ASD-unique decrease (SZ-unique increase) [percentage, number] | ASD-weaker decrease in common decrease [percentage, number] | ASD-weaker increase in common increase [percentage, number] | All measures passing ANOVA [number] |
|---|---|---|---|---|---|---|---|
| Functional networks from ICA | [35.5%, 2592] | [39.9%, 2906] | [13.9%, 1013] | [10.7%, 782] | [85.3%, 2212] | [94.4%, 2744] | 7293 |
| FNC from ICA | [42.5%, 191] | [49.9%, 224] | [3.8%, 17] | [3.8%, 17] | [90.1%, 172] | [91.2%, 204] | 449 |
| FC from AAL ROIs | [43.4%, 401] | [42.3%, 391] | [5.2%, 48] | [9.1%, 84] | [90.0%, 361] | [96.4%, 377] | 924 |
| FC from Brainconnectom ROIs | [48.4%, 349] | [29.9%, 215] | [14.6%, 105] | [7.1%, 51] | [92.0%, 321] | [93.5%, 201] | 720 |
| Gray matter volume | [40.2%, 45898] | [2.0%, 2205] | [57.8%, 65957] | [0%, 0] | [100%, 45891] | [41%, 903] | 114060 |
| Gray matter density | [89.8%, 185750] | [0.1%, 155] | [10.1%, 20833] | [0%, 0] | [100%, 185724] | [96.8%, 150] | 206738 |

For each type of brain change, the related number of measures and its percentage in all measures passing ANOVA are included. Regarding the common decrease (increase), the number and percentage of measures showing weaker decrease (increase) in ASD than SZ (relative to HC) are also included. For the brain functional networks derived from ICA, the percentage and number of voxels were computed across all 53 networks.

summary can be seen in Figs. 4 and 5e. We found, in general, ASD showed many common FNC changes (relative to HC) with SZ, consistent with the findings from spatial brain functional networks. The percentages measured by FNC were 42.5% for the common decrease and 49.9% for the common increase (see Table 1). As shown in Fig. 5a, b, the commonly decreased connectivity was observed within the SM and DM domains, between SC and CB domains, between SM and VI domains, between SM and AU domains, and between CC and CB domains; the commonly increased connectivity was observed between the DM and CC domains, between SC and AU domains, between SC and SM domains, between SC and VI domains, between SM and CB domains, and between VI and CB domains. Notably, regarding >90% of the commonly changed FNCs that showed similar changing trends in SZ and ASD relative to HC, ASD showed weaker (smaller) changes than SZ in each of the FNCs (see Table 1). The finding was consistent with that from the functional network analyses.

*Disorder-unique changes of SZ and ASD (relative to HC) in FNC.* Only a few FNCs showed disorder-unique changes, as shown in Figs. 4 and 5c, d. The percentages of SZ-unique decrease and ASD-unique decrease were both 3.8% (see Table 1). The SZ-unique decreased connectivity included that between DM and SM (e.g., IC 94-PCC and IC 9-postcentral gyrus), between DM and VI (e.g., IC 51-precuneus and IC 5-middle occipital gyrus), between CC and SC (e.g., IC 33-insula and IC 98-putamen), between CC and SM (e.g., IC 33-insula and IC 2-paracentral lobule), and the connectivity within CC domain (e.g., IC 33-insula and IC 88-middle frontal gyrus). Primary ASD-unique decreased connectivity was found between VI and CC (e.g., IC 12-middle occipital gyrus and IC 88-middle frontal gyrus), between VI and SC (e.g., IC 8-lingual gyrus and IC 99-caudate, IC 15-cuneus and IC 99-caudate), between SM and CC (e.g., IC 9-postcentral gyrus and IC 83-hippocampus), and between SM and CB (e.g., IC 66-precentral gyrus and IC 13-cerebellum).

*Group differences are reliable.* To verify our finding, we also performed a permutation test to examine group differences in FNCs. Our results support that group differences were reliable to different analysis methods (see Supplementary Fig. S3 and Fig. 4 for a comparison). In addition, the overall results were replicated in a meta-analysis using separate datasets, as shown in the Supplementary Fig. S4.

**FC findings are consistent for using ROI-based approaches.** FC impairments in the two disorders were validated via the ROI analysis, defined by Automated Anatomical Labeling (AAL)[37] and Brainnetome[38] atlases. Using AAL atlas, SZ and ASD had over 40% overlap in both decreased and increased FC alterations, and SZ was more severely affected than ASD for the most commonly impaired FCs (>90%) (see Table 1). The percentage relating to the disorder-unique change was relatively low (14.3%). Comparing Figs. 6 and 4, we found that in general the group differences obtained from different FC analysis methods showed similarity, especially the interaction between the SC domain and other domains (e.g., cerebellum, SM, and VI regions), between the cerebellum and other domains (e.g., SM, VI, CC, and DM regions), and the interactions within the VI domain and within the DM domain. Using Brainnetom atlas, our results (Supplementary Fig. S5) provided further evidence that the hypothesis-based FC analyses supported the findings from the data-driven ICA.

**SZ and ASD show shared and unique changes in gray matter volume and density**
*Disorder-common changes of SZ and ASD (relative to HC) in gray matter volume and density.* In addition to the brain functional

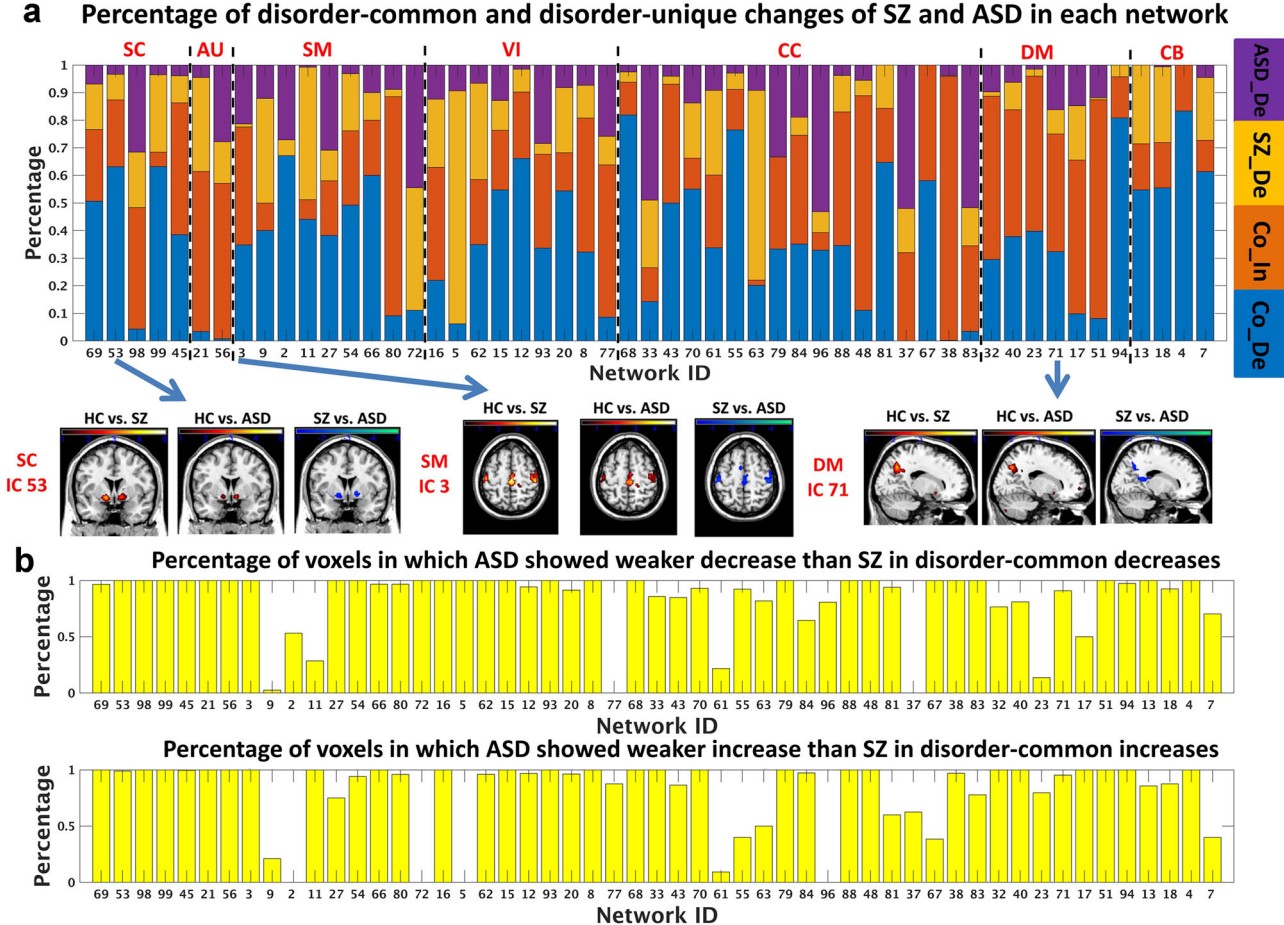

**Fig. 3 Results of the brain functional network analysis using ICA. a** Percentages of the common and unique changes of SZ and ASD in each functional network. The common changes corresponded to the commonly decreased (Co_De) and commonly increased (Co_In) Z-scores of network voxels in SZ and ASD, relative to HC. The SZ-unique decrease (SZ_De) represented decreased Z-score in SZ but increased Z-score in ASD, compared to HC. The ASD-unique decrease (ASD_De) represented decreased Z-score in ASD but increased Z-score in SZ, compared to HC. In **a**, three exemplar networks are shown for visualization of common decreases. **b** Percentages of voxels in which ASD showed weaker changes than SZ within the disorder-common changes.

measures, we assessed brain structural changes of the two disorders. We found that the percentage of the common change between SZ and ASD (relative to HC) was smaller in the brain structural measures compared to the brain FC measures. Measured by the original *T*-values of two-sample *t*-tests, both SZ and ASD showed overall decreased gray matter volume (40.2% overlap) and density (89.8% overlap) compared to HC (Table 1). As displayed in Fig. 7, which shows the group differences after multiple comparison correction, the regions with the commonly decreased gray matter volume primarily involved the middle occipital gyrus and cerebellum (see the Supplementary Table S6), and the regions with the commonly decreased gray matter density primarily involved the precentral gyrus, frontal gyrus, occipital gyrus, postcentral gyrus, temporal gyrus, cerebellum, insula, thalamus, parahippocampal gyrus, superior parietal lobule, supramarginal gyrus, and angular gyrus (see the Supplementary Table S7). In contrast to the shared decrease, there was a low percentage in the disorder-common increases (2% overlap in gray matter volume) primarily relating to the cerebellum. Notably, ASD also had weaker impairments than SZ in those common changes.

*Disorder-unique changes of SZ and ASD (relative to HC) in gray matter volume and density.* Regarding the disorder-unique structural changes, 57.8% and 10.1% of voxels had increased values in ASD but reduced values in SZ for the gray matter volume and density measures, respectively. As summarized in Supplementary

Tables S6 and S7, the regions showing the SZ-unique decrease primarily included the temporal gyrus, frontal gyrus, and cingulate gyrus, identified by the gray matter volume. For the gray matter density, there were only some scattered voxels showing the SZ-unique decrease. Strikingly, no brain regions showed increased gray matter in SZ but decreased gray matter in ASD for both measures.

**Brain changes are associated with symptom scores and are not correlated with medication.** To evaluate the relationship between neuroimaging measure and symptom score (i.e., the positive and negative syndrome scale (PANSS)-positive score and PANSS-negative score for SZ, autism diagnostic observation schedule (ADOS) total score, and social responsiveness scale (SRS) for ASD), we computed both Pearson's correlation and Spearman's rank correlation between them for the SZ and ASD groups, separately. We found that some important neuroimaging measures were linked to the symptom scores.

As shown in the Supplementary Fig. S6, two FNCs showing the disorder-common decreases were negatively correlated with the symptom scores (p-value < 0.01 in both Pearson's and Spearman's correlations) such as ADOS and SRS in ASD. Interestingly, they were all from the within-domain connectivity (SM and CC). In sum, our results suggest that decreased connectivity strengths between brain regions within the SM and CC domains may relate to worse clinical presentations in disorders.

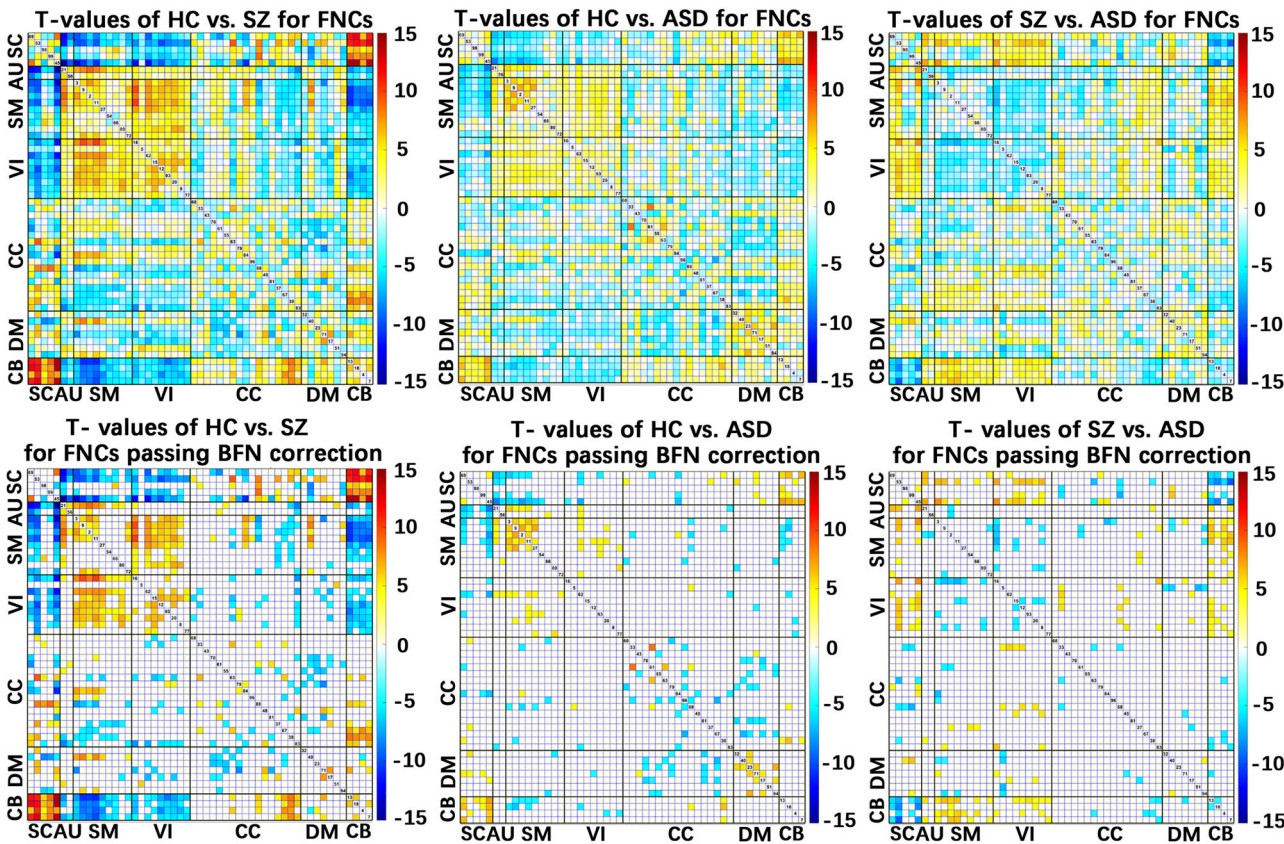

**Fig. 4 Results of the functional network connectivity (FNC) analysis using ICA.** Upper subfigures: *T*-value maps showing the group differences in FNCs, obtained by two-sample *t*-tests for HC vs. SZ, HC vs. ASD, and SZ vs. ASD. Taking HC vs. SZ, e.g., positive *T*-values represented that HCs showed higher connectivity strengths than SZs. Lower subfigures: *T*-value maps of FNCs after Bonferroni (BFN) corrections.

Five disorder-common increased FNCs, including those between SM and CB, SC and VI, SC and DM, and CC and DM, were positively correlated with the symptom scores in ASD and SZ. That means that increased strengths in those connections (such as the connectivity between CC and DM) could result in worse clinical presentations for both SZ and ASD.

Two disorder-unique FNCs showing decreased strengths in ASD but increased strengths in SZ were also found to be correlated with the SRS in ASD and PANSS-positive scores in SZ. Notably, each was a connectivity between the VI and CC domains, again indicating the unique property of VI impairment. Moreover, the correlation trends were consistent with the group difference results.

In addition, we did not find significant associations between the neuroimaging measures and medication as assessed via chlorpromazine (CPZ) dosage equivalents for SZ patients, using a multiple linear regression model.

**Brain changes show consistency using the subjects with matched age and the subjects with no motion difference.** As more data help to generate reliable findings, we used all available data in the above-mentioned statistical analyses. In our work, we also investigated group differences in FNC using two additional sample sets. The two sets had different age ranges of subjects and each set only included age-matched three groups. In addition, we selected some subjects with no motion difference in fMRI data to test the group differences. Our results (shown in the Supplementary Fig. S7) suggest that the group differences using age-matched subjects or no-motion-difference subjects tended to show similar patterns with the results using all available subjects (shown in Fig. 4), supporting that

the nuisance effects (such as age and motion) had been carefully removed out and our findings were robust.

**Brain changes can successfully distinguish SZ and ASD.** We were also interested in whether the identified brain changes can represent promising biomarkers to distinguish the two disorders. As described in the "Methods" section, we used the disorder-unique measures and the ASD-weaker measures within the disorder-common changes as the features for classification. Our results support that FNC measures performed well in classifying SZ and ASD patients. Table 2 includes the classification results from 12 classification experiments that took different datasets as the training and testing data for a comprehensive evaluation. The mean accuracy, sensitivity, and specificity across all classifications were 75%, 83%, and 63%, respectively. The best results reached up to 80.0% accuracy, 90.0% sensitivity, and 68.0% specificity. The most frequently used features across classifications included the unique changes of connectivity between SC (e.g., putamen, caudate, and thalamus) and SM (e.g., superior parietal lobule and paracentral lobule) domains, and also included the ASD-weaker changes in the common changes of connectivity between SC and VI/CB domains, and between SM and VI domains. In our work, relatively lower specificity (compared to sensitivity) meant that SZ patients were more likely misdiagnosed as ASD using these connectivity measures.

## Discussions

SZ and ASD are recognized as distinct illnesses following a long-standing nosological development[4,20]. Their similarity in clinical symptoms characterized by social and communication deficits

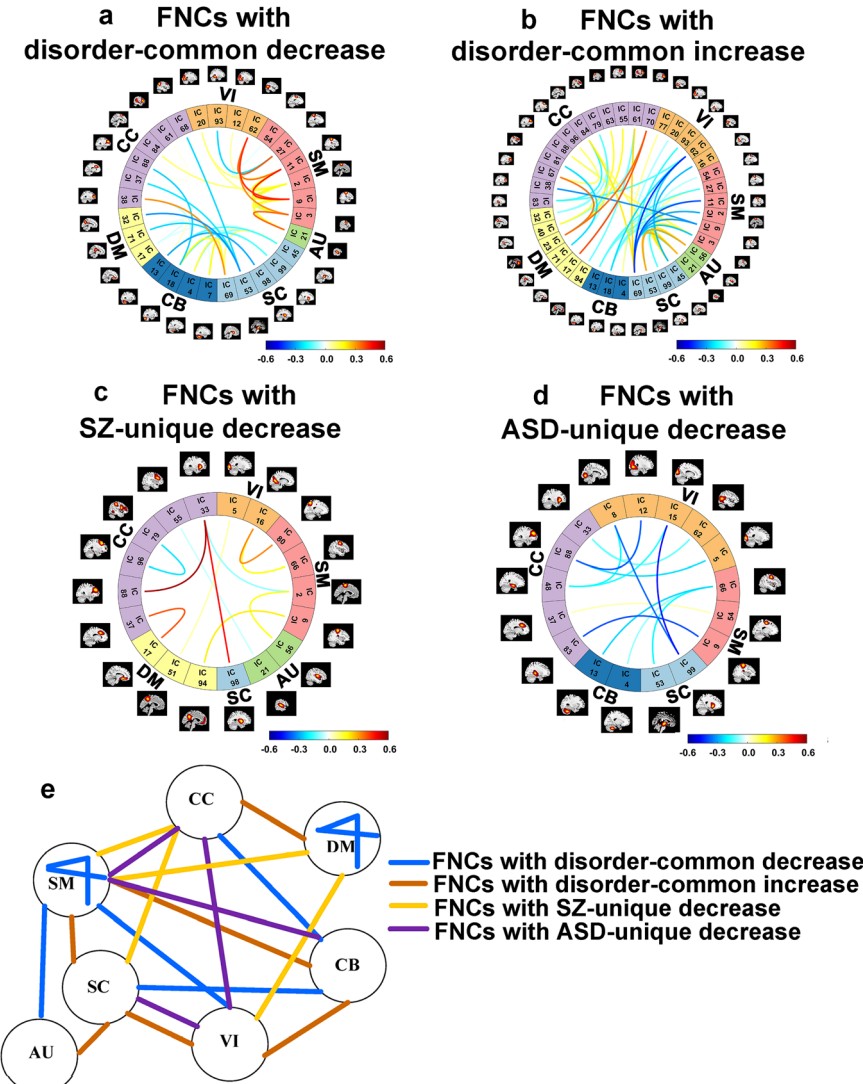

**Fig. 5 Visualization of the FNC results. a–d** Visualization of the FNCs with the disorder-common decrease, the FNCs with the disorder-common increase, the FNCs with SZ-unique decrease, and the FNCs with ASD-unique decrease after Bonferroni (BFN) corrections. For each of these FNCs, the mean FNC strength in the HC group is shown here. **e** Summary of the disorder-common and -unique FNC changes.

and sensory abnormalities[8] may originate from the sharing of functional and structural alterations[22,39]. Therefore, it is greatly needed to unravel how SZ and autism are related and unique in brain abnormality. This is so far the largest study that investigates the commonality and specificity between SZ and ASD in both brain FC and gray matter impairments by investigating them simultaneously and directly, showing neuroimaging evidence to help elucidate their neural substrates (see Fig. 8 for a summary).

**Shared functional abnormalities between SZ and ASD.** Our work provides evidence that SZ and ASD have a large common overlap (76–93%) in the functional changing patterns relative to HCs, primarily involving the DM, CC, and SM circuits. Although some studies have suggested that the two disorders are functionally related[24,40], our work provides quantitative evaluation supporting they are largely overlapped in brain functional abnormality.

In particular, we found that the decreased interaction between regions in the DM domain was prominent in both SZ and ASD, although previous studies have shown the DM dysfunction for them separately[41,42]. We also observed reduced intra-connectivity in the PCC and increased intra-connectivity in precuneus for both illnesses, aligning with previous findings[36]. Interestingly, we revealed that the

integration within the SM domain was also diminished in both disorders, supporting their SM abnormality[43–46].

In addition to the within-domain dysfunction, our results supported that the interactions between CC and DM regions were commonly increased in SZ and ASD. As one previous study[47] observed that schizophrenic patients' FC between DM and executive control was increased and related to hallucinations severity, this work confirmed the similarity of the two disorders regarding this aspect. Our study also found that other inter-domain interactions displayed shared alterations in both disorders, including the decreased connectivity between the CB and SC/CC domains and between the SM and VI/AU domains, and the increased connectivity between the SC and AU/SM/ VI domains and between the CB and SM/VI domains. Although previous studies[5,48] already reported cortico-SC FC abnormality in ASD relative to HC, our work highlights that cerebellum receiving information from the sensory systems and also participating cognition[49,50], SC regions involved in memory and emotion, and vision-related regions were similarly impaired in both disorders.

**Shared structural abnormalities between SZ and ASD.** SZ and ASD showed similar structural alterations in the brain that can be mainly summarized as regional reductions in gray matter volume

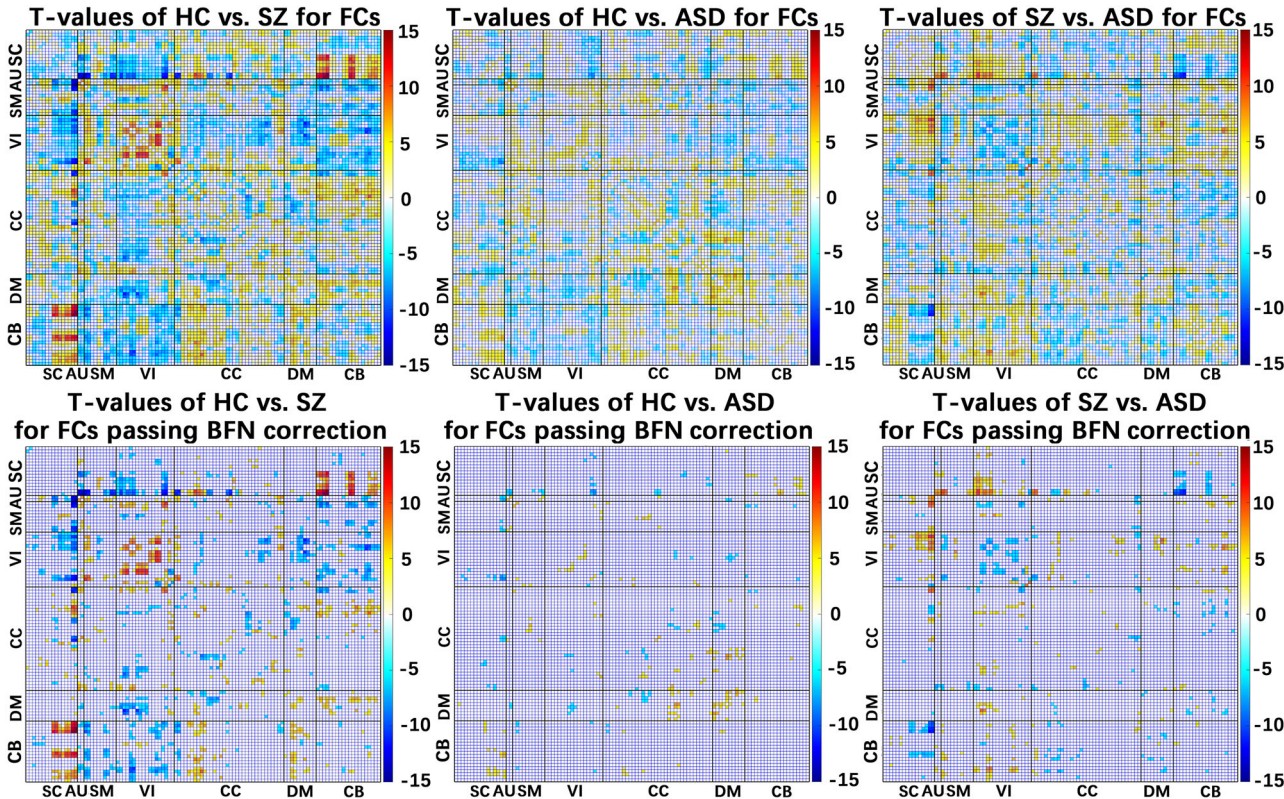

**Fig. 6 Results of functional connectivity (FC) analysis using ROIs of Automated Anatomical Labeling (AAL) atlas.** Upper figures: the original *T*-value maps representing group differences in FCs revealed by two-sample *t*-tests for HC vs. SZ, HC vs. ASD, and SZ vs. ASD. Lower figures: the *T*-value maps of FCs after Bonferroni (BFN) corrections.

and density. Prominent regions with reduced gray matter volume included the middle occipital gyrus and cerebellum, and the regions with decreased gray matter density involved more regions such as the precentral gyrus, frontal gyrus, occipital gyrus, postcentral gyrus, temporal gyrus, cerebellum, insula, thalamus, parahippocampal gyrus, superior parietal lobule, supramarginal gyrus, and angular gyrus. Many studies support our findings. A previous study revealed decreased gray matter density in the frontal gyrus, occipital gyrus, and temporal gyrus in SZ compared to HC[51]. Jiang et al.[52] found that compared with HC subjects, SZ patients showed reduced gray matter volume in the frontal lobe, precentral gyrus, postcentral gyrus, temporal gyrus, occipital cortex, and cerebellum, and did not found regions with increased gray matter volume. Regarding ASD, Foster et al.[53] found that gray matter of ASD decreases near the temporoparietal junction compared to typically developing children and Liu et al.[54] found that pediatric ASD individuals showed significant gray matter decreases in the left cerebellum and left postcentral gyrus, compared to HC subjects. As for the cerebellum, it is associated with both motor and cognition function impairments in SZ and ASD, which has been confirmed by previous studies[50,55]. In addition, our finding supports that the cerebellum showed both disorder-common decrease and disorder-common increase in gray matter volume, which may be an instruction for the subdivision of the cerebellum. Taken together, our study provided important evidence about the shared brain structural impairments between SZ and ASD.

**ASD-weaker changes compared to SZ in the disorder-common changes**. Another interesting finding in this study is that patients with ASD showed a smaller magnitude of impairment than patients with SZ for most of the shared neuroimaging

changes. Within the common changes, 85.3~96.4% across FC/network and 41~100% across gray matter measures were altered less in ASD than SZ. Taking together, our findings provide multi-modal evidence that ASD and SZ had a considerable overlap in deficits and the ASD-weaker impairments were evident, which might account for their complex historical relationship that ASD was ever considered as the early phase of SZ.

**Disorder-unique changes of SZ and ASD**. We identified unique changing patterns of the two disorders to provide informative clues on the disorder-specific etiology and pathophysiology. Functionally, our results supported that the interaction between DM and VI functions, and the interaction between DM and SM functions showed decreases in SZ but increases in ASD, supporting that different manifestations in the communication between self-related processing and VI/SM functions underlie the differences between SZ and ASD. It has been known that both disorders experience VI processing abnormalities; however, they have different phenotypes. For example, VI hallucinations occur relatively frequently in SZ[56,57] and ASD often presents an atypical pattern in eye contact[58]. A previous work by Park et al.[59] used cortical anatomy measures to investigate SZ, ASD, and attention deficit hyperactivity disorder (ADHD), and found that different subcomponents of the extended VI network are affected in SZ patients compared with those with ASD and ADHD, supporting our finding in terms of their different VI function deficits. Although these studies suggested their specificity in vision, our work further provides evidence that the way how the visual processing interacts with social understanding may be different. Regarding the SM presentation, the two disorders also have different diagnostic features, as ASD patients often show repetitive

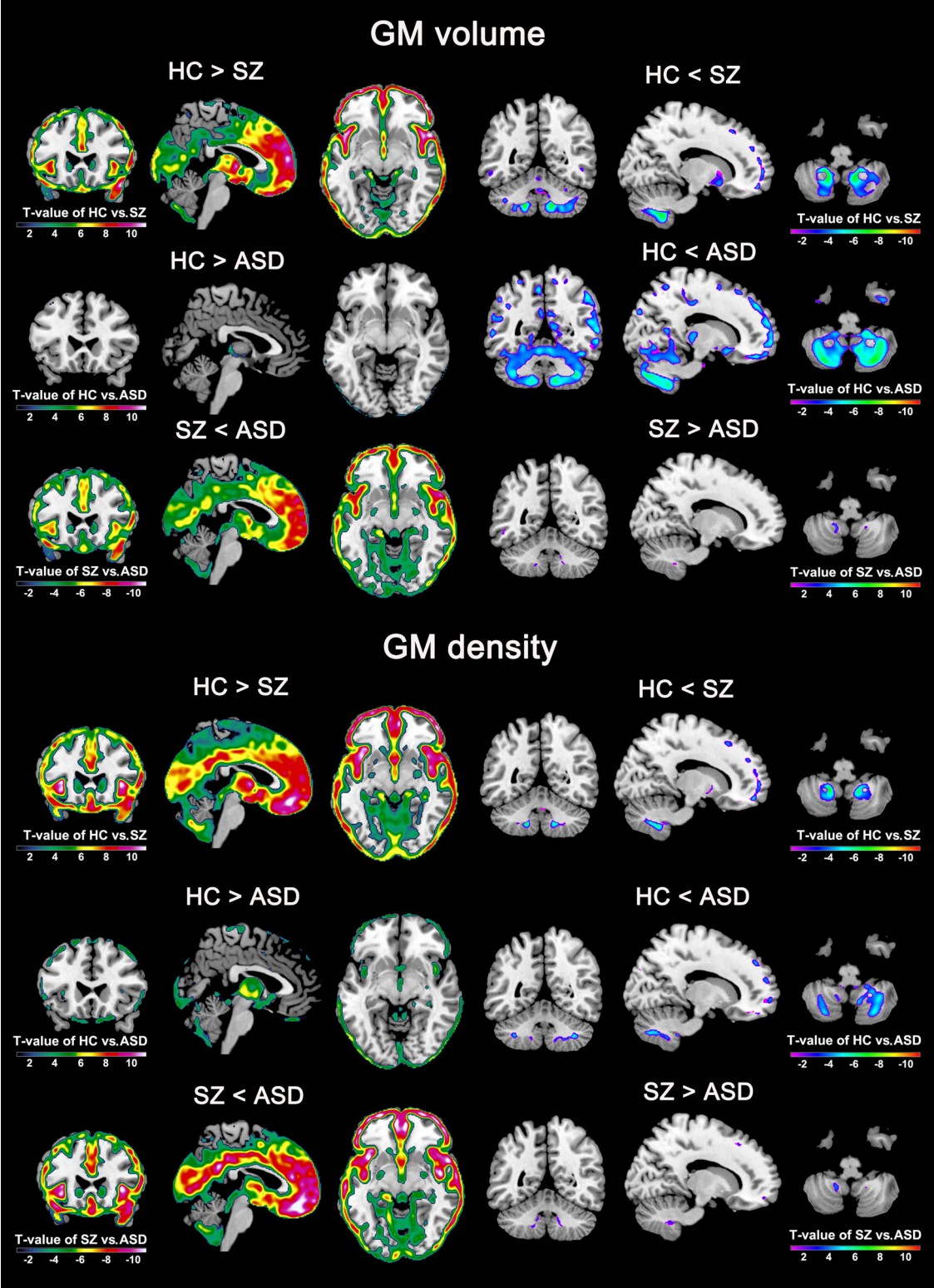

**Fig. 7 Group differences in the gray matter volume and density measures after multiple comparison correction.** For gray matter volume and density, the T-value maps from two-sample t-tests (p < 0.01, false discovery rate corrected) for HC vs. SZ, HC vs. ASD, and SZ vs. ASD are shown.

motor movements[43] that are less observed in SZ. The differences in SM function have also been disclosed by data analysis methods. For example, by comparing the fMRI response to somatosensory stimuli, a study from Haigh et al.[60] confirmed that differential sensory fMRI signatures were present between SZ and ASD, with SZ having smaller responses amplitude and ASD having less trial-to-trial reliability. Except for the discrepancy in the SM system, our work observed their unique impairments in terms of the interaction between DM and SM regions, highlighting the importance of motor systems and their interaction with high-level association systems in mental disorders.

**Table 2 Classification result evaluation of distinguishing SZ and ASD patients using different datasets as the training and testing data.**

| Training data | | Testing data | | Accuracy | Sensitivity | Specificity |
|---|---|---|---|---|---|---|
| SZ datasets (N) | ASD dataset (N) | SZ datasets (N) | ASD dataset (N) | | | |
| BSNIP & FBIRN (319) | ABIDEI (398) | COBRE & MPRC (218) | ABIDEII (380) | 75.8% | 84.2% | 61.0% |
| BSNIP & COBRE (250) | ABIDEI (398) | FBIRN & MPRC (287) | ABIDEII (380) | 73.9% | 85.3% | 58.9% |
| BSNIP & MPRC (332) | ABIDEI (398) | FBIRN & COBRE (205) | ABIDEII (380) | 78.1% | 80.5% | 73.7% |
| FBIRN & COBRE (205) | ABIDEI (398) | BSNIP & MPRC (332) | ABIDEII (380) | 67.4% | 88.7% | 43.1% |
| FBIRN & MPRC (287) | ABIDEI (398) | BSNIP & COBRE (250) | ABIDEII (380) | 78.9% | 84.7% | 70.0% |
| COBRE & MPRC (218) | ABIDEI (398) | BSNIP & FBIRN (319) | ABIDEII (380) | 80.0% | 90.0% | 68.0% |
| BSNIP & FBIRN (319) | ABIDEII (380) | COBRE & MPRC (218) | ABIDEI (398) | 77.9% | 80.2% | 73.9% |
| BSNIP & COBRE (250) | ABIDEII (380) | FBIRN & MPRC (287) | ABIDEI (398) | 77.2% | 85.4% | 65.9% |
| BSNIP & MPRC (332) | ABIDEII (380) | FBIRN & COBRE (205) | ABIDEI (398) | 73.1% | 76.4% | 66.8% |
| FBIRN & COBRE (205) | ABIDEII (380) | BSNIP & MPRC (332) | ABIDEI (398) | 70.4% | 88.2% | 49.1% |
| FBIRN & MPRC (287) | ABIDEII (380) | BSNIP & COBRE (250) | ABIDEI (398) | 73.8% | 78.1% | 66.8% |
| COBRE & MPRC (218) | ABIDEII (380) | BSNIP & FBIRN (319) | ABIDEI (398) | 71.7% | 77.9% | 63.9% |

N represents the subject number.

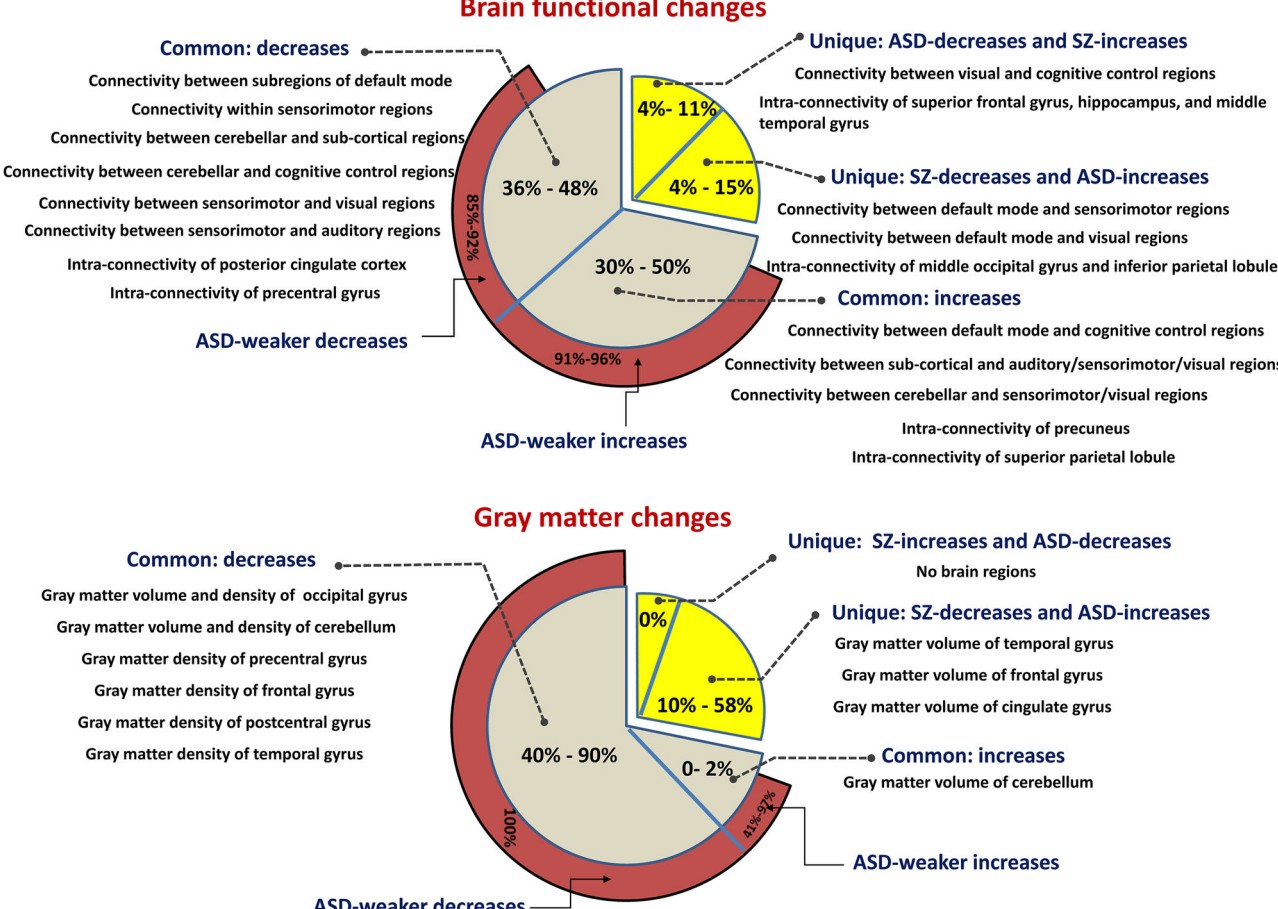

**Fig. 8 Summary of the common and unique brain changes between SZ and ASD (relative to HC).** For both the brain functional changes and the gray matter changes, we summarize the disorder-common decreases, the disorder-common increases, the SZ-unique decreases (i.e., SZ-unique decreases), and the ASD-unique decreases (i.e., SZ-unique increases). Regarding each kind of change, we include the related brain connectivity or brain regions, as well as its percentage. We also include the percentage of ASD-weaker change in the disorder-common abnormalities.

The disorder-specific abnormality identified in our study also involved the communication between the VI and CC functions showing decreases in ASD but increases in SZ. Indeed, the important relationship between VI processing and social perception and cognition in both SZ[61] and ASD[62] has been reported. From a data analysis angle, Eack et al.[63] investigated differences between SZ and ASD using fMRI data under a design of a VI perspective-taking task and revealed their unique frontotemporal connectivity changes compared to the healthy group. Although these studies have indicated the disorder differences in VI and cognitive functions, our results support that the interaction between VI and cognitive regions had divergent alterations in the

two disorders. Measured by spatial functional networks, SZ-unique decreases related to the VI and sensory information processing in the middle occipital gyrus and inferior parietal lobule, and the ASD-unique decreases involved higher cognitive functions such as the superior frontal gyrus. Overall, our findings suggest that unique neural mechanisms may be more related to the VI function intertwined with cognition and SM.

Regarding the brain structural abnormality, we found that the SZ-unique decreases (i.e., ASD-unique increases) related to the temporal gyrus, frontal gyrus, and cingulate gyrus, whereas no voxels showed increased gray matter in SZ and decreased gray matter in ASD (relative to HC). These unique structural abnormalities were among sensory receptive and cognitive areas. Remarkably, one work by Katz et al.[64] supports our findings, as they also noted significant gray matter volume increases in ASD compared with SZ for several cognition-related regions but did not find gray matter increases in the SZ group relative to the ASD group. Given relevant neuroimage-based studies directly comparing SZ and ASD are very rare, our work using multi-modal data demonstrates the value of neuroimaging in identifying subtle differences between the two disorders.

**Association with clinical symptoms**. Some disorder-common and disorder-unique neuroimaging measures were associated with the clinical symptom scores such as the ADOS and SRS in ASD and PANSS scores in SZ. Interestingly, the commonly decreased connectivity correlated with the symptom scores was all from the within-domains including the SM and CC domains, suggesting that lower integration ability in these functions may relate to worse clinical presentation in the two disorders. The commonly increased connectivity correlated with clinical symptoms primarily linked different domains (e.g., the connectivity between DM and CC). Our finding indicated that stronger interaction between the DM and CC functions could account for greater social and cognitive impairments.

The disorder-unique connectivity measures correlated with the symptom scores were all between VI and CC domains, again highlighting that the unique changes in the functional interaction between VI and CC functions are prominent. ASD patients with higher symptom scores showed lower functional interaction between VI and CC domains; conversely, SZ patients with higher symptom scores presented higher functional interaction between them.

**Validation of the finding**. In this study, we validated our findings using all available subjects, age-matched subjects, and no-motion-difference subjects. In addition, we further verified our results by performing a meta-analysis based on separate datasets. Our findings are reliable, as the group differences obtained using different manners showed consistent patterns. We also found that the FNCs with disorder-unique changes and the FNCs with ASD-weaker changes in the disorder-common abnormalities performed well in distinguishing the two disorders. Satisfactory performances were reliably achieved even under the situation of different datasets as the training data/testing data with varied sample sizes. The connectivity between the SC domain and other domains such as SM, VI, and CB areas, as well as the connectivity between the SM and VI domains played an important role in distinguishing the two disorders, which indicates potential biomarkers between the two disorders. To the best of our knowledge, this is the first study applying large-sample multi-site fMRI data to perform the direct classification between SZ and ASD. Our work achieved relatively higher classification accuracy than a previous study[25], which also directly classified the two disorders using FC features. They trained a classifier on 72 SZs and 37 ASDs, and resulted in 75% classification accuracy on independent

5 SZ and 27 ASD patients. A recent study from Yoshihara et al.[40] employed FC features to explore the complex relationship between SZ and ASD as well. In their work, dual classifiers were applied to discriminate ASD (or SZ) from HC to investigate the two disorders using a dimensional method, demonstrating the overlapping but asymmetrical relationship between ASD and SZ. Interestingly, they found that SZ subjects showed increased classification certainty for the ASD dimension, while the ASD subjects did not for the SZ dimension. The findings are consistent with our classification results to some extent, as SZ was more likely to be misclassified into ASD than the opposing situation in our classification experiments.

**Limitations and future directions**. Our study has some limitations that should be considered in future work. In this work, we used the age-matched subjects to validate the group differences in FNC and also employed a classification strategy to show the effectiveness of the identified differences in differentiating the two disorders. However, we did not explore other measures using these procedures due to limited space. Further validations on other functional and structural measures can be conducted in future work. Another limitation is the effects of "noisy" factors. Minimizing the influences of age, gender, site, and motion is often a difficulty. As there is no ground truth of the group differences, it is hard to judge whether these "noises" were fully regressed out. In our study, comprehensive processing was implemented to handle those covariates. Our results also validated the reliable differences using age-matched subjects. However, more advanced algorithms are still needed to deal with those effects. The third shortcoming lies in the comparison among different neuroimaging measures. In our work, both data-driven and hypothesis-derived FC measures were investigated. However, it was not easy to compare the FC results due to their various parcellations and preprocessing procedures. In addition, we only summarized the relation between functional and structural changes according to the affected brain regions. Fusion or more advanced methods[27] may be better for a cross-modal link in a direct manner. Another point that needs more concern is that the biomarker-symptom associations in this work were not very significant (correlations were about 0.2). In the future, validation using independent data may be needed to further validate the associations. For another possible future direction, we think the identified brain similarity and uniqueness in our study could help develop new biotypes within the two disorders, as an interesting work[65] has shown greater differences between biotypes than original DSM-driven categories among SZ, ASD, and bipolar disorders.

**Summary**. Our work provides strong evidence that SZ and ASD are substantially similar in both functional and structural brain abnormalities. The primary overlap is linked to DM, CC, and SM circuits, but also expands to other domains. The disorder-unique changes were found in regions such as VI and CC domains. In addition, these affected neuroimaging measures are correlated with clinical symptoms, with a consistent trend with the identified group differences. Our finding can help elucidate the links between the two disorders, which has been a long-standing unsolved issue.

## Methods
**Data and preprocessing**. Resting-state fMRI and sMRI data were from six multi-site datasets including the Bipolar-Schizophrenia Network for Intermediate Phenotypes phase I (BSNIP-1), Function Biomedical Informatics Research Network (FBIRN), Centers of Biomedical Research Excellence (COBRE), Maryland Psychiatric Research Center (MPRC), Autism Brain Imaging Data Exchange phase I (ABIDEI), and ABIDEII. Among those datasets, BSNIP, FBIRN, COBRE, and MPRC included HC and SZ subjects, and ABIDEI and ABIDEII included HC and

ASD subjects. We preprocessed the fMRI and sMRI data using the statistical parametric mapping toolbox (SPM12).

For fMRI data, we removed the first six time points and then performed the rigid body motion correction to correct the subject head motion, followed by the slice-timing correction to account for timing difference in slice acquisition. The fMRI data were subsequently warped into the standard Montreal Neurological Institute (MNI) space using an echo-planar imaging template and were then resampled to $3 \times 3 \times 3$ mm$^3$ isotropic voxels. The resampled fMRI images were further smoothed using a Gaussian kernel with a full width at half maximum (FWHM) = 6 mm. The smoothed fMRI data were used for ICA[66,67]. For the ROI-based method, the smoothed fMRI data were further detrended and band-pass filtered with [0.01–0.15 Hz], followed by regressing out nuisance covariates including six head motion parameters, white matter signal, cerebrospinal fluid signal, and global mean signal[68].

For sMRI data, the T1-weighted images were first segmented into gray matter, white matter, and cerebrospinal fluid by using the standard unified segmentation model[69]. The Diffeomorphic Anatomical Registration Through Exponentiated Lie Algebra (DARTEL) algorithm was employed to create a group template for spatial normalization of the segmented images of each subject. Then, the flow fields generated by DARTEL were used to estimate individual-subject images. After that, individual-subject gray matter images were spatially normalized to the MNI space, modulated or unmodulated, resliced (1.0 mm isotropic voxels), and smoothed (6 mm FWHM Gaussian kernel). Finally, the obtained gray matter volume (modulated data) and density (unmodulated data) were used for voxel-based morphometry analysis.

In the study, we set out criteria to select fMRI and sMRI data separately, resulting in high-quality data and brain masks for further analysis. Regarding the fMRI data, we selected subjects with the following properties: (1) data with head motions <3° rotations and 3 mm transitions along the whole scanning period; (2) data with >120 time points in fMRI acquisition; and (3) data providing a successful normalization in the full brain. For the third point, as good normalization of fMRI data to standard brain template is necessary to group ICA, we proposed a method to evaluate the normalization quality of fMRI data by comparing the individual brain mask with the group brain mask. Our method was inspired by the Group ICA of fMRI Toolbox (https://trendscenter.org/software/gift/) in which a group brain mask is usually obtained based on the individual brain mask and fMRI data within the group brain mask are used for ICA. As such, the similarity between the individual brain mask and group brain mask can be used to reflect the quality of individual fMRI data and then select the subjects. Our method was applied to each dataset's fMRI data, separately. First, using the three-dimensional image in the first time point of fMRI data, the individual mask was calculated for each subject by setting the brain voxels showing greater values than 90% of the whole-brain mean to 1. Then, we generated a group mask by setting voxels included in >90% of the individual masks to 1. After that, the spatial correlations between the group mask and the individual mask were evaluated for each subject. The spatial correlations were calculated using the voxels within the top ten slices of the mask, within the bottom ten slices of the mask, and within the whole mask, respectively, resulting in three correlation values for each subject. If a subject had correlations >0.75 for the top 10 slices, >0.55 for the bottom 10 slices, and >0.8 for the whole mask, we included this subject for further fMRI analysis. Finally, the group brain mask of each dataset was computed again based on the selected subjects' brain masks.

For sMRI, we chose the data with good quality by comparing the individual structural image with the group mean structural image. This processing was applied to each dataset's modulated or unmodulated sMRI data, separately. First, we calculated the group mean structural image across all subjects and then computed a group mask by preserving the voxels that have values greater than a constant threshold of 0.2, consistent with previous work by others[70]. Next, for each subject, we calculated spatial similarity between the group mean structural image and the individual structural image within the group mask. The spatial correlations were calculated using the voxels within the top ten slices of the mask, within the bottom ten slices of the mask, and within the whole-brain mask, resulting in three correlation values for each subject. If a subject had correlations >0.6 for the top 10 slices, >0.6 for the bottom 10 slices, and >0.8 for the whole mask, we included this subject for further sMRI analysis. Finally, the group mask of each dataset was computed again based on the selected subjects.

After the quality control, the fMRI data of 2980 subjects including 1665 HCs, 537 SZs, and 778 ASDs, the modulated sMRI data of 3148 subjects (1661 HCs, 517 SZs, and 970 ASDs), and the unmodulated sMRI data of 3374 subjects (1789 HCs, 555 SZs, and 1030 ASDs) were remained. As different criteria and thresholds were used for the quality control of fMRI and sMRI data, the selected subjects were slightly different among the three types of data (fMRI, modulated sMRI, and unmodulated sMRI). However, the sample sizes of the remaining subjects were comparable. As mentioned above, detailed information of the selected subjects is included in the Supplementary Tables S1–S4. There were no significant group differences (p-value < 0.01) in head motion for each dataset, except the head motion transition measure in the ABIDEI (p-value = 0.0084), measured by two-sample t-tests. If combining the subjects from all datasets together (totally 2980 subjects), there were some group differences in the head motion measures among the HC, SZ, and ASD groups, tested by ANOVA. However, the absolute difference across three groups in the mean motion translation was <0.05 mm and the absolute difference across the three groups in the mean motion rotation was

<0.07°. Furthermore, we regressed out the head motion effects from the neuroimaging measures before statistical analyses. As mentioned above, the fMRI data were preprocessed by regressing out six head motion parameters before the ROI-based FC estimation. Regarding functional networks estimated from ICA, motion-related noises were removed out by decomposing the data into different components including motion-related components[31]. For the FNC from ICA, we also regressed out the motion effects from the time series of functional networks before computing the FNC.

As the selected subjects had group differences in age and gender, and the data were collected from different sites, we carefully regressed out the influences of age, gender, and site effects for each subject from the estimated functional and structural measures. The regression procedure included three steps. In the first step, we regressed out the age, gender, site information, the interaction between age and site, and the interaction between gender and site for all subjects in each dataset (e.g., FBIRN). In the second step, we estimated the data effects using HCs' measures for the six datasets. The third step was further regressing out the data effect from each subject's measures that already removed the age, gender, and site effects. The processing is consistent with our previous work[29] and is also similar to some other studies[71].

**Investigating brain functional networks revealed by ICA.** Figure 1 shows the basic analysis pipeline that utilized multiple neuroimaging measures including brain functional networks, multi-scale FC, and cerebral gray matter measures to identify shared and distinct brain abnormalities between SZ and ASD. In this section, we describe how we investigated brain functional networks.

We compared the HC, SZ, and ASD groups to identify their group differences in brain functional networks that were derived by a data-driven spatial ICA method. In spatial ICA, each functional network is indicated by a spatial IC[72]. In this study, we applied our recently proposed NeuroMark pipeline[29] that leverages the GIG-ICA[30,31] with prior network templates as guidance to estimate the subject-specific brain functional networks for each of the 2980 subjects. The network templates were obtained by performing ICA on two independent large-sample groups (i.e., 823 HCs in the human connectome project and 1005 HCs in the genomics superstruct project), followed by the selection of reproducible and meaningful group-level networks as the network templates. Guided by 53 network templates, 53 corresponding subject-specific networks were obtained for each subject using a multiple-objective optimization framework in GIG-ICA. Each resulting subject-specific functional network includes brain regions with high intra-connectivity, with higher Z-scores meaning higher intra-connectivity.

We evaluated the spatial similarity across the corresponding subject-specific functional networks to verify that the functional network patterns were consistent and comparable across different subjects. In particular, for the functional networks estimated under the guidance of the same network template, we computed Pearson's correlation coefficients between any two subject-specific functional networks and then averaged all coefficients to reflect the inter-subject similarity of the functional network. After that, the inter-subject similarity values of all 53 functional networks were further summarized. To investigate whether the inter-subject similarity of functional networks is relatively stable across different data, we performed the above processing for the subjects in each dataset or each group in one dataset.

As the functional networks are comparable, we investigated abnormalities of SZ and ASD (relative to HC) in each functional network, aiming to explore which networks showed similar changes between the two disorders and which networks had disorder-unique impairments. Regarding each functional network, voxel-wise right-tailed one-sample t-tests (p < 0.01, Bonferroni (BFN) corrected) were first used to extract significant voxels showing positive Z-scores, consistent with our previous work[73]. Then, ANOVA on the three groups (p < 0.05) followed by two-tailed two-sample t-tests on any pair of groups, including HC vs. SZ, HC vs. ASD, and SZ vs. ASD (p < 0.05, false discovery rate correction), were performed to each significant voxel's Z-score in networks to identify group differences. As mentioned above, the age, gender, and site effects were regressed out prior to ANOVA and two-sample t-tests.

Next, based on the results from two-sample t-tests, we summarized the disorder-common and disorder-unique changes. In terms of the voxels passing ANOVA in each network, the voxels with T-values > 0 in both HC vs. SZ and HC vs. ASD reflected the disorder-common decreases compared to HC, and the voxels with T-values < 0 in both HC vs. SZ and HC vs. ASD reflected the disorder-common increases compared to HC. The disorder-unique changes included SZ-unique decrease (i.e., ASD-unique increase) and ASD-unique decrease (i.e., SZ-unique increase). The SZ-unique decrease involved voxels with both T-values > 0 in HC vs. SZ and T-values < 0 in HC vs. ASD. That means for these voxels, the network Z-score showed a decrease in SZ compared to HC, but showed an increase in ASD compared to HC. Similarly, the ASD-unique decrease corresponded to the voxels with both T-values < 0 in HC vs. SZ and T-values > 0 in HC vs. ASD. After that, we computed the voxel percentage for each of the four types of changes within all voxels showing group differences in ANOVA. Furthermore, among the voxels showing a disorder-common decrease, we summarized the percentage of the voxels that had weaker decreases in ASD than SZ (i.e., T-values < 0 in SZ vs. ASD), and we called the type of change as ASD-weaker decrease within the common decrease. Similarly, we obtained the associated results for ASD-weaker increase within the common increase. The procedure is outlined in Fig. 2.

**Investigating FNC revealed by ICA**. We also identified shared and distinct changes of the two disorders in FNC[74] representing the interaction between different networks. FNC matrix of each subject was obtained by computing Pearson's correlation coefficients between the post-processed time series of networks. Each time series was processed by the $Z$-score transformation, regressing motion, detrending, de-spiking, and band-filtering with 0.01–0.15 Hz before computing correlations. Thus, each element of the FNC matrix (size: $53 \times 53$) represented temporal connectivity between two functional networks. FNC strengths were transformed to Fisher's $Z$-score for further statistical analyses. For each FNC strength in the FNC matrix, we regressed out age, gender, and site effects, and then performed ANOVA on the three groups ($p < 0.01$, BFN correction) and two-tailed two-sample $t$-tests on any pair of groups to investigate group differences ($p < 0.01$, BFN correction). Similar to the functional network analyses (Fig. 2), we calculated the percentages/numbers of the disorder-common and disorder-unique FNC changes within all FNCs passing ANOVA and also summarized the percentages/numbers of the ASD-weaker FNC changes within the common FNC changes.

To examine whether group differences are sensitive to the statistical analysis method, we also conducted a permutation test (with 1000 permutations) instead of the above-mentioned direct two-sample $t$-test, to investigate group differences. Taking HC vs. SZ as an example, we introduce how the permutation test was applied. In each of 1000 permutations, we randomly rearranged the subjects (all HC and SZ subjects) into two dummy groups, each of which had the same number of subjects as the original group, and then applied two-sample $t$-tests on the dummy groups to evaluate the group differences in FNCs. After that, for each FNC, we calculated the occurring frequency of the case where the $p$-value obtained from the two-sample $t$-test using the dummy groups was smaller than the corresponding $p$-value obtained from the two-sample $t$-test using the original groups, and then took the frequency as the final $p$-value for the FNC. Smaller frequency represents a lower possibility of false positives of the identified group differences. The final $p$-values of FNCs were corrected by BFN correction.

In addition, we evaluated brain changes found using another meta-analysis, aiming to validate whether the group differences found using the whole data consistently exist when using data from separate datasets to investigate the abnormality of SZ or ASD (e.g., HC vs. SZ differences using FBIRN; HC vs. ASD differences using ABIDEI) and using data from any two datasets involving SZ and ASD to investigate their differences (e.g., SZ vs. ASD differences using SZ data from FBRIN and ASD data from ABIDEI). For this, we performed two-sample $t$-tests using separate datasets and then employed a meta-analysis to summarize the group differences from separate analyses. Supplementary Table S8 includes all our investigations about how we identified group differences using separate datasets. Taking HC vs. SZ as an example, we performed two-sample $t$-tests on FNC measures using each of the four datasets (BSNIP, FBIRN, COBRE, and MPRC) and then combined the $p$-values from the four comparisons using Fisher's method. Regarding the combined $p$-value, we performed multiple comparison correction ($p < 0.01$, BFN correction) and then show the mean $T$-value (from the four comparisons) for each FNC passing the correction.

**Investigating FC estimated using ROI-based methods**. In addition to the above data-driven functional network analysis method, we also employed a ROI-based connectivity analysis method to explore whether the differences in the FC between whole-brain regions supported a consistent relationship with the altered FNC. In this study, two brain atlases including AAL (116 regions)[37] and Brainnetome with the cerebellum (274 regions)[38,75] (http://atlas.brainnetome.org/download.html) were utilized to define ROIs separately. The ROIs were also grouped into seven functional domains for comparison with FNC measures. Regarding each atlas, we averaged the time series of voxels within each ROI to yield a representative time series for the ROI, and then the ROIs whose representative time series have very low variances and mean values across most subjects were excluded from the following analyses. Next, we obtained FC strengths between the remaining ROIs by computing Pearson's correlations using their representative time series. Thus, an FC matrix (size: $110 \times 110$ for AAL atlas and $261 \times 261$ for Brainnetome atlas) was computed for each subject. For each connectivity value in the FC matrices, we regressed out age, gender, and site effect first, and then performed ANOVA on the three groups ($p < 0.01$, BFN corrected) and two-tailed two-sample $t$-tests between any two groups to examine the group differences ($p < 0.01$, BFN correction). Finally, we obtained summary evaluations of the convergence and divergence between SZ and ASD in whole-brain FC.

**Investigating gray matter volume and density**. In addition to studying the brain functional abnormalities of the two disorders, we explored their brain structural measures and the associations between the functional and structural impairments. We included gray matter volume and density for analysis. For the gray matter volume (or density) of each voxel, ANOVA ($p < 0.05$) and two-tailed two-sample $t$-tests ($p < 0.05$, false discovery rate correction) were performed after regressing out age, gender, and site effects, to evaluate the brain structural differences between these groups. As we expected to investigate how the changes in gray matter support or complement the findings from using brain functional measures, we calculated

the disorder-common and disorder-unique percentage for each of two brain structural measures, and also summarized the brain structural changes in related brain regions according to the AAL atlas.

**Exploring the association between the neuroimaging measures and symptom scores**. To link the identified brain values with clinical symptoms, we calculated the Pearson's correlation between the values of each neuroimaging measure and the symptom severity ratings for the SZ group and the ASD group, respectively, to explore the association between network measures and symptoms. Spearman's rank correlation between them was also computed for reliability. For SZ, the symptom scores included the PANSS-positive score and PANSS-negative score. The symptoms of ASD consisted of ADOS total score and SRS. The significance level was set to $p < 0.01$ for the correlation analyses.

**Exploring the association between the neuroimaging measures and medication**. For SZ patients with available dose-level medication data, we converted all available anti-psychotic data to their respective CPZ dosage equivalents, as described by Andreasen et al.[76]. We then used a multiple linear regression model to evaluate associations between CPZ measures and each neuroimaging measure ($p < 0.05$ with BFN correction, i.e., $p$-value threshold = 0.05/(the number of measures)). For ASD patients, we compared their neuroimaging measures between medicated and unmedicated subjects using two-sample $t$-tests ($p < 0.05$ with BFN correction).

**Exploring brain changes using subjects with matched age and subjects with no motion difference**. As the onset of ASD often occurs during early childhood and the onset of SZ is more observed in adults, the data used in the above analyses had some differences in age between the two disorders. To address this issue, we also investigated group differences by only using age-matched groups (with no significant group difference in age, tested by ANOVA). More interestingly, we selected two sample sets with varied age ranges (see the Supplementary Table S9 for details). For each sample set, the HC vs. SZ, HC vs. ASD, and SZ vs. ASD differences in FNC measures were evaluated using an analysis method in Fig. 2. Finally, we compared the brain changes obtained using these age-matched subjects with the results obtained from all available subjects.

In the work, there were some group differences in head motion while combining subjects from all datasets, although in general there were no significant group differences in motion for each dataset. Therefore, from the large-size sample, we selected some subjects with no motion differences (see the Supplementary Table S10 for information) and verified the group differences using the same analysis method.

**Classifying SZ and ASD using identified brain changes**. It is important to examine whether the identified brain changes can be used as biomarkers to distinguish the two disorders. To avoid bias, we conducted two-class (SZ and ASD) classification by taking different datasets for training and the remaining datasets for testing based on the FNC measures. As there were 4 datasets relating to the SZ group and 2 datasets relating to the ASD group, in total we performed 12 classification experiments using different data assignments. Feature extraction and model training were implemented only using the training data and then the testing data were classified and compared with their true class labels. Regarding feature selection, we first implemented the above-mentioned statistical analyses on the training data, and then took the FNCs with disorder-unique changes (i.e., SZ-unique decrease and ASD-unique decrease) and the ASD-weaker changes within the disorder-common changes (i.e., ASD-weaker decrease within the common decrease and ASD-weaker increase within the common increase) as the features. A linear support vector machine with a Bayesian optimization technology[77] to optimize the parameter was applied for the model building. Finally, the classification results were evaluated using accuracy, sensitivity, and specificity.

**Reporting summary**. Further information on research design is available in the Nature Research Reporting Summary linked to this article.

## Data availability
Restrictions apply to the data use, as we applied for the use of the raw fMRI and sMRI data from the third party. We can guide how to compute the neuroimaging measures if the readers have the data use agreement from the third party.

## Code availability
The codes that support the findings of this study are available from the corresponding author [Y.D.] upon reasonable request.

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

## Acknowledgements

This work was supported by National Natural Science Foundation of China (Grant number 62076157 and 61703253, to Y.D.), Fund Program for the Scientific Activities of Selected Returned Overseas Professionals in Shanxi Province (to Y.D.), the 1331 Engineering Project of Shanxi Province of China, and the National Institutes of Health grant (Grant number R01MH118695, to V.D.C.). Special thanks to Srinivas Rachakonda, as he added the GIG-ICA method to the GIFT toolbox and provided the batch script for performing GIG-ICA more easily. We also acknowledge the FBIRN team who coordinated and performed the data acquisition including Theo G.M. van Erp, Aysenil Belger, Juan R. Bustillo, Kelvin O. Lim, Daniel S. O'Leary, Judith M. Ford, Daniel H. Mathalon, Jessica A. Turner, and Steven G. Potkin.

## Author contributions

Y.D. proposed the whole analysis framework and implemented the analyses on spatial networks from ICA, functional connectivity network, ROI-based connectivity, gray matter volume and density, as well as the association analyses between neuroimaging measures and medication/symptom scores. Y.D. drafted and revised the whole manuscript, and prepared the tables and figures. Y.D. programed the orginial code of GIG-ICA. Z.F. downloaded and organized the datasets, preprocessed the fMRI and sMRI data, performed brain mask generation and GIG-ICA on each dataset, drafted, and revised the paper. Y.X. performed the classification experiments under the guidance of Y.D. D.L. preprocessed the COBRE and FBIRN datasets, and edited the paper. A.A., M.S., and S.Q. discussed the framework and revised the paper. P.K. and L.E.H. provided the MPRC data and revised the paper. G.P. edited the paper. V.D.C. supervised the work and edited the paper. All authors have given final approval of this version of the article.

## Competing interests

The authors declare no competing interests.
