## [Peer Review File · Communications Biology]

Reviewers' comments:

Reviewer #1 (Remarks to the Author):

The authors combined data from several publicly available data sets of autism and schizophrenia to form a super-set of ~1600 healthy controls, ~500 schizophrenia (SZ) patients and 770-1000 autism spectrum disorder patients (ASD) patients to characterize the similar and distinct features of network-level activation and network-level and regional functional connectivity as well as voxel-wise gray matter volume and density.

Data were from six multi-site studies

- 1) Bipolar Schizophrenia Network for Intermediate Phenotypes BSNIP-1
- 2) Functional Biomedical Informatics Research Network (FBIRN)
- 3) Centers of Biomedical Research Excellence (COBRE)
- 4) Maryland Psychiatric Research Center (MPRC)
- 5) Autism Brain Imaging Data Exchange I (ABIDE I)
- 6) Autism Brain Imaging Data Exchange II (ABIDE II)

For the network-level functional connectivity analyses, the authors make use of a method called NeuroMark they developed, previously published in preprint, which is a modification of group information guided ICA, to identify network-level independent components from resting state functional MRI. The derived network-level time-series resulting from the use of this method are then used to calculate network-level correlation matrices which are then compared pairwise across groups (controls, and those with SZ and ASD). They conclude that both SZ and ASD exhibit lower functional connectivity within the default mode network and sensorimotor domains and increased functional connectivity between cognitive control and default mode networks. SZ and ASD unique network-level changes were observed in the interaction between visual and cognitive control networks. Both ASD and SZ groups had reduced gray matter in the anterior cingulate and insula. Some increases unique to the ASD patients were found in gray matter volume. Although there was significant overlap in the observed changes in both disorders, aberrations in the ASD were reported to be less significant.

The extant body of work comparing functional and structural changes across disorders is still relatively small, making the topic of the paper and the potential conclusions of the work of interest to the community. However, the methods and conclusions are unclear in a number of important ways described below.

1. It is not clear if the result of the NeuroMark method, used to define independent components, results in a single component for each network, the time series of which is compared across individuals in a group-wise manner, or if it results in a different component for each subject. The manuscript seems to imply it is the latter. "Guided by the 53 group-level spatial network maps, 53 corresponding subject-specific networks were obtained for each subject using a multiple-objective optimization framework in GIG-ICA. Each resulting subject-specific functional network reflects brain regions with similar activation or high intra-connectivity." However, there was no analysis of how consistent these networks were across individuals, making the subsequent comparison across diagnoses hard to interpret.

2. The authors say that their results are replicated using separate data sets. It appears that they really mean that they performed the same analysis on the data from the individual studies (BNSIP-1, BIRN, COBRE, MPRC, ABIDE I, ABIDE II) and then performed a meta-analysis to compare the results over the individual studies to that of the data set formed by the combination of all data sets. However,

the meta-analysis is not described in the methods section and the results of the analysis do not appear to be included in the paper. I think it would be interesting to determine whether any of the features identified through the NeuroMark method were sufficient to classify patient groups. Instead of a meta-analysis, a classifier could be built on some of the data and tested on other parts of the data in order to determine if these features have predictive power beyond group-wise means.

3. The authors provide no explanation of why "activation" is an important measure to consider and provide no interpretation for these results. In this section, it is not very clear what is meant by weaker changes in ASD – i.e. does the ASD group exhibit changes in fewer voxels or are the magnitude of the changes smaller. Also this part of the analysis is not mentioned in the methods section so it is not clear what regressions were performed before doing this analysis.

4. The authors use the Brainnetome atlas, one of two atlases for regional functional connectivity. The authors indicate the atlas has 274 regions and that they used a smaller subset of 261 of those. However, the Brainnetome atlas only has 246 regions.

5. Mean global signal is reportedly regressed from the time series of the regional functional connectivity analysis, but not the time series used in the groups ICA analysis. As mean global signal regression is known to induce negative correlations, it seems risky to perform this step in preprocessing data for some but not all of the analysis, and makes it more difficult to compare the results across them.

6. The authors did not provide any motivation for the choice of parameters used to determine successful normalization of the imaging data in their quality control procedure. Different values were chosen for different imaging modalities, which appear to have resulted in a different number of subjects for different analyses even within the structural-only analysis:
modulated structural images 3148: 1661 controls, 517 SZ and 970 ASD
unmodulated structural images 3374; 1789 controls, 555 SZ and 1030 ASD
The authors should provide further explanation as to the reason these choices were made.

7. Several places throughout the results section, percentages are discussed and it is not always clear what they refer to.

8. Weaker changes in the ASD group are reported in relation to the functional connectivity results as well and could be made more clear. Do weaker changes indicate fewer changes or smaller differences in correlation or both?

9. In my opinion stacked bar plots are difficult to read. In this case I do not feel bar plots are necessary, as the authors appear to report the relative number of subjects in each portion of the analysis in an easily interpretable table.

10. The BSNIP-1 data set contained subjects who were also bipolar. Perhaps these subjects should have been removed from consideration as they add to the heterogeneity of the schizophrenia group.

Reviewer #2 (Remarks to the Author):

The authors investigated similarities and differences in resting-state fMRI intra-network activation and structural MRI grey and white matter density and volume in a large group of individuals with autism and individuals with schizophrenia compared to neurotypical controls. Common networks and structures were impaired in autism and in schizophrenia, but were more impaired in schizophrenia compared to autism. This is an interesting topic, and the large sample size and sophisticated methods will greatly benefit the field. Identifying what is common across conditions and what is pathology-specific assists with identifying biomarkers for identification and treatment.

However, the big-picture issues that drive the rationale for a study like this are currently lost in the writing. The main contributor to this is that the results and discussion sections read like a list of structures and networks that are abnormal in one condition or in both. Could the authors please add some context to the results to make this easier to read?

For example, what do abnormalities in these structures/networks reveal for understanding autism and/or schizophrenia? What has been previously identified as being related to specific behaviors? Do these indicate potential biomarkers? How are these findings useful? Some of this begins to be addressed in sections such as "SZ and ASD: unique abnormalities", but the discussion still contains vague phrasing such as "communication between self-related processing and other functions... may underlie the differences between SZ and ASD". Can the authors be more concrete in what they are referring to?

In short, relating these results to what has been found previously and what still needs to be done will assist in highlighting why these findings are interesting (which they are).

I have a couple of other specific points.

Introduction:

- I am missing the difference between hypothesis 1 compared to 2 and 3?
- The sentences following "However, there are notable differences between populations afflicted with the two disorders." Needs references to support the claims.
- Other references to include directly comparing autism and schizophrenia that may help with adding context to the rationale and discussion points:

Eack, S. M., Bahorik, A. L., McKnight, S. A. F., Hogarty, S. S., Greenwald, D. P., Newhill, C. E., ...

Minshew, N. J. (2013). Commonalities in social and non-social cognitive impairments in adults with autism spectrum disorder and schizophrenia. *Schizophrenia Research*, 148(1-3), 24-28.

<https://doi.org/http://dx.doi.org/10.1016/j.schres.2013.05.013>

Eack, S. M., Wojtalik, J. A., Keshavan, M. S., & Minshew, N. J. (2017). Social-cognitive brain function and connectivity during visual perspective-taking in autism and schizophrenia. *Schizophrenia Research*, 183, 102-109. <https://doi.org/10.1016/j.schres.2017.03.009>

Haigh, S. M., Eack, S. M., Keller, T., Minshew, N. J., & Behrmann, M. (2019). White matter structure in schizophrenia and autism: Abnormal diffusion across the brain in schizophrenia. *Neuropsychologia*, 135, 107233. <https://doi.org/https://doi.org/10.1016/j.neuropsychologia.2019.107233>

Haigh, S. M., Gupta, A., Barb, S. M., Glass, S. A. F., Minshew, N. J., Dinstein, I., ... Behrmann, M. (2016). Differential sensory fMRI signatures in autism and schizophrenia: Analysis of amplitude and trial-to-trial variability. *Schizophrenia Research*, 175(1-3), 12-19.

<https://doi.org/http://dx.doi.org/10.1016/j.schres.2016.03.036>

Methods:

- The correlation cutoffs for fMRI and sMRI differ. What was the reason? How were these thresholds obtained?
- Did motion parameters significantly differ between groups?
- While the different datasets were included as a factor in the analyses, the datasets themselves were comprised from data from multiple sites. Was this accounted for in the analyses too?
- Similarly, while comparisons between autism and controls were (roughly) age matched, and similar with schizophrenia and their controls, autism and schizophrenia individuals were widely different in age. Age was included in the analysis; however, comparing a majority pediatric population to an all-adult population seems a little strange regardless of the analysis conducted. The ABIDE datasets include a number of adults with autism. Could the authors match a subset of the autism group to schizophrenia to verify that their results still hold?

Results:

- Reducing the number of acronyms will help with the readability.
- Not sure what sentences like this mean: "Notably, within the common changes, ASD in general presented weaker changes than SZ (Fig. 3(B)), with 77% and 90% of voxels in the common decreases and increases, respectively (Table 1)."
- A suggestion for a way to make the results easier to follow is to include subsections to individuate when discussing common findings across autism and schizophrenia and what findings are diagnosis specific. Similarly, separating the description of which networks showed increased compared to decreased activity might help.
- The analysis with symptoms measures seems to suggest that worse default-mode activity relates to worse symptoms rather than anything specific. Is this the case? Could the authors include the direction of the correlations so that the reader does not have to go to the supplementary materials section to be able to tell.

Discussion:

- Limitations and future directions missing

Reviewer #3 (Remarks to the Author):

The manuscript submitted by Du et al takes on a very interesting problem related to the common and unique variation explained across autism spectrum disorder (ASD) and schizophrenia (SZ) using functional and structural MRI data. The authors leverage the leverages the group information guided ICA that allows uses network templates as priors to estimate the subject-specific spatial functional network. The authors first extract network level impairment – and for each network the authors use a t-test to extract networks that show a positive activation, followed by an ANOVA to determine group level differences. They further examine group differenced in functional network connectivity using the ICA components. While I recognize that there may be some interesting work here, I have significant reservations about the methods and how they were applied. Further the reporting of the main results is very confusing and hard to follow.

- 1) From a very basic standpoint there is significant evidence that data quality will impact the measures that the authors choose to use here. There is really good evidence that these data will poorly impact all downstream measures. I am somewhat satisfied with how the authors treated the fMRI analysis – however, I would advise them to perhaps do what has been done in the Satterthwaite papers and do a sensitivity analysis examining what happens when some of the data is fed back into the analysis. Further, it would be good if the authors cited some of the best practices that they chose to follow. On the structural side, however, I would imagine that the authors should have done some manual quality control on their data. There is plenty of evidence that this is important, especially in the context of ASD and younger populations. A recent paper by Bedford et al (PMID: PMID: 31028290) clearly demonstrates the importance of detailed quality control and its impact on analytical outputs. Work by Pardoe et al (PMID: PMID: 27153982) further motivates this significant need. The ad-hoc heuristic used here, does not seem sufficient.
- 2) Other groups have provided some overlap of neural phenotypes across disorders. The work by Park et al. (PMID: 2968887), Stefanik (PMID: PMID: 29105664), and Yoshihara (PMID: 32300809) all uses a network-related methods and are not discussed. The authors may want to look into others.
- 3) The segmentation method used for the VBM-style analysis is not referenced and is poorly described.
- 4) Dealing with site as nuisance variable is problematic. This may simply provide a "statistical" double hit where samples are different along sex and age – which is more than likely to be the case in SZ and ASD given the differences in age of onset. It would be more appropriate to use a linear mixed effects model with site as a random effect, some sort of meta-analytic technique, or COMBAT.
- 5) The methods going from individual ICA components to group-level differences extremely hard to follow. I would suggest, at least, that the authors consider adding figures to the methods to better

describe what is going on in this context.

6) The figures are really hard to parse. The circle plots are tiny and are difficult to understand without labelling and nudging towards the parts of the plots that you expect the readers to take away information from. Fig 5 does not convince me that these results survive across different parcellation techniques. Fig 6 needs to have the colour bars clearly labelled or to have one common colour bar across the figure. Given what I understand of figure 6 - it seems unlikely that there is significant morphological overall between disorders. I think this needs to be better described. These figs are the "meat" of the paper and without them being clearer, I find it hard to believe the take-away message.

Response letter for manuscript COMMSBIO-20-0724-T

Evidence of shared and distinct functional and structural brain signatures in schizophrenia and autism spectrum disorder

Yuhui Du, Zening Fu, Ying Xing, Dongdong Lin, Godfrey Pearlson, Peter Kochunov, L Elliot Hong, Shile Qi, Mustafa Salman, Anees Abrol, Vince D. Calhoun

Reviewer #1:

The authors combined data from several publicly available data sets of autism and schizophrenia to form a super-set of ~1600 healthy controls, ~500 schizophrenia (SZ) patients and 770-1000 autism spectrum disorder patients (ASD) patients to characterize the similar and distinct features of network-level activation and network-level and regional functional connectivity as well as voxel-wise gray matter volume and density.

Data were from six multi-site studies

- 1) Bipolar Schizophrenia Network for Intermediate Phenotypes BSNIP-1
- 2) Functional Biomedical Informatics Research Network (FBIRN)
- 3) Centers of Biomedical Research Excellence (COBRE)
- 4) Maryland Psychiatric Research Center (MPRC)
- 5) Autism Brain Imaging Data Exchange I (ABIDE I)
- 6) Autism Brain Imaging Data Exchange II (ABIDE II)

For the network-level functional connectivity analyses, the authors make use of a method called NeuroMark they developed, previously published in preprint, which is a modification of group information guided ICA, to identify network-level independent components from resting state functional MRI. The derived network-level time-series resulting from the use of this method are then used to calculate network-level correlation matrices which are then compared pairwise across groups (controls, and those with SZ and ASD). They conclude that both SZ and ASD exhibit lower functional connectivity within the default mode network and sensorimotor domains and increased functional connectivity between cognitive control and default mode networks. SZ and ASD unique network-level changes were observed in the interaction between visual and cognitive control networks. Both ASD and SZ groups had reduced gray matter in the anterior cingulate and insula. Some increases unique to the ASD patients were found in gray matter volume. Although there was significant overlap in the observed changes in both disorders, aberrations in the ASD were reported to be less significant.

The extant body of work comparing functional and structural changes across disorders is still relatively small, making the topic of the paper and the potential conclusions of the work of interest to the community. However, the methods and conclusions are unclear in a number of important ways described below.

Response:

We appreciate your favorable evaluation. To address your concerns, we have added some experiments and significantly improved our manuscript. Please see the following responses for the details.

1. It is not clear if the result of the NeuroMark method, used to define independent components, results in a single component for each network, the time series of which is compared across individuals in a group-wise manner, or if it results in a different component for each subject. The manuscript seems to imply it is the latter. “Guided by the 53 group-level spatial network maps, 53 corresponding subject-specific networks were obtained for each subject using a multiple-objective optimization framework in GIG-ICA. Each resulting subject-specific functional network reflects brain regions with similar activation or high intra-connectivity.” However, there was no analysis of how consistent these networks were across individuals, making the subsequent comparison across diagnoses hard to interpret.

Response:

Thank you for your comment.

We want to explain that our previously proposed NeuroMark (Du, Fu et al. 2020) framework is an extended group information guided independent component analysis (GIG-ICA) that can analyze multi-subject fMRI data with prior network templates as guidance. For each subject, NeuroMark yields multiple subject-specific brain functional networks (each of which is represented by one independent component) and the networks associated time series (each of which corresponds to the time course of one component). In our work, 53 functional networks were obtained for each subject, and then 53 time series reflecting the fluctuations of those functional networks were used to compute the functional network connectivity (FNC) of each subject by computing the Pearson correlations between time series. The functional network and FNC measures were used for the statistical analyses.

Using NeuroMark, the inter-subject correspondence of subject-specific functional networks can be achieved by utilizing the multi-objective optimization of GIG-ICA, since the multi-objective optimization highlights the similarity between each network template and individual-level network, at the same time optimizes the independence of each individual-level network. In our previous work, we evaluated GIG-ICA using the simulations and test-retest fMRI data (Du and Fan 2013, Du, Allen et al. 2016, Du, Lin et al. 2017), which showed its effectiveness in estimating individual-level components (functional networks). Using simulations, our previous studies demonstrated that individual-level independent components can be estimated with higher accuracy than other comparative group ICA methods, under the conditions with different parameter settings. Using test-retest fMRI data, our method yielded higher intra class coefficients (ICCs) in the estimated networks than traditional independent vector analysis (IVA) method. In our another work (Salman,

Du et al. 2019), our results supported that GIG-ICA can result in higher classification accuracy than the dual regression method in distinguishing schizophrenia patients from healthy controls.

To further address your concern, we have also added some experiments in this paper to show that the individual-level networks were comparable and consistent across different subjects.

The detailed processing is described in “Methods” section: “We evaluated the spatial similarity across the corresponding subject-specific functional networks to verify that the functional network patterns were consistent and comparable across different subjects. Particularly, for the functional networks estimated under the guidance of a same network template, we computed Pearson correlation coefficients between any two subject-specific functional networks, and then averaged all coefficients to reflect the inter-subject similarity of the functional network. After that, the inter-subject similarity values of all 53 functional networks were further summarized. To investigate if the inter-subject similarity of functional networks is relatively stable across different data, we performed the above processing for the subjects in each dataset or each group in one dataset.”

The results are included in the subsection “SZ and ASD show shared and distinct changes in brain functional networks” of “Results” section. The descriptions are: “We evaluated the spatial similarity of subject-specific functional networks and show the results in the supplementary Fig. S2, supporting that each functional network (estimated from one same network template) was comparable across different subjects and feasible for further statistical analyses among groups. In addition, the network correspondence was relatively stable across different datasets and groups.”

Fig. S2 Inter-subject similarity of functional networks is shown in (A) for all subjects in each dataset and (B) for the subjects in each group of dataset, using errorbars. For each dataset or each group in one dataset, the inter-subject similarity of 53 networks is shown using an errorbar.

2. The authors say that their results are replicated using separate data sets. It appears that they really mean that they performed the same analysis on the data from the individual studies (BNSIP-1, BIRN, COBRE, MPRC, ABIDE I, ABIDE II) and then performed a meta-analysis to compare the results over the individual studies to that of the data set formed by the combination of all data sets. However, the meta-analysis is not described in the methods section and the results of the analysis do not appear to be included in the paper. I think it would be interesting to determine whether any of the features identified through the NeuroMark method were sufficient to classify patient groups. Instead of a meta-analysis, a classifier could be built on some of the data and tested on other parts of the data in order to determine if these features have predictive power beyond group-wise means.

Response:

Thank you for your comments and suggestions.

(1) In our previously submitted paper, we described the analysis procedure and results relating to the separate datasets in the “Methods” subsection “Comparing functional network connectivity

revealed by ICA” and the supplementary materials. Considering your concern, we have added more details in the updated paper. In the current supplementary S3 section, we added the following sentences:

“Since the primary analyses in our study used all subjects’ data that were available from six datasets (BSNIP, FBIRN, COBRE, MPRC, ABIDEI and ABIDEII), it is necessary to validate if the group differences found using the whole data consistently exist when using data from each dataset to investigate the abnormality of SZ or ASD (e.g., HC vs. SZ differences using FBIRN; HC vs. ASD differences using ABIDEI) and using data from any two datasets involving SZ and ASD to investigate their differences (e.g., SZ vs. ASD differences using SZ data from FBIRN and ASD data from ABIDEI). So, we investigated group differences by performing two-sample t-tests using separate datasets and then employed a meta-analysis to summarize the group differences from separate analyses. Table S6 includes all our investigations about how we identified group differences using separate datasets. Taking HC vs. SZ for example, we performed two-sample t-tests on FNC measures using each of the four datasets (BSNIP, FBIRN, COBRE, and MPRC), and then combined the p-values from the four comparisons using Fisher’s method. Regarding the combined p-value, we performed multiple comparison correction ($p < 0.01$, Bonferroni correction) and then show the mean T-value (from the four comparisons) for each FNC passing the correction. Fig. S3 demonstrates the combined group differences for HC vs. SZ, HC vs. ASD, and SZ vs. ASD. We found that group differences in Fig. S3 showed a similar pattern with that in Fig. 4, supporting that the identified overlap and uniqueness of brain abnormality were relatively reliable.”

(2) According to your suggestions, we have added 12 classification experiments to assess the distinguishing ability of our estimated network measures using different datasets. Our results support that the neuroimaging measures can classify SZ and ASD well.

We added one subsection “Classifying SZ and ASD using identified brain changes” to the “Methods” section. The added contents are included here for your reference.

“It is important to examine whether the identified brain changes can be used as biomarkers to distinguish the two disorders. To avoid bias, we conducted two-group (SZ and ASD) classification by taking different datasets for training and the remaining datasets for testing based on FNC measures. Since there were four datasets relating to the SZ group and two datasets relating to the ASD group, totally we performed 12 classification experiments using different data assignments. Feature extraction and model training were implemented only using the training data, and then the testing data were classified and compared with their true class labels. In our work, we extracted the disorder-unique and ASD-weaker common FNC changes for every two groups (SZ and ASD) using the above mentioned statistical analyses, and then used the union set of those changes as the features. Linear support vector machine with a Bayesian optimization technology (A Snoek, A Larochelle et al. 2012) to optimize the parameter was applied for the model building. Finally, the classification results were evaluated using accuracy, sensitivity, and specificity.”

In the “Results” section, we added the following texts in a new subsection “Brain changes can successfully distinguish SZ and ASD”:

“We were also interested in whether the identified brain changes can be used as promising biomarkers to distinguish the two disorders. Our results support that FNC measures performed well in classifying SZ and ASD patients. Table 2 shows the classification results that were obtained by taking different datasets as training data/testing data (totally 12 classification runs). The mean accuracy, sensitivity and specificity across all 12 classification runs were 75%, 83%, and 63%, respectively. The best results reached up to 80.0% accuracy, 90.0% sensitivity, and 68.0% specificity, even only using features from statistical analyses. The most frequently used features in all classification runs included the unique changes of connectivity between SC (e.g. putamen, caudate, and thalamus) and SM (e.g. superior parietal lobule, paracentral lobule) domains, and also included the ASD-weaker common changes of connectivity between SC and VI/CB domains and between SM and VI domains. In our work, relatively lower specificity (compared to sensitivity) meant that SZ patients were more likely misdiagnosed as ASD using these connectivity measures.”

Table 2. Classification result evaluation of distinguishing SZ and ASD patients using different datasets as the training and testing data.

Training data		Testing data		Accuracy	Sensitivity	Specificity
SZ datasets (N)	ASD dataset (N)	SZ datasets (N)	ASD dataset (N)			
BSNIP & FBIRN (319)	ABIDEI (398)	COBRE & MPRC (218)	ABIDEII (380)	75.8%	84.2%	61.0%
BSNIP & COBRE (250)	ABIDEI (398)	FBIRN & MPRC (287)	ABIDEII (380)	73.9%	85.3%	58.9%
BSNIP & MPRC (332)	ABIDEI (398)	FBIRN & COBRE (205)	ABIDEII (380)	78.1%	80.5%	73.7%
FBIRN & COBRE (205)	ABIDEI (398)	BSNIP & MPRC (332)	ABIDEII (380)	67.4%	88.7%	43.1%
FBIRN & MPRC (287)	ABIDEI (398)	BSNIP & COBRE (250)	ABIDEII (380)	78.9%	84.7%	70.0%
COBRE & MPRC (218)	ABIDEI (398)	BSNIP & FBIRN (319)	ABIDEII (380)	80.0%	90.0%	68.0%
BSNIP & FBIRN (319)	ABIDEII (380)	COBRE & MPRC (218)	ABIDEI (398)	77.9%	80.2%	73.9%
BSNIP & COBRE (250)	ABIDEII (380)	FBIRN & MPRC (287)	ABIDEI (398)	77.2%	85.4%	65.9%
BSNIP & MPRC (332)	ABIDEII (380)	FBIRN & COBRE (205)	ABIDEI (398)	73.1%	76.4%	66.8%

FBIRN & COBRE (205)	ABIDEII (380)	BSNIP & MPRC (332)	ABIDEI (398)	70.4%	88.2%	49.1%
FBIRN & MPRC (287)	ABIDEII (380)	BSNIP & COBRE (250)	ABIDEI (398)	73.8%	78.1%	66.8%
COBRE & MPRC (218)	ABIDEII (380)	BSNIP & FBIRN (319)	ABIDEI (398)	71.7%	77.9%	63.9%

Footnote: N represents the subject number.

In the “Discussions” section, we also further summarized the point as below:

“We also found that the disorder-unique and ASD-weaker common abnormality of SZ and ASD performed well in distinguishing the two disorders. Satisfactory performances were reliably achieved even under the situation of different datasets as the training data/testing data with varied sample sizes. The connectivity between the sub-cortical domain and other domains such as sensorimotor, visual and cerebellar areas as well as the connectivity between the sensorimotor and visual domains played an important role in distinguishing the two disorders, which indicates potential biomarkers between the two disorders. To the best of our knowledge, this is the first study applying large-sample multi-site fMRI data to perform the direct classification between SZ and ASD. Our work achieved relatively higher classification accuracy than a previous study (Mastrovito, Hanson et al. 2018) that also directly classified the two disorders using functional connectivity features. They trained a classifier on 72 SZs and 37 ASDs and resulted in 75% classification accuracy on independent 5 SZ and 27 ASD patients. A recent study from Yoshihara (Yoshihara, Lisi et al. 2020) employed functional connectivity features to explore the complex relationship between SZ and ASD as well. In their work, dual classifiers were applied to discriminate ASD (or SZ) from HC so as to investigate the two disorders using a dimensional method, demonstrating the overlapping but asymmetrical relationship between ASD and SZ. Interestingly, they found that SZ subjects showed increased classification certainty for the ASD dimension while the ASD subjects did not for the SZ dimension. The findings are consistent to our classification results to some degree, as SZ was more likely to be grouped into ASD than the opposing situation in our classification experiments.”

3. The authors provide no explanation of why “activation” is an important measure to consider and provide no interpretation for these results. In this section, it is not very clear what is meant by weaker changes in ASD – i.e. does the ASD group exhibit changes in fewer voxels or are the magnitude of the changes smaller. Also this part of the analysis is not mentioned in the methods section so it is not clear what regressions were performed before doing this analysis.

Response:

Sorry for the confusion.

(1) In our paper, the brain functional networks were estimated using an ICA-based pipeline (i.e. NeuroMark) (Du, Fu et al. 2020). ICA has been very successful in identifying functional networks and network-based biomarkers in mental disorders. Using ICA, each brain functional network is represented by one spatial independent component (IC) including Z-score for each brain voxel. In each functional network, voxels with higher Z-scores have relatively higher activation (or intra-connectivity) in the network. We have added descriptions to highlight the importance of functional network analysis using ICA. Considering your concern, we changed “activation” to “intra-connectivity”. The descriptions are included in the “Results” section. We also include them as below for your convenience.

“By using NeuroMark, each brain functional network is represented by one spatial independent component (IC) including Z-score of each brain voxel, in which the voxels with greater Z-scores have relatively higher intra-connectivity in network. Many studies have employed the voxel-wise Z-scores as intra-network connectivity measures to study brain abnormality in patients relative to healthy controls (Ongur, Lundy et al. 2010, Baggio, Segura et al. 2015) and distinguish patients with different disorders (Du, Pearlson et al. 2015, Osuch, Gao et al. 2018, Salman, Du et al. 2019). In this work, comprehensive statistical analyses were performed to investigate the shared and distinct changes between SZ and ASD (see Fig. 2 for detailed procedure).”

(2) In our updated paper, we have revised the descriptions to clarify the statistical analyses. The ASD-weaker changes means that the magnitude of ASD changes (relative to HC) was smaller than that of SZ changes. The following sentences and one pipeline figure (Fig. 2) have been added to the “Investigating brain functional networks revealed by ICA” subsection for addressing your concern.

The statements are: “Next, based on the results from two-sample t-tests, we summarized the disorder-common and disorder-unique changes. In terms of the voxels passing ANOVA in each network, the voxels with T-values > 0 in both HC vs. SZ and HC vs. ASD reflected the disorder-common decreases, and the voxels with T-values < 0 in both HC vs. SZ and HC vs. ASD reflected the disorder-common increases. The disorder-unique changes included SZ-unique decrease (i.e. ASD-unique increase) and ASD-unique decrease (i.e. SZ-unique increase). The SZ-unique decrease involved voxels with both T-values > 0 in HC vs. SZ and T-values < 0 in HC vs. ASD. Similarly, the ASD-unique decrease corresponded to the voxles with both T-values < 0 in HC vs. SZ and T-values > 0 in HC vs. ASD. After that, we computed the voxel percentage for each of the four types of changes within all voxels showing group differences in ANOVA. Furthermore, among the voxles showing common decrease, we summarized the percentage of the voxels that had weaker decreases in ASD than SZ (i.e. T-values < 0 in SZ vs. ASD), and we called the type of change as ASD-weaker decrease within the decrease overlap. Similarly, we obtained the associated results for ASD-weaker increase within the increase overlap. The procedure is outlined in Fig. 2.”

In the updated subsection “SZ and ASD show shared and distinct changes in brain functional

networks”, we revised the texts as “Notably, for the voxels with commonly decreased changes, 85.3% voxels showed weaker (smaller) changes in ASD than SZ; for the voxels with commonly increased changes, 94.4% voxels had weaker (smaller) changes in ASD than SZ, supporting that in general ASD presented weaker changes than SZ for the shared abnormalities.”

Fig. 2 Statistical analysis outline for identifying and summarizing the shared and distinct brain abnormalities of SZ and ASD. The statistical analysis method is consistent for all neuroimaging measures, including functional networks, functional network connectivity, functional connectivity, and gray matter volume and density. Regarding each measure, ANOVA and two-tailed two-sample t-tests were first performed, resulting in group differences for HC vs. SZ, HC vs. ASD, and SZ vs. ASD. For the measures passing ANOVA, different types of changes were then summarized, including the common decreases (the voxels with T-values > 0 in both HC vs. SZ and HC vs. ASD), the common increases (the voxels with T-values < 0 in both HC vs. SZ and HC vs. ASD), the SZ-unique decrease (the voxels with both T-values < 0 in HC vs. ASD and T-values > 0 in HC vs. SZ), and the ASD-unique decrease (the voxles with both T-values < 0 in HC vs. SZ and T-values > 0 in HC vs. ASD). The percentage of each type of change was calculated as the number of neuroimaging measures relating to the change divided by the number of

measures passing ANOVA. Among the neuroimaging measures with the common decrease (or increase), the percentage of ASD-weaker decrease (or increase) that showed T-values < 0 in SZ vs. ASD in the common decrease (or showed T-values > 0 in SZ vs. ASD in the common increase) were further summarized.

(3) In the “Methods” section, we have updated the regression processing about how we removed the effects of age, gender and site effects. The sentences are: “Since the selected subjects had group differences in age and gender and the data were collected from different sites, we carefully regressed out the influences of age, gender and site effects for each subject from the estimated functional and structural measures. The regression procedure included three steps. In the first step, we regressed out the age, gender, site information, the interaction between age and site, and the interaction between gender and site for all subjects in each dataset (e.g. FBIRN). In the second step, we estimated the data effects using HCs’ measures for the six datasets that were compared. The third step was further regressing out the data effect from each subject’s measures that were already removed the age, gender and site effects. The processing is consistent to our previous work (Du, Fu et al. 2020), and is also similar to some other studies (Nakano, Takamura et al. 2020).”

4. The authors use the Brainnetome atlas, one of two atlases for regional functional connectivity. The authors indicate the atlas has 274 regions and that they used a smaller subset of 261 of those. However, the Brainnetome atlas only has 246 regions.

Response:

Thank you for your comment.

In the Brainnetome atlas, there are 274 regions when including cerebellum. We downloaded the BN_Atlas_274_combined.nii.gz from the official website (<http://atlas.brainnetome.org/download.html>). We have added some references in the paper to address your concern. The sentences are: “In this paper, two brain atlas templates including Automated Anatomical Labeling (AAL) (116 regions) (Tzourio-Mazoyer, Landeau et al. 2002) and Brainnetome with cerebellum (274 regions) (Jiang 2013, Fan, Li et al. 2016) (<http://atlas.brainnetome.org/download.html>) were utilized to define ROIs separately.”

To guarantee the data quality, the ROIs whose representative time series have very low variances and mean values across most subjects were excluded from the analyses, resulting in final 261 ROIs.

5. Mean global signal is reportedly regressed from the time series of the regional functional connectivity analysis, but not the time series used in the groups ICA analysis. As mean global signal regression is known to induce negative correlations, it seems risky to perform this step in

preprocessing data for some but not all of the analysis, and makes it more difficult to compare the results across them.

Response:

Thank you for your comment.

In studying functional network (or connectivity), the preprocessing steps on fMRI data are often slightly different between using the data-driven methods (such as ICA) and using the hypothesis-based methods (such as ROI-based method) due to different method properties. Consistent with many previous studies (Allen, Erhardt et al. 2011, Chen and Calhoun 2018), we used the spatially smoothed data as input for our ICA method, as ICA can denoise the data by decomposing artifacts as independent components (like head motion-related component) (Zuo, Kelly et al. 2010, Du and Fan 2013). Compared to ICA method, the preprocessing of fMRI data is usually stricter for the ROI-based functional connectivity analysis, because the time series of ROIs are directly used for computing correlations. In our work, the smoothed data were further detrended and band-pass filtered (0.01-0.15Hz), followed by regressing out nuisance covariates including six head motion parameters, white matter signal, cerebrospinal fluid signal, and global mean signal, before the ROI-based functional connectivity analysis.

We also want to point out that in using ICA we performed additional preprocessing on the time courses of functional networks before computing functional network connectivity, and the preprocessing steps also including detrending, filtering, regressing motion were similar to the preprocessing for ROI-based analysis. The preprocessing of the time series of functional networks is described in subsection “Investigating functional network connectivity revealed by ICA”. The sentences are “Each time series was processed by the Z-score transformation, regressing motion, de-trending, de-spiking, and band-filtering with 0.01-0.15 Hz prior to computing correlations.”

As you noted, we regressed out the global mean signal before ROI-based analysis, as previous work (Lydon-Staley, Ciric et al. 2019) suggested that the use of global signal regression (GSR) in preprocessing pipelines is effective at reducing the association between motion and functional connectivity. For ICA-based functional network connectivity, we did not implement GSR as it is unusual to perform global mean signal regression on time series of functional networks.

We confess that it is somewhat difficult to compare functional connectivity under different scales, estimated using different methods. However, we hope to provide insights by linking the results from different methods, so we assigned the brain atlas-related ROIs to different functional domains, consistent with the assignment of ICA components. Our results support that the findings are relatively reliable. But, considering your comments, we have added some discussions to point out this limitation in our updated paper. The texts are: “The third shortcoming lies in the comparison among different neuroimaging measures. In our work, the data-driven and hypothesis-derived functional connectivity measures were analyzed and compared with each other. However, it was not

easy to compare the functional connectivity results due to their various parcellations and preprocessing procedures. In addition, we only summarized the association between functional and structural changes according to the affected brain regions. Fusion or more advanced methods (Park, Raznahan et al. 2018) may be better for cross-modal link in a direct manner.”

6. The authors did not provide any motivation for the choice of parameters used to determine successful normalization of the imaging data in their quality control procedure. Different values were chosen for different imaging modalities, which appear to have resulted in a different number of subjects for different analyses even within the structural-only analysis:

modulated structural images 3148: 1661 controls, 517 SZ and 970 ASD

unmodulated structural images 3374; 1789 controls, 555 SZ and 1030 ASD

The authors should provide further explanation as to the reason these choices were made.

Response:

Thank you for your questions. In our work, we used different thresholds for the quality control in selecting subjects for fMRI and sMRI respectively, as the two modalities have different properties and we wanted to keep the remaining subjects to have balanced sample sizes. The details about quality control are described in supplementary “S1. Quality control on preprocessed fMRI and sMRI data”.

Since fMRI data were analyzed by an ICA-based method (i.e. NeuroMark), the consistency between individual brain mask and group brain mask is important. In the Group ICA of fMRI Toolbox (GIFT) (<https://trendscenter.org/software/gift/>) that is a widely-applied ICA-based fMRI analysis toolbox, a group brain mask is usually obtained based on the individual brain mask and then fMRI data within the group brain mask are used for ICA. As such, the similarity between the individual brain mask and group brain mask can be used to reflect the quality of individual fMRI data and select the subjects. In this paper, the computation methods of the individual and group brain masks are all the same to that used in GIFT. In particular, we used three correlations to reflect the similarity measures between the two kinds of mask for the top, bottom, and whole parts in brain. Different thresholds were used for the three correlations as they corresponded to different parts in brain.

For the sMRI data, our subject selection method was also based on previous experiences. Researchers often used the group mean structural image to determine a group mask for multi-subject sMRI data analysis. In this paper, we used a threshold 0.2, consistent with previous work by others (Ashburner 2015, Zhu, Zhang et al. 2020) to generate the group mask. We used the similarity between the individual structural data and group mean structural data within the group mask to reflect the quality of each subject’s sMRI data. Similar to fMRI data, we also computed three correlations for measuring their similarity in the top, bottom and whole parts. It is worth noting that

the three thresholds associated the three similarity measures were set to different values between fMRI and sMRI due to their data diversity and their difference in computing correlations. Different thresholds were used in order to make the selected sample size comparable across fMRI, modulated sMRI and unmodulated sMRI. The resulting modulated and unmodulated structural images had slightly different subject numbers after the subject selection, because we used the same thresholds for the same modality (i.e. sMRI). We believe our subject selection methods are useful for determining high-quality data and brain masks for analysis.

To address your concerns, we have added more details and explanations into the subsection “Data and preprocessing” of the “Methods” section. The texts are: “Since different criteria and thresholds were used for the quality control of fMRI and sMRI data, the selected subjects were slightly different among the three types of data (fMRI, modulated sMRI, and unmodulated sMRI). However, the sample sizes of the remaining subjects were comparable.”

7. Several places throughout the results section, percentages are discussed and it is not always clear what they refer to.

Response:

Thank you for your comment.

As mentioned before, we have revised the descriptions and added Fig. 2 to clarify the statistical analysis steps and explained how we computed the percentages in more detail. The percentage of each type of change, including common decreases/increases between ASD and SZ, ASD-unique decreases (i.e. SZ-unique increases), and SZ-unique decreases (i.e. ASD-unique increases), was calculated as the number of measures relating to the change divided by the number of measures passing ANOVA. Among the neuroimaging measures with the common decrease (or increase), the percentage of ASD-weaker decrease (or increase) was calculated using the neuroimaging measures that showed T-values < 0 in SZ vs. ASD in the common decrease (or measures that showed T-values > 0 in SZ vs. ASD in the common increase).

8. Weaker changes in the ASD group are reported in relation to the functional connectivity results as well and could be made more clear. Do weaker changes indicate fewer changes or smaller differences in correlation or both?

Response:

Thank you for your comment.

We have carefully revised the subsection “SZ and ASD show shared and distinct changes in functional network connectivity”. The updated descriptions are: “Notably, regarding more than 90% of the commonly changed FNCs that showed similar changing trends in SZ and ASD relative to HC,

ASD showed weaker (smaller) changes than SZ in each of the FNCs. The finding was consistent with that from the functional network analyses.”

9. In my opinion stacked bar plots are difficult to read. In this case I do not feel bar plots are necessary, as the authors appear to report the relative number of subjects in each portion of the analysis in an easily interpretable table.

Response:

According to your suggestion, we have removed the original Fig. S1. Thanks.

10. The BSNIP-1 data set contained subjects who were also bipolar. Perhaps these subjects should have been removed from consideration as they add to the heterogeneity of the schizophrenia group.

Response:

Thank you for the question. In this work, we did not include the patients with psychotic bipolar disorders from the BSNIP-1 data for avoiding the heterogeneity. We only included the schizophrenia group for the comparisons with autism spectrum disorder group.

Thank you very much for your review and good suggestions!

Reviewer #2

The authors investigated similarities and differences in resting-state fMRI intra-network activation and structural MRI grey and white matter density and volume in a large group of individuals with autism and individuals with schizophrenia compared to neurotypical controls. Common networks and structures were impaired in autism and in schizophrenia, but were more impaired in schizophrenia compared to autism. This is an interesting topic, and the large sample size and sophisticated methods will greatly benefit the field. Identifying what is common across conditions and what is pathology-specific assists with identifying biomarkers for identification and treatment. However, the big-picture issues that drive the rationale for a study like this are currently lost in the writing. The main contributor to this is that the results and discussion sections read like a list of structures and networks that are abnormal in one condition or in both. Could the authors please add some context to the results to make this easier to read? For example, what do abnormalities in these structures/networks reveal for understanding autism and/or schizophrenia? What has been previously identified as being related to specific behaviors? Do these indicate potential biomarkers? How are these findings useful? Some of this begins to be addressed in sections such as “SZ and ASD: unique abnormalities”, but the discussion still contains vague phrasing such as “communication between self-related processing and other functions... may underlie the differences between SZ and ASD”. Can the authors be more concrete in what they are referring to? In short, relating these results to what has been found previously and what still needs to be done will assist in highlighting why these findings are interesting (which they are).

Response:

We appreciate the favorable evaluation and constructive suggestions. According to your comments, we have significantly revised our paper, especially the result and discussion parts.

In the “Results” section, we added the subsections to explain the common changes and unique changes separately to make the manuscript easier to read. We also added two subsections to validate that the identified group differences are reliable and can be taken as potential biomarkers in distinguishing schizophrenia and autism spectrum disorder. The two subsections are: “Brain changes show consistency using age-matched subjects” and “Brain changes can successfully distinguish SZ and ASD”.

In the “Discussions” section, we have revised the texts and added more references so as to make the discussion concrete and interesting. In the first paragraph of the “Discussions” section, we also summarize the rationale of our study: “SZ and ASD are recognized as distinct illnesses following a long-standing nosological development (Sasson, Pinkham et al. 2011, Hommer and Swedo 2015). Their similarity in clinical symptoms characterized by social and communication deficits and sensory abnormalities (Eack, Bahorik et al. 2013) may originate from the sharing of functional and structural alterations (Zheng, Zheng et al. 2018, Rabany, Brocke et al. 2019). Therefore, it is greatly needed to unravel how schizophrenia and autism are related and unique in brain abnormality. This is so far the

largest study that investigates the commonality and specificity between SZ and ASD in both brain functional connectivity and gray matter impairments by investigating them simultaneously and directly, showing neuroimaging evidence to help elucidate their neural substrates (see Fig. 8 for a summary).”

I have a couple of other specific points.

Introduction:

- I am missing the difference between hypothesis 1 compared to 2 and 3?

Response:

Thank you very much. Considering your comments, we have updated the descriptions. The texts are “In this paper, we propose to use a multi-modality data informed method to comprehensively study the similarity and differences between two disorders, aiming to provide statistically powerful evidence to address the following questions: (1) What are the common abnormalities between SZ and ASD in brain functional networks, functional connectivity, and gray matter volume and density? (2) What are the disorder-unique changes of the two disorders in these neuroimaging measures? (3) Measured by these biologically meaningful measures, to what extent ASD resembles SZ?”

- The sentences following “However, there are notable differences between populations afflicted with the two disorders.” Needs references to support the claims.

Response:

Thank you for your careful review and good suggestions. We have added references and revised the texts.

The updated descriptions are: “However, there are also notable differences between populations afflicted with the two disorders, especially the presence of different atypical behaviors in them (Trevisan, Foss-Feig et al. 2020). Patients with SZ often experience hallucinations and delusional thoughts that are uncommon in ASD. There is much greater likelihood of restricted and repetitive behaviors (Leekam, Prior et al. 2011), stereotyped language, and seizures (Besag 2018) in ASD than in SZ. Patients with SZ often show progressive loss of contact, but children with ASD lack contact from the start. Therefore, the unique mechanism of the two disorders also needs to be further explored.”

- Other references to include directly comparing autism and schizophrenia that may help with adding context to the rationale and discussion points:

Eack, S. M., Bahorik, A. L., McKnight, S. A. F., Hogarty, S. S., Greenwald, D. P., Newhill, C. E., ... Minshew, N. J. (2013). Commonalities in social and non-social cognitive impairments in adults with autism spectrum disorder and schizophrenia. *Schizophrenia Research*, 148(1–3), 24–28. <https://doi.org/http://dx.doi.org/10.1016/j.schres.2013.05.013>

Eack, S. M., Wojtalik, J. A., Keshavan, M. S., & Minshew, N. J. (2017). Social-cognitive brain function and connectivity during visual perspective-taking in autism and schizophrenia. *Schizophrenia Research*, 183, 102–109. <https://doi.org/10.1016/j.schres.2017.03.009>

Haigh, S. M., Eack, S. M., Keller, T., Minshew, N. J., & Behrmann, M. (2019). White matter structure in schizophrenia and autism: Abnormal diffusion across the brain in schizophrenia. *Neuropsychologia*, 135, 107233. <https://doi.org/https://doi.org/10.1016/j.neuropsychologia.2019.107233>

Haigh, S. M., Gupta, A., Barb, S. M., Glass, S. A. F., Minshew, N. J., Dinstein, I., ... Behrmann, M. (2016). Differential sensory fMRI signatures in autism and schizophrenia: Analysis of amplitude and trial-to-trial variability. *Schizophrenia Research*, 175(1–3), 12–19. <https://doi.org/http://dx.doi.org/10.1016/j.schres.2016.03.036>

Response:

Thank you very much for providing the references. They are all useful and helpful to our work. We have revised our manuscript by adding discussions and insights based on these references.

In the second paragraph of “Introduction” section, we added “Remarkably, Eack et al. (Eack, Bahorik et al. 2013) found a high degree of shared impairments between the two disorders in both social and non-social cognitive domains, especially the slow processing speed and inability to understand emotion. All the findings suggest the possibility of common underlying neurological mechanisms in SZ and ASD.”

In the third paragraph of “Introduction” section, we added “In addition to using fMRI and sMRI data, Haigh et al. used diffusion data to study the two disorders, revealing SZ-specific changes in its greater mean diffusivity than both ASD and HC (Haigh, Eack et al. 2019).”

In the “Discussions” section, the updated texts in the first paragraph are “SZ and ASD are recognized as distinct illnesses following a long-standing nosological development (Sasson, Pinkham et al. 2011, Hommer and Swedo 2015). Their similarity in clinical symptoms characterized by social and communication deficits and sensory abnormalities (Eack, Bahorik et al. 2013) may originate from the sharing of functional and structural alterations (Zheng, Zheng et al. 2018, Rabany, Brocke et al. 2019).”

In the “Discussions” section, we added “The differences in sensorimotor function have been disclosed by data analysis methods. For example, by comparing fMRI response to somatosensory stimuli, a study from Haigh et al. (Haigh, Gupta et al. 2016) confirms that differential sensory fMRI signatures were present between SZ and ASD, with SZ having smaller responses amplitude and ASD

having less trial-to-trial reliability. Except the discrepancy in sensorimotor system, our work observed their unique impairments in terms of the interaction between default mode and sensorimotor regions, highlighting the importance of motor systems and their interaction with high-level association systems in mental disorders.”

In the “Discussions” section, the revised texts include: “The disorder-specific abnormality identified in our study also involved the communication between the visual and cognitive control functions showing decreases in ASD but increases in SZ. In fact, the important relationship between visual processing and social perception and cognition in both SZ (Sergi, Rassovsky et al. 2006) and ASD (Hellendoorn, Langstraat et al. 2014) has been reported. From a data analysis angle, Eack et al. (Eack, Wojtalik et al. 2017) investigated differences between SZ and ASD using fMRI data under a design of a visual perspective-taking task, and revealed their unique fronto-temporal connectivity changes compared to healthy group. While these studies have indicated the disorder differences in visual and cognitive functions, our results clearly support that the interaction between visual and cognitive regions had divergent alterations in the two disorders. Measured by spatial functional networks, SZ-unique decreases related to the visual and sensory information processing in the middle occipital gyrus and inferior parietal lobule, and the ASD-unique decreases involved higher cognitive functions such as the superior frontal gyrus. Overall, our findings suggest that unique neural mechanisms may be more related to vision intertwined with cognition and sensorimotor.”

Methods:

- The correlation cutoffs for fMRI and sMRI differ. What was the reason? How were these thresholds obtained?

Response:

Thank you for your questions.

In our work, we used different thresholds for the quality control in selecting subjects for fMRI and sMRI respectively, as the two modalities have different properties and we wanted to keep the remaining subjects to have balanced sample sizes. The details about the quality control are described in supplementary “S1. Quality control on preprocessed fMRI and sMRI data”.

Since fMRI data were analyzed by an ICA-based method (i.e. NeuroMark), the consistency between individual brain mask and group brain mask is important. In the Group ICA of fMRI Toolbox (GIFT) (<https://trendscenter.org/software/gift/>) that is a widely-applied ICA-based fMRI analysis toolbox, a group brain mask is usually obtained based on the individual brain mask and then fMRI data within the group brain mask are used for ICA. As such, the similarity between the individual brain mask and group brain mask can be used to reflect the quality of individual fMRI data and select the subjects. In this paper, the computation methods of the individual and group brain masks are all the same to that used in GIFT. In particular, we used three correlations to reflect the

similarity measures between the two types of mask for the top, bottom, and whole parts in brain. Different thresholds were used for the three correlations as they corresponded to different parts in brain.

For the sMRI data, our subject selection method was also based on previous experiences. Researchers often used the group mean structural image to determine a group mask for multi-subject sMRI data analysis. In this paper, a threshold 0.2 was used to generate the group mask, consistent with previous work by others (Ashburner 2015, Zhu, Zhang et al. 2020). We employed the similarity between individual-subject and group mean structural image within the group mask to reflect the quality of each subject's sMRI data. Similar to fMRI data, we also computed three correlations for measuring their similarity in the top, bottom and whole parts. It is worth noting that the three thresholds associated the three similarity measures were set to different values between fMRI and sMRI due to their data diversity and their difference in computing correlations. Different thresholds were used in order to make the selected sample size comparable across fMRI, modulated sMRI and unmodulated sMRI. The resulting modulated and unmodulated structural images had slightly different subject numbers after the subject selection, because we used the same thresholds for them (i.e. sMRI). We believe our subject selection methods are useful for determining high-quality data and brain masks for analysis.

To address your concerns, we have added more details and explanations into the subsection "Data and preprocessing" of the "Methods" section. The texts are: "Since different criteria and thresholds were used for the quality control of fMRI and sMRI data, the selected subjects were slightly different among the three types of data (fMRI, modulated sMRI, and unmodulated sMRI). However, the sample sizes of the remaining subjects were comparable."

- Did motion parameters significantly differ between groups?

Response:

Thanks for the question. Table S1 in the supplementary materials shows the demographic information and head motion measures of fMRI data for each dataset. For each dataset, we also list the head motion differences between groups estimated by two-sample t-tests in the following table for your convenience. The motion translation measure of each subject was computed by averaging translation parameters across time points as well as x, y, and z axes. The motion rotation measure of each subject was computed by averaging rotation parameters across time points as well as pitch, yaw and roll. It is seen that there was no significant group difference in head motion for each dataset. If combining the subjects from all datasets together (totally 2980 subjects), there were some differences in motion measures, measured by the p-value from ANOVA (see Table S4). However, the absolute difference across the three groups in mean motion translation was smaller than 0.05 mm, and the absolute difference across the three groups in mean motion rotation was smaller than 0.07 degree.

Furthermore, we tried our best to remove out the motion effects from the neuroimaging measures. The fMRI data were preprocessed by regressing out six head motion parameters before ROI-based functional connectivity analysis. Regarding the functional networks estimated from ICA, motion-related artifacts were removed out by decomposing the data into different components including motion-related component (Zuo, Kelly et al. 2010, Du and Fan 2013). For functional network connectivity estimated from ICA, we also regressed out the motion parameters from the time series of functional networks before computing the functional network connectivity.

Table. The head motion measures of fMRI data for each dataset. The motion difference between groups was estimated by two-sample t-tests.

		SZ	HC	p-value
BSNIP	transitions: mean (std)	0.1451 (0.1485)	0.1269 (0.1030)	0.1397
	rotations: mean (std)	0.1159 (0.1193)	0.1312 (0.1205)	0.1968
COBRE	transitions: mean (std)	0.1984 (0.1202)	0.2184 (0.1464)	0.3624
	rotations: mean (std)	0.1769 (0.1208)	0.1891 (0.1177)	0.5258
FBIRN	transitions: mean (std)	0.1790 (0.1269)	0.1921 (0.1465)	0.4239
	rotations: mean (std)	0.1982 (0.1592)	0.2132 (0.1551)	0.4223
MPRC	transitions: mean (std)	0.1016 (0.1028)	0.0864 (0.0569)	0.0617
	rotations: mean (std)	0.0819 (0.0977)	0.0728 (0.0553)	0.2419
ABIDE I	transitions: mean (std)	0.2018 (0.1314)	0.1798 (0.1148)	0.0084
	rotations: mean (std)	0.2103 (0.1377)	0.1925 (0.1237)	0.0459
ABIDE II	transitions: mean (std)	0.1930 (0.1483)	0.1929 (0.1458)	0.9894
	rotations: mean (std)	0.2036 (0.1560)	0.1936 (0.1437)	0.3285

- While the different datasets were included as a factor in the analyses, the datasets themselves were comprised from data from multiple sites. Was this accounted for in the analyses too?

Response:

Thank you for the question. As you noted, each dataset (e.g. BSNIP) included the data collected at different sites. We did remove all site effects in our analyses. The detailed processing is described in the “Data and preprocessing” subsection of the “Methods” section.

- Similarly, while comparisons between autism and controls were (roughly) age matched, and similar

with schizophrenia and their controls, autism and schizophrenia individuals were widely different in age. Age was included in the analysis; however, comparing a majority pediatric population to an all-adult population seems a little strange regardless of the analysis conducted. The ABIDE datasets include a number of adults with autism. Could the authors match a subset of the autism group to schizophrenia to verify that their results still hold?

Response:

In our work, we carefully regressed out the effect of age. However, we also think your suggestion is a good idea to test the reliability of our finding. According to your comments, we have added experiments that only included age-matched subjects across the three groups to investigate the group differences. The added analyses and results are described in two subsections: “Exploring brain changes only using age-matched subjects” in the “Methods” section and “Brain changes show consistency using age matched subjects” in the “Results” section. We include the added subsections as below for your convenience.

Exploring brain changes only using age-matched subjects

Since the onset of ASD often occurs during early childhood and the onset of SZ is more observed in adults, the data used in above analyses had some differences in age between the two disorders. To address this, we also investigated the brain changes using HC, SZ and ASD groups with age-matched subjects (with no significant group difference in age, tested by ANOVA). More interestingly, we selected two sample sets with varied age ranges (see supplementary Table S7 for details). Regarding each sample set, we tested the HC vs. SZ, HC vs. ASD, and SZ vs. ASD differences in FNC measures using a similar analysis method in Fig. 2. Finally, we compared the brain changes obtained using these age-matched subject sets with the results obtained from all available subjects.

Table S7. Two sample sets with slightly different age ranges. Each sample set included age-matched HC, SZ, and ASD subjects.

		HC	SZ	ASD	
Sample set 1	Subject number	442	222	130	
	Age	Range	[21, 35]	[21, 35]	[21, 36]
		Mean	26.61	27.09	26.91
		Std	3.88	4.17	4.32
	p-value of age among the three groups, tested by ANOVA		0.0797		
Sample set 2	Subject number	461	248	104	
	Age	Range	[23, 42]	[23, 42]	[23, 42]
		Mean	30.50	31.09	29.60
		Std	5.67	5.74	5.49
	p-value of age among the three		0.0731		

Brain changes show consistency using age matched subject

As more data help to generate reliable findings, we used all available data in the above analyses. Moreover, we also investigated brain differences in FNC using two sample sets, each of which included age-matched three groups (see supplementary Table S7 for the age information). The HC vs. SZ, HC vs. ASD, and SZ vs. ASD differences (shown in Fig. 7) support that regardless of the sample sets, the group differences using age-matched subjects tended to show similar patterns with the results using all available subjects (shown in Fig. 4(A)), suggesting that our findings are relatively reliable.

Fig. 7 The HC vs. SZ, HC vs. ASD, and SZ vs. ASD differences in FNC using the sample set 1 and 2 (see Table S7) are shown in (A) and (B), respectively. The T-values obtained using two-sample t-tests on any pair of groups are displayed for comparing with the results using all subjects (Fig. 4(A)).

Results:

- Reducing the number of acronyms will help with the readability.

Response:

Thank you for your suggestion. We have revised some acronyms.

- Not sure what sentences like this mean: “Notably, within the common changes, ASD in general presented weaker changes than SZ (Fig. 3(B)), with 77% and 90% of voxels in the common decreases and increases, respectively (Table 1).”

Response:

Sorry for the confusion.

In our updated paper, we have revised the descriptions to clarify the statistical analyses and the results. The ASD-weaker changes means that the magnitude of ASD changes (relative to HC) was smaller than that of SZ. The following texts and one pipeline figure (Fig. 2) have been added to the “Investigating brain functional networks revealed by ICA” subsection for addressing your concern.

The statements are: “Next, based on the results from two-sample t-tests, we summarized the disorder-common and disorder-unique changes. In terms of the voxels passing ANOVA in each network, the voxels with T-values > 0 in both HC vs. SZ and HC vs. ASD reflected the disorder-common decreases, and the voxels with T-values < 0 in both HC vs. SZ and HC vs. ASD reflected the disorder-common increases. The disorder-unique changes included SZ-unique decrease (i.e. ASD-unique increase) and ASD-unique decrease (i.e. SZ-unique increase). The SZ-unique decrease involved voxels with both T-values > 0 in HC vs. SZ and T-values < 0 in HC vs. ASD. Similarly, the ASD-unique decrease corresponded to the voxles with both T-values < 0 in HC vs. SZ and T-values > 0 in HC vs. ASD. After that, we computed the voxel percentage for each of the four types of changes within all voxels showing group differences in ANOVA. Furthermore, among the voxles showing common decrease, we summarized the percentage of the voxels that had weaker decreases in ASD than SZ (i.e. T-values < 0 in SZ vs. ASD), and we called the type of change as ASD-weaker decrease within the decrease overlap. Similarly, we obtained the associated results for ASD-weaker increase within the increase overlap. The procedure is outlined in Fig. 2.”

Fig. 2 Statistical analysis outline for identifying and summarizing the shared and distinct brain abnormalities of SZ and ASD. The statistical analysis method is consistent for all neuroimaging measures, including functional networks, functional network connectivity, functional connectivity, and gray matter volume and density. Regarding each measure, ANOVA and two-tailed two-sample t-tests were first performed, resulting in group differences for HC vs. SZ, HC vs. ASD, and SZ vs. ASD. For the measures passing ANOVA, different types of changes were then summarized, including the common decreases (the voxels with T-values > 0 in both HC vs. SZ and HC vs. ASD), the common increases (the voxels with T-values < 0 in both HC vs. SZ and HC vs. ASD), the SZ-unique decrease (the voxels with both T-values < 0 in HC vs. ASD and T-values > 0 in HC vs. SZ), and the ASD-unique decrease (the voxels with both T-values < 0 in HC vs. SZ and T-values > 0 in HC vs. ASD). The percentage of each type of change was calculated as the number of neuroimaging measures relating to the change divided by the number of measures passing ANOVA. Among the neuroimaging measures with the common decrease (or increase), the percentage of ASD-weaker decrease (or increase) that showed T-values < 0 in SZ vs. ASD in the common decrease (or showed T-values > 0 in SZ vs. ASD in the common increase) were further summarized.

- A suggestion for a way to make the results easier to follow is to include subsections to individuate when discussing common findings across autism and schizophrenia and what findings are diagnosis specific. Similarly, separating the description of which networks showed increased compared to decreased activity might help.

Response:

Thank you so much for the good suggestion. We have revised the contents by adding different subsections to discuss the disorder-common changes and disorder-unique changes separately. Also, we have separated the descriptions with respect to the decreases and increases of brain changes.

- The analysis with symptoms measures seems to suggest that worse default-mode activity relates to worse symptoms rather than anything specific. Is this the case? Could the authors include the direction of the correlations so that the reader does not have to go to the supplementary materials section to be able to tell.

Response:

Thank you for the helpful comment. We have revised the descriptions in the “Results” section to clarify the correlations between the symptom scores and neuroimaging measures as well as the consistency between the correlation directions and group differences. The updated texts are as below.

“The neuroimaging measures were linked to the clinical scores. For each neuroimaging measure, we computed its correlation with the symptom scores for the SZ and ASD groups, separately. As shown in Fig. S5, four disorder-common hypo-FNCs were negatively correlated with the ADOS (or SRS) in ASD and PANSS positive symptom scores in SZ. Interestingly, they were all from the within-domain of SM, AU, and CC. Our results suggest that decreased connectivity strengths between regions within the SM, AU and CC domains may relate to worse clinical presentations in SZ and ASD.

Eight disorder-common hyper-FNCs, including those between SM and CB, SC and VI, SC and DM, and CC and DM, were positively correlated with the symptom scores in ASD and SZ. That means that increased strengths in those connections (such as the connectivity between CC and DM) could result in worse clinical presentations for both SZ and ASD.

Two disorder-unique FNCs showing decreased strengths in ASD but increased strengths in SZ were also found to be correlated with the SRS and PANSS positive scores. Notably, each was connectivity between the VI and CC domains, again indicating the unique property of visual impairment. Moreover, both correlation trends were consistent to the group difference results.”

Discussion:

- Limitations and future directions missing

Response:

Thank you for the suggestion. We have added the following texts to the “Discussions” section.

“Our study has some limitations that should be considered in future work. In this work, we used the age-matched subjects to validate the group differences in FNC and also employed a classification strategy to show the effectiveness of the identified differences in differentiating the two disorders. However, we did not explore other measures using these procedures due to limited space. Further validations on other brain functional and structural measures can be conducted in future work. Another limitation is about the effects of “noisy” factors. Minimizing the influences of age, gender, site, and motion is often a difficulty. Since there is no ground truth of the group differences, it is hard to judge whether these “noises” were clearly regressed out. In our paper, comprehensive processing was implemented to handle those covariates. Our results also validated the reliable differences using age-matched subjects. However, more advanced algorithms are still needed to deal with those effects. The third shortcoming lies in the comparison among different neuroimaging measures. In our work, the data-driven and hypothesis-derived functional connectivity measures were analyzed and compared with each other. However, it was not easy to compare the functional connectivity results due to their various parcellations and preprocessing procedures. In addition, we only summarized the association between functional and structural changes according to the affected brain regions. Fusion or more advanced methods (Park, Raznahan et al. 2018) might be better for cross-modal link in a direct manner. For another possible future direction, we think the identified brain similarity and uniqueness in our study could help developing new biotypes within the two disorders, as an interesting work (Stefanik, Erdman et al. 2018) has shown greater differences between biotypes than original DSM-driven categories among SZ, ASD and bipolar disorders.”

Thank you for your time and careful review. Hope you will be satisfied with the revisions.

Reviewer #3

The manuscript submitted by Du et al takes on a very interesting problem related to the common and unique variation explained across autism spectrum disorder (ASD) and schizophrenia (SZ) using functional and structural MRI data. The authors leverage the leverages the group information guided ICA that allows uses network templates as priors to estimate the subject-specific spatial functional network. The authors first extract network level impairment – and for each network the authors use a t-test to extract networks that show a positive activation, followed by an ANOVA to determine group level differences. They further examine group differenced in functional network connectivity using the ICA components. While I recognize that there may be some interesting work here, I have significant reservations about the methods and how they were applied. Further the reporting of the main results is very confusing and hard to follow.

1. From a very basic standpoint there is significant evidence that data quality will impact the measures that the authors choose to use here. There is really good evidence that these data will poorly impact all downstream measures. I am somewhat satisfied with how the authors treated the fMRI analysis – however, I would advise them to perhaps do what has been done in the Satterthwaite papers and do a sensitivity analysis examining what happens when some of the data is fed back into the analysis. Further, it would be good if the authors cited some of the best practices that they chose to follow. On the structural side, however, I would imagine that the authors should have done some manual quality control on their data. There is plenty of evidence that this is important, especially in the context of ASD and younger populations. A recent paper by Bedford et al (PMID: PMID: 31028290) clearly demonstrates the importance of detailed quality control and its impact on analytical outputs. Work by Pardoe et al (PMID: PMID: 27153982) further motivates this significant need. The ad-hoc heuristic used here, does not seem sufficient.

Response:

Thank you very much for your careful review and comments. We would like to address your concerns from the following four points.

(1) We also think that quality control is very important for further analysis. Regarding the fMRI and sMRI data, we did perform the quality control to select subjects with good data quality. We have improved the descriptions about the quality control in the supplementary subsection “S1. Quality control on preprocessed fMRI and sMRI data”. The texts are included here for your convenience.

“Regarding the fMRI data, we selected subjects with the following properties: 1) data with head motions less than 3° rotations and 3 mm translations along the whole scanning period; 2) data with more than 120 time points in fMRI acquisition; 3) data providing a successful normalization in the full brain. For the third point, since good normalization of fMRI data to standard brain template is necessary to group ICA, we proposed a method to evaluate the normalization quality of fMRI data by comparing the individual brain mask with the group brain mask. Our method was inspired by the

Group ICA of fMRI Toolbox (GIFT) (<https://trendscenter.org/software/gift/>) in which a group brain mask is usually obtained based on the individual brain mask and fMRI data within the group brain mask are used for ICA. As such, the similarity between the individual brain mask and group brain mask can be used to reflect the quality of individual fMRI data and then select the subjects. Our method was applied to each dataset's fMRI data, separately. First, using the 3D image in the first time point of fMRI data, the individual mask was calculated for each subject by setting the brain voxels showing greater values than 90% of the whole-brain mean to 1. Then, we generated a group mask by setting voxels included in more than 90% of the individual masks to 1. After that, the spatial correlations between the group mask and the individual mask was evaluated for each subject. The spatial correlations were calculated using the voxels within the top 10 slices of the mask, within the bottom 10 slices of the mask, and within the whole mask, respectively, resulting in three correlation values for each subject. If a subject had correlations larger than 0.75 for the top 10 slices, larger than 0.55 for the bottom 10 slices, and larger than 0.8 for the whole mask, we included this subject for further fMRI analysis. Finally, the group mask of each dataset was computed again based on the selected subjects' individual masks.

For sMRI, we chose the data with good quality by comparing the individual structural image with the group mean structural image. This processing was applied to each dataset's modulated or unmodulated sMRI data, separately. First, we calculated the group mean structural image across all subjects and then computed a group mask by preserving the voxels of group mean image which have values greater than a constant threshold of 0.2, consistent with previous work by others (Zhu, Zhang et al. 2020). Next, for each subject, we calculated spatial similarity between the group mean structural image and the individual structural image within the group mask. The spatial correlations were calculated using the voxels within the top 10 slices of the mask, within the bottom 10 slices of the mask, and within the whole brain mask, resulting in three correlation values for each subject. If a subject had correlations larger than 0.6 for the top 10 slices, larger than 0.6 for the bottom 10 slices, and larger than 0.8 for the whole mask, we included this subject for further sMRI analysis. Finally, the group mask of each dataset was computed again based on the selected subjects.”

(2) We confess that differences in age and gender among groups could influence the results, as you pointed out using the reference (Bedford, Park et al. 2020). In our work, we removed out the age and gender effects before the statistical analyses on neuroimaging measures. We have revised the descriptions to clarify the processing. Please see the last paragraph in the subsection “Data and preprocessing” for the details.

After considering your comments, we have preformed additional experiments that only included the age-matched subjects for the statistical analyses. The new results are consistent to our previous results. The contents are included in the subsection of “Exploring brain changes using age-matched subjects” of the method part and the subsection “Brain changes show consistency using age-matched

subjects” of the result part. We also include the descriptions as below for your convenience.

Exploring brain changes using age-matched subjects

Since the onset of ASD often occurs during early childhood and the onset of SZ is more observed in adults, the data used in above analyses had some differences in age between the two disorders. To address this, we also investigated the brain changes using HC, SZ and ASD groups with age-matched subjects (with no significant group difference in age, tested by ANOVA). More interestingly, we selected two sample sets with varied age ranges (see supplementary Table S7 for details). Regarding each sample set, we tested the HC vs. SZ, HC vs. ASD, and SZ vs. ASD differences in FNC measures using a similar analysis method in Fig. 2. Finally, we compared the brain changes obtained using these age-matched subject sets with the results obtained from all available subjects.

Table S7. Two sample sets with slightly different age ranges. Each sample set included age-matched HC, SZ, and ASD subjects.

		HC	SZ	ASD	
Sample set 1	Subject number	442	222	130	
	Age	Range	[21, 35]	[21, 35]	[21, 36]
		Mean	26.61	27.09	26.91
		Std	3.88	4.17	4.32
	p-value of age among the three groups, tested by ANOVA		0.0797		
Sample set 2	Subject number	461	248	104	
	Age	Range	[23, 42]	[23, 42]	[23, 42]
		Mean	30.50	31.09	29.60
		Std	5.67	5.74	5.49
	p-value of age among the three groups, tested by ANOVA		0.0731		

Brain changes show consistency using age-matched subjects

As more data help to generate reliable findings, we used all available data in the above analyses. Moreover, we also investigated brain differences in FNC using two sample sets, each of which included age-matched three groups (see supplementary Table S7 for the age information). The HC vs. SZ, HC vs. ASD, and SZ vs. ASD differences (shown in Fig. 7) support that regardless of the sample sets, the group differences using age-matched subjects tended to show similar patterns with the results using all available subjects (shown in Fig. 4(A)), suggesting that our findings are relatively reliable.

Fig. 7 The HC vs. SZ, HC vs. ASD, and SZ vs. ASD differences in FNC using the sample set 1 and 2 (see Table S7) are shown in (A) and (B), respectively. The T-values obtained using two-sample t-tests on any pair of groups are displayed for comparing with the results using all subjects (Fig. 4(A)).

(3) As you mentioned using the reference (Pardoe, Kucharsky Hiess et al. 2016), motion is also an effect that could bias the results. Table S1 in the supplementary materials shows the demographic information and head motion measures of fMRI data for each dataset. For each dataset, we also list the head motion differences between groups estimated by two-sample t-tests in a table (see below) for your convenience. The motion translation measure of each subject was computed by averaging translation parameters across time points as well as x, y, and z axes. The motion rotation measure of each subject was computed by averaging rotation parameters across time points as well as pitch, yaw and roll. It is seen that there was no significant group difference in head motion for each dataset, measured by two-sample t-tests. If combining the subjects from all datasets together (totally 2980 subjects), there were some differences in motion measures, measured by the p-value from ANOVA (see Table S4), due to the big sample size. However, the absolute difference across the three groups in mean motion translation was smaller than 0.05 mm, and the absolute difference across the three groups in mean motion rotation was smaller than 0.07 degree.

Moreover, we tried our best to remove out the motion effects from the neuroimaging measures. The fMRI data were preprocessed by regressing out six head motion parameters, before ROI-based functional connectivity analysis. Regarding the functional networks estimated from ICA, motion-related artifacts were removed out by decomposing the data into different components including motion-related component (Zuo, Kelly et al. 2010, Du and Fan 2013). For functional network connectivity estimated from ICA, we also regressed out the motion parameters from the

time series of functional networks before computing the functional network connectivity.

Table. The head motion measures of fMRI data for each dataset. The motion difference between groups was estimated by two-sample t-tests.

BSNIP		SZ	HC	p-value
	transitions: mean (std)	0.1451 (0.1485)	0.1269 (0.1030)	0.1397
	rotations: mean (std)	0.1159 (0.1193)	0.1312 (0.1205)	0.1968
COBRE		SZ	HC	p-value
	transitions: mean (std)	0.1984 (0.1202)	0.2184 (0.1464)	0.3624
	rotations: mean (std)	0.1769 (0.1208)	0.1891 (0.1177)	0.5258
FBIRN		SZ	HC	p-value
	transitions: mean (std)	0.1790 (0.1269)	0.1921 (0.1465)	0.4239
	rotations: mean (std)	0.1982 (0.1592)	0.2132 (0.1551)	0.4223
MPRC		SZ	HC	p-value
	transitions: mean (std)	0.1016 (0.1028)	0.0864 (0.0569)	0.0617
	rotations: mean (std)	0.0819 (0.0977)	0.0728 (0.0553)	0.2419
ABIDEI		ASD	HC	p-value
	transitions: mean (std)	0.2018 (0.1314)	0.1798 (0.1148)	0.0084
	rotations: mean (std)	0.2103 (0.1377)	0.1925 (0.1237)	0.0459
ABIDEII		ASD	HC	p-value
	transitions: mean (std)	0.1930 (0.1483)	0.1929 (0.1458)	0.9894
	rotations: mean (std)	0.2036 (0.1560)	0.1936 (0.1437)	0.3285

(4) Considering your comments, we have also added 12 classification experiments to assess the sensitivity of our estimated network measures to different datasets. Our results support that these neuroimaging measures can classify SZ and ASD well.

We added one subsection “Classifying SZ and ASD using identified brain changes” to the “Methods” section. The added texts are included as below for your reference.

“It is important to examine whether the identified brain changes can be used as biomarkers to distinguish the two disorders. To avoid bias, we conducted two-group (SZ and ASD) classification by taking different datasets for training and the remaining datasets for testing based on FNC measures. Since there were four datasets relating to the SZ group and two datasets relating to the ASD group, totally we performed 12 classification experiments using different data assignments. Feature extraction and model training were implemented only using the training data, and then the testing data were classified and compared with their true class labels. In our work, we extracted the disorder-unique and ASD-weaker common FNC changes for every two groups (SZ and ASD) using the above mentioned statistical analyses, and then used the union set of those changes as the features.

Linear support vector machine with a Bayesian optimization technology (A Snoek, A Larochelle et al. 2012) to optimize the parameter was applied for the model building. Finally, the classification results were evaluated using accuracy, sensitivity, and specificity.”

In the “Results” section, we added the following sentences in a new subsection “Brain changes can successfully distinguish SZ and ASD”:

“We were also interested in whether the identified brain changes can be used as promising biomarkers to distinguish the two disorders. Our results support that FNC measures performed well in classifying SZ and ASD patients. Table 2 shows the classification results that were obtained by taking different datasets as training data/testing data (totally 12 classification runs). The mean accuracy, sensitivity and specificity across all 12 classification runs were 75%, 83%, and 63%, respectively. The best results reached up to 80.0% accuracy, 90.0% sensitivity, and 68.0% specificity, even only using features from statistical analyses. The most frequently used features in all classification runs included the unique changes of connectivity between SC (e.g. putamen, caudate, and thalamus) and SM (e.g. superior parietal lobule, paracentral lobule) domains, and also included the ASD-weaker common changes of connectivity between SC and VI/CB domains and between SM and VI domains. In our work, relatively lower specificity (compared to sensitivity) meant that SZ patients were more likely misdiagnosed as ASD using these connectivity measures.”

Table 2. Classification result evaluation of distinguishing SZ and ASD patients using different datasets as the training and testing data.

Training data		Testing data		Accuracy	Sensitivity	Specificity
SZ datasets (N)	ASD dataset (N)	SZ datasets (N)	ASD dataset (N)			
BSNIP & FBIRN (319)	ABIDEI (398)	COBRE & MPRC (218)	ABIDEII (380)	75.8%	84.2%	61.0%
BSNIP & COBRE (250)	ABIDEI (398)	FBIRN & MPRC (287)	ABIDEII (380)	73.9%	85.3%	58.9%
BSNIP & MPRC (332)	ABIDEI (398)	FBIRN & COBRE (205)	ABIDEII (380)	78.1%	80.5%	73.7%
FBIRN & COBRE (205)	ABIDEI (398)	BSNIP & MPRC (332)	ABIDEII (380)	67.4%	88.7%	43.1%
FBIRN & MPRC (287)	ABIDEI (398)	BSNIP & COBRE (250)	ABIDEII (380)	78.9%	84.7%	70.0%
COBRE & MPRC (218)	ABIDEI (398)	BSNIP & FBIRN (319)	ABIDEII (380)	80.0%	90.0%	68.0%
BSNIP & FBIRN (319)	ABIDEII (380)	COBRE & MPRC (218)	ABIDEI (398)	77.9%	80.2%	73.9%
BSNIP &	ABIDEII	FBIRN &	ABIDEI (398)	77.2%	85.4%	65.9%

COBRE (250)	(380)	MPRC (287)				
BSNIP & MPRC (332)	ABIDEII (380)	FBIRN & COBRE (205)	ABIDEI (398)	73.1%	76.4%	66.8%
FBIRN & COBRE (205)	ABIDEII (380)	BSNIP & MPRC (332)	ABIDEI (398)	70.4%	88.2%	49.1%
FBIRN & MPRC (287)	ABIDEII (380)	BSNIP & COBRE (250)	ABIDEI (398)	73.8%	78.1%	66.8%
COBRE & MPRC (218)	ABIDEII (380)	BSNIP & FBIRN (319)	ABIDEI (398)	71.7%	77.9%	63.9%

Footnote: N represents the subject number.

In the “Discussions” section, we also further summarized the point as below:

“We also found that the disorder-unique and ASD-weaker common abnormality of SZ and ASD performed well in distinguishing the two disorders. Satisfactory performances were reliably achieved even under the situation of different datasets as the training data/testing data with varied sample sizes. The connectivity between the sub-cortical domain and other domains such as sensorimotor, visional and cerebellar areas as well as the connectivity between the sensorimotor and visional domains played an important role in distinguishing the two disorders, which indicates potential biomarkers between the two disorders. To the best of our knowledge, this is the first study applying large-sample multi-site fMRI data to perform the direct classification between SZ and ASD. Our work achieved relatively higher classification accuracy than a previous study (Mastrovito, Hanson et al. 2018) that also directly classified the two disorders using functional connectivity features. They trained a classifier on 72 SZs and 37 ASDs and resulted in 75% classification accuracy on independent 5 SZ and 27 ASD patients.”

2. Other groups have provided some overlap of neural phenotypes across disorders. The work by Park et al. (PMID: 2968887), Stefanik (PMID: PMID: 29105664), and Yoshihara (PMID: 32300809) all uses a network-related methods and are not discussed. The authors may want to look into others.

Response:

Thanks for these very useful references. We have added reviews and discussions about all the references that you pointed to our paper.

In the third paragraph of “Introduction” section, we added the descriptions: “Park et al. found significant cortical thickness changes in both ASD and SZ in brain regions relating to the frontoparietal and limbic networks, however SZ was found to show decreased cortical thickness, in

contrast ASD presented increases in cortical thickness (Park, Raznahan et al. 2018).”

In the “Limitation and future direction” section, we added “The third shortcoming lies in the comparison among different neuroimaging measures. In our work, the data-driven and hypothesis-derived functional connectivity measures were analyzed and compared with each other. However, it was not easy to compare the functional connectivity results due to their various parcellations and preprocessing procedures. In addition, we only summarized the association between functional and structural changes according to the affected brain regions. Fusion or more advanced methods (Park, Raznahan et al. 2018) may be better for cross-modal link in a direct manner. For another possible future direction, we think the identified brain similarity and uniqueness in our study could help developing new biotypes within the two disorders, as an interesting work (Stefanik, Erdman et al. 2018) has shown greater differences between biotypes than original DSM-driven categories among SZ, ASD and bipolar disorders.”

In the first paragraph of “Discussions” section, we added “While some studies have suggested that the two disorders are functionally related (Chen, Uddin et al. 2017, Yoshihara, Lisi et al. 2020), our work provides quantitative evaluation supporting they are largely overlapped in brain functional abnormality.”

In the seventh paragraph of “Discussions” section, we added “A previous work by Park (Park, Raznahan et al. 2018) used cortical anatomy measures to investigate SZ, ASD and attention deficit hyperactivity disorder (ADHD), and found that different subcomponents of the extended visual network are affected in SZ patients compared with those with ASD and ADHD, supporting our finding in terms of their different visual function deficits. While these studies suggested their specificity in vision, our work further provides evidence that the way how the visual processing interacts with social understanding may be different.”

In the subsection of “Validation of the finding” in the “Discussions” section, we added “A recent study from Yoshihara (Yoshihara, Lisi et al. 2020) employed functional connectivity features to explore the complex relationship between SZ and ASD as well. In their work, dual classifiers were applied to discriminate ASD (or SZ) from HC so as to investigate the two disorders using a dimensional method, demonstrating the overlapping but asymmetrical relationship between ASD and SZ. Interestingly, they found that SZ subjects showed increased classification certainty for the ASD dimension while the ASD subjects did not for the SZ dimension. The findings are consistent to our classification results to some degree, as SZ was more likely to be grouped into ASD than the opposing situation in our classification experiments.”

3. The segmentation method used for the VBM-style analysis is not referenced and is poorly described.

Response:

Sorry for the unclear descriptions. We have added more details to clarify the VBM analysis. The texts are: “For sMRI data, the T1-weighted images were first segmented into gray matter, white matter, and cerebrospinal fluid by using the standard unified segmentation model (Ashburner and Friston 2005). The Diffeomorphic Anatomical Registration Through Exponentiated Lie Algebra (DARTEL) algorithm was employed to create a group template for spatial normalization of the segmented images of each subject. Then, the flow fields generated by DARTEL were used to estimate individual-subject images. After that, individual-subject gray matter images were spatially normalized to the MNI space, modulated or unmodulated, resliced (1.0-mm isotropic voxels), and smoothed (6-mm full-width at half maximum Gaussian kernel). Finally, the obtained gray matter volume (modulated data) and density (unmodulated data) can be used for voxel-based morphometry (VBM).”

4. Dealing with site as nuisance variable is problematic. This may simply provide a “statistical” double hit where samples are different along sex and age – which is more than likely to be the case in SZ and ASD given the differences in age of onset. It would be more appropriate to use a linear mixed effects model with site as a random effect, some sort of meta-analytic technique, or COMBAT.

Response:

Thanks for your comments.

(1) Before identifying group differences on brain network measures, we carefully regressed out the influences of age, gender and site effects from the estimated network measures for each subject. The regression procedure included three steps. In the first step, we regressed out the age, gender, site information, the interaction between age and site, and the interaction between gender and site for all subjects in each dataset (e.g. FBIRN). In the second step, we estimated the data effects using HCs’ measures for the six datasets that were compared. The third step was further regressing out the data effect from each subject’s measures that were already removed the age, gender and site effects. The processing is consistent to our previous work (Du, Fu et al. 2020), and is also similar to some other studies using regression (Nakano, Takamura et al. 2020).

(2) Considering your comments, we have performed additional experiments that only included the age-matched subjects for the statistical analyses. The contents are included in the subsection of “Exploring brain changes using age-matched subjects” of the method part and the subsection “Brain changes show consistency using age-matched subjects” of the result part. We selected two sample sets with varied age ranges (see supplementary Table S7 for details). Regarding each sample set, we tested the HC vs. SZ, HC vs. ASD, and SZ vs. ASD differences in FNC measures using a similar analysis method in Fig. 2. The results are shown in Fig. 7 (see below). By comparing the Fig. 7 and Fig. 4(A), we think the identified group differences are relatively reliable.

Fig. 7 The HC vs. SZ, HC vs. ASD, and SZ vs. ASD differences in FNC using the sample set 1 and 2 (see Table S7) are shown in (A) and (B), respectively. The T-values obtained using two-sample t-tests on any pair of groups are displayed for comparing with the results using all subjects (Fig. 4(A)).

Fig. 4 (A) Upper subfigures: the T-value maps showing the group differences in FNCs, obtained by two-sample t-tests for HC vs. SZ, HC vs. ASD, and SZ vs. ASD. Taking HC vs. SZ for example, positive T-values represented that HCs showed higher connectivity strengths than SZs. Lower subfigures: The T-value maps of FNCs after Bonferroni (BFN) corrections.

(3) In addition, we have performed meta-analysis, and the results are included in the supplementary “S3. Meta-analysis of FNC using data from separate datasets”. Fig. S3 demonstrates the combined group differences for HC vs. SZ, HC vs. ASD, and SZ vs. ASD using meta-analysis. It can be seen that group differences in Fig. S3 showed a similar pattern with that in Fig. 4, supporting that the identified overlap and uniqueness of brain abnormality were relatively reliable.

Fig. S3 The combined group differences (HC vs. SZ, HC vs. ASD, and SZ vs. ASD) that were obtained from meta-analyses on separate datasets. The mean T-values across different comparisons are shown for the FNCs that past Bonferroni (BFN) correction in terms of the combined p-values.

5. The methods going from individual ICA components to group-level differences extremely hard to follow. I would suggest, at least, that the authors consider adding figures to the methods to better describe what is going on in this context.

Response:

Sorry for the confusion.

In our updated paper, we have revised the descriptions to clarify the statistical analyses and the results. The statistical analysis steps are consistent across all neuroimaging measures including functional networks from ICA, functional network connectivity from ICA, functional connectivity from ROI-based method, and gray matter volume and density. The following sentences and one pipeline figure (Fig. 2) have been added to the “Investigating brain functional networks revealed by ICA” subsection for addressing your concern.

The statements are: “Next, based on the results from two-sample t-tests, we summarized the disorder-common and disorder-unique changes. In terms of the voxels passing ANOVA in each network, the voxels with T-values > 0 in both HC vs. SZ and HC vs. ASD reflected the disorder-common decreases, and the voxels with T-values < 0 in both HC vs. SZ and HC vs. ASD reflected the disorder-common increases. The disorder-unique changes included SZ-unique decrease (i.e. ASD-unique increase) and ASD-unique decrease (i.e. SZ-unique increase). The SZ-unique decrease involved voxels with both T-values > 0 in HC vs. SZ and T-values < 0 in HC vs. ASD. Similarly, the ASD-unique decrease corresponded to the voxles with both T-values < 0 in HC vs. SZ

and T-values > 0 in HC vs. ASD. After that, we computed the voxel percentage for each of the four types of changes within all voxels showing group differences in ANOVA. Furthermore, among the voxles showing common decrease, we summarized the percentage of the voxels that had weaker decreases in ASD than SZ (i.e. T-values < 0 in SZ vs. ASD), and we called the type of change as ASD-weaker decrease within the decrease overlap. Similarly, we obtained the associated results for ASD-weaker increase within the increase overlap. The procedure is outlined in Fig. 2.”

Fig. 2 Statistical analysis outline for identifying and summarizing the shared and distinct brain abnormalities of SZ and ASD. The statistical analysis method is consistent for all neuroimaging measures, including functional networks, functional network connectivity, functional connectivity, and gray matter volume and density. Regarding each measure, ANOVA and two-tailed two-sample t-tests were first performed, resulting in group differences for HC vs. SZ, HC vs. ASD, and SZ vs. ASD. For the measures passing ANOVA, different types of changes were then summarized, including the common decreases (the voxels with T-values > 0 in both HC vs. SZ and HC vs. ASD), the common increases (the voxels with T-values < 0 in both HC vs. SZ and HC vs. ASD), the SZ-unique decrease

(the voxels with both T-values < 0 in HC vs. ASD and T-values > 0 in HC vs. SZ), and the ASD-unique decrease (the voxles with both T-values < 0 in HC vs. SZ and T-values > 0 in HC vs. ASD). The percentage of each type of change was calculated as the number of neuroimaging measures relating to the change divided by the number of measures passing ANOVA. Among the neuroimaging measures with the common decrease (or increase), the percentage of ASD-weaker decrease (or increase) that showed T-values < 0 in SZ vs. ASD in the common decrease (or showed T-values > 0 in SZ vs. ASD in the common increase) were further summarized.

6. The figures are really hard to parse. The circle plots are tiny and are difficult to understand without labelling and nudging towards the parts of the plots that you expect the readers to take away information from. Fig 5 does not convince me that these results survive across different parcellation techniques. Fig 6 needs to have the colour bars clearly labelled or to have one common colour bar across the figure. Given what I understand of figure 6 - it seems unlikely that there is significant morphological overall between disorders. I think this needs to be better described. These figs are the “meat” of the paper and without them being clearer, I find it hard to believe the take-away message.

Response:

Thank you. Considering your comments, we have revised all figures to make them clear.

Fig. 5 shows the group difference results of functional connectivity estimated using the AAL template. From Fig. 5 (please see below), it is observed that ASD showed a similar changing trend with SZ (as shown in the left and middle columns). Furthermore, the changes of ASD were weaker than that of SZ (as indicated in the right column of Fig. 5). The finding is consistent to the results using ICA based functional connectivity (shown in Fig. 4). Comparing Fig. 5 and Fig. 4(A), we found that in general the group differences obtained from different functional connectivity analysis methods showed similarity, especially the interaction between the sub-cortical domain and other domains (e.g. cerebellum, sensorimotor, and visional regions), between the cerebellum and other domains (e.g. sensorimotor, visional, cognitive control, and default mode regions), and the interactions within the visional domain and within default mode domain.

As you noted, the overlapping change degree in the brain structural measures was smaller, compared to the brain functional connectivity measures. Table 1 summarized the quantitative results that were computed based on the original T-values of the two-sample t-tests (please see Fig. 2 for the statistical analysis outline). Fig. 6 shows the group differences in the gray matter volume and density measures after multiple comparison correction. We have revised the texts to clarify more. The descriptions are: “In addition to brain functional measures, we also assessed brain structural changes of the two disorders. The overlapping change degree in the brain structural measures was smaller, compared to the brain functional connectivity measures. Measured by the original T-values of two-sample t-tests, both SZ and ASD showed overall decreased gray matter volume (40.2% overlap) and density (90% overlap) compared to HC (Table 1). As displayed in Fig. 6 that shows the group differences after multiple comparison corrections, the regions with the commonly decreased gray matter primarily involved anterior cingulate cortex, insula, parahippocampal gyrus, and

hippocampus. In contrast to the shared decrease, there was a low degree in the disorder-common increases (2% in volume and 0.07% in density) relating to the superior frontal gyrus and cerebellum. Notably, ASD also had weaker impairments than SZ in those common changes.”

We confess that it is somewhat difficult to compare functional connectivity under different scales, estimated using different methods. However, we hope to provide insights by linking the results from different methods, so we assigned the brain atlas-related ROIs to different functional domains, consistent with the assignment of ICA components. Our results support that the findings are relatively reliable. But, considering your comments, we have added some discussions to point out this limitation in our updated paper. The texts are: “The third shortcoming lies in the comparison among different neuroimaging measures. In our work, the data-driven and hypothesis-derived functional connectivity measures were analyzed and compared with each other. However, it was not easy to compare the functional connectivity results due to their various parcellations and preprocessing procedures. In addition, we only summarized the association between functional and structural changes according to the affected brain regions. Fusion or more advanced methods (Park, Raznahan et al. 2018) may be better for cross-modal link in a direct manner.”

We appreciate your careful review and constructive suggestions. Hope you will be satisfied with the revisions.

Fig. 5 Results from functional connectivity (FC) analysis using ROIs of Automated Anatomical Labeling (AAL) template. Upper figures: the original T-value maps representing group differences in FCs revealed by two-sample t-tests for HC vs. SZ, HC vs. ASD, and SZ vs. ASD. Lower figures: the T-value maps of FCs after Bonferroni (BFN) corrections.

References:

- A Snoek, J., H. A Larochelle and R. P. A Adams (2012). "Practical Bayesian Optimization of Machine Learning Algorithms." NIPS: 2960-2968.
- Allen, E. A., E. B. Erhardt, E. Damaraju, W. Gruner, J. M. Segall, R. F. Silva, M. Havlicek, S. Rachakonda, J. Fries, R. Kalyanam, A. M. Michael, A. Caprihan, J. A. Turner, T. Eichele, S. Adelsheim, A. D. Bryan, J. Bustillo, V. P. Clark, S. W. Feldstein Ewing, F. Filbey, C. C. Ford, K. Hutchison, R. E. Jung, K. A. Kiehl, P. Koditwakk, Y. M. Komesu, A. R. Mayer, G. D. Pearlson, J. P. Phillips, J. R. Sadek, M. Stevens, U. Teuscher, R. J. Thoma and V. D. Calhoun (2011). "A baseline for the multivariate comparison of resting-state networks." Front Syst Neurosci **5**: 2.
- Ashburner, J. (2015). "VBM Tutorial."
- Ashburner, J. and K. J. Friston (2005). "Unified segmentation." Neuroimage **26**(3): 839-851.
- Baggio, H. C., B. Segura, R. Sala-Llonch, M. J. Marti, F. Valldeoriola, Y. Compta, E. Tolosa and C. Junque (2015). "Cognitive impairment and resting-state network connectivity in Parkinson's disease." Hum Brain Mapp **36**(1): 199-212.
- Bedford, S. A., M. T. M. Park, G. A. Devenyi, S. Tullo, J. Germann, R. Patel, E. Anagnostou, S. Baron-Cohen, E. T. Bullmore, L. R. Chura, M. C. Craig, C. Ecker, D. L. Floris, R. J. Holt, R. Lenroot, J. P. Lerch, M. V. Lombardo, D. G. M. Murphy, A. Raznahan, A. N. V. Ruigrok, E. Smith, M. D. Spencer, J. Suckling, M. J. Taylor, A. Thurm, M. C. Lai and M. M. Chakravarty (2020). "Large-scale analyses of the relationship between sex, age and intelligence quotient heterogeneity and cortical morphometry in autism spectrum disorder." Molecular Psychiatry **25**(3): 614-628.
- Besag, F. M. C. (2018). "Epilepsy in patients with autism: links, risks and treatment challenges." Neuropsychiatric Disease and Treatment **14**: 1-10.
- Chen, H., L. Q. Uddin, X. J. Duan, J. J. Zheng, Z. L. Long, Y. X. Zhang, X. N. Guo, Y. Zhang, J. P. Zhao and H. F. Chen (2017). "Shared atypical default mode and salience network functional connectivity between autism and schizophrenia." Autism Research **10**(11): 1776-1786.
- Chen, Z. and V. Calhoun (2018). "Effect of Spatial Smoothing on Task fMRI ICA and Functional Connectivity." Front Neurosci **12**: 15.
- Du, Y., Z. Fu, J. Sui, S. Gao, Y. Xing, D. Lin, M. Salman, A. Abrol, M. Rahaman, A. J. Chen, L. Hong, E. P. Kochunov, E. Osuch, A and V. Calhoun, D. (2020). "NeuroMark: An automated and adaptive ICA based pipeline to identify reproducible fMRI markers of brain disorders." NeuroImage: Clinical **28**: 102375.
- Du, Y. H., E. A. Allen, H. He, J. Sui, L. Wu and V. D. Calhoun (2016). "Artifact removal in the context of group ICA: A comparison of single-subject and group approaches." Hum Brain Mapp **37**(3): 1005-1025.
- Du, Y. H. and Y. Fan (2013). "Group information guided ICA for fMRI data analysis." Neuroimage **69**: 157-197.
- Du, Y. H., D. D. Lin, Q. B. Yu, J. Sui, J. Y. Chen, S. Rachakonda, T. Adali and V. D. Calhoun (2017). "Comparison of IVA and GIG-ICA in Brain Functional Network Estimation Using fMRI Data." Frontiers in Neuroscience **11**: 267.
- Du, Y. H., G. D. Pearlson, J. Y. Liu, J. Sui, Q. B. Yu, H. He, E. Castro and V. D. Calhoun (2015). "A group ICA based framework for evaluating resting fMRI markers when disease categories are unclear: application to schizophrenia, bipolar, and schizoaffective disorders." Neuroimage **122**: 272-280.
- Eack, S. M., A. L. Bahorik, S. A. McKnight, S. S. Hogarty, D. P. Greenwald, C. E. Newhill, M. L. Phillips, M. S. Keshavan and N. J. Minshew (2013). "Commonalities in social and non-social cognitive impairments in adults with autism spectrum disorder and schizophrenia." Schizophr Res **148**(1-3): 24-28.
- Eack, S. M., J. A. Wojtalik, M. S. Keshavan and N. J. Minshew (2017). "Social-cognitive brain function and connectivity during visual perspective-taking in autism and schizophrenia." Schizophr Res **183**: 102-109.
- Fan, L., H. Li, J. Zhuo, Y. Zhang, J. Wang, L. Chen, Z. Yang, C. Chu, S. Xie, A. R. Laird, P. T. Fox, S. B. Eickhoff, C. Yu and T. Jiang (2016). "The Human Brainnetome Atlas: A New Brain Atlas Based on Connectional Architecture." Cereb Cortex **26**(8): 3508-3526.
- Haigh, S. M., S. M. Eack, T. Keller, N. J. Minshew and M. Behrmann (2019). "White matter structure in schizophrenia

and autism: Abnormal diffusion across the brain in schizophrenia." *Neuropsychologia* **135**: 107233.

Haigh, S. M., A. Gupta, S. M. Barb, S. A. F. Glass, N. J. Minshew, I. Dinstein, D. J. Heeger, S. M. Eack and M. Behrmann (2016). "Differential sensory fMRI signatures in autism and schizophrenia: Analysis of amplitude and trial-to-trial variability." *Schizophr Res* **175**(1-3): 12-19.

Hellendoorn, A., I. Langstraat, L. Wijnroks, J. K. Buitelaar, E. van Daalen and P. P. Leseman (2014). "The relationship between atypical visual processing and social skills in young children with autism." *Res Dev Disabil* **35**(2): 423-428.

Hommer, R. E. and S. E. Swedo (2015). "Schizophrenia and autism-related disorders." *Schizophr Bull* **41**(2): 313-314.

Jiang, T. Z. (2013). "Brainnetome: A new -ome to understand the brain and its disorders." *Neuroimage* **80**: 263-272.

Leekam, S. R., M. R. Prior and M. Uljarevic (2011). "Restricted and Repetitive Behaviors in Autism Spectrum Disorders: A Review of Research in the Last Decade." *Psychological Bulletin* **137**(4): 562-593.

Lydon-Staley, D. M., R. Ciric, T. D. Satterthwaite and D. S. Bassett (2019). "Evaluation of confound regression strategies for the mitigation of micromovement artifact in studies of dynamic resting-state functional connectivity and multilayer network modularity." *Netw Neurosci* **3**(2): 427-454.

Mastrovito, D., C. Hanson and S. J. Hanson (2018). "Differences in atypical resting-state effective connectivity distinguish autism from schizophrenia." *Neuroimage Clin* **18**: 367-376.

Nakano, T., M. Takamura, N. Ichikawa, G. Okada, Y. Okamoto, M. Yamada, T. Suhara, S. Yamawaki and J. Yoshimoto (2020). "Enhancing Multi-Center Generalization of Machine Learning-Based Depression Diagnosis From Resting-State fMRI." *Front Psychiatry* **11**: 400.

Ongur, D., M. Lundy, I. Greenhouse, A. K. Shinn, V. Menon, B. M. Cohen and P. F. Renshaw (2010). "Default mode network abnormalities in bipolar disorder and schizophrenia." *Psychiatry Res* **183**(1): 59-68.

Osuch, E., S. Gao, M. Wammes, J. Theberge, P. Willimason, R. J. Neufeld, Y. Du, J. Sui and V. Calhoun (2018). "Complexity in mood disorder diagnosis: fMRI connectivity networks predicted medication-class of response in complex patients." *Acta Psychiatr Scand* **138**(5): 472-482.

Pardoe, H. R., R. Kucharsky Hiess and R. Kuzniecky (2016). "Motion and morphometry in clinical and nonclinical populations." *Neuroimage* **135**: 177-185.

Park, M. T. M., A. Raznahan, P. Shaw, N. Gogtay, J. P. Lerch and M. M. Chakravarty (2018). "Neuroanatomical phenotypes in mental illness: identifying convergent and divergent cortical phenotypes across autism, ADHD and schizophrenia." *J Psychiatry Neurosci* **43**(3): 201-212.

Park, M. T. M., A. Raznahan, P. Shaw, N. Gogtay, J. P. Lerch and M. M. Chakravarty (2018). "Neuroanatomical phenotypes in mental illness: identifying convergent and divergent cortical phenotypes across autism, ADHD and schizophrenia." *J Psychiatry Neurosci* **43**(2): 170094.

Rabany, L., S. Brocke, V. D. Calhoun, B. Pittman, S. Corbera, B. E. Wexler, M. D. Bell, K. Pelphrey, G. D. Pearlson and M. Assaf (2019). "Dynamic functional connectivity in schizophrenia and autism spectrum disorder: Convergence, divergence and classification." *Neuroimage Clin* **24**: 101966.

Salman, M. S., Y. Du, D. Lin, Z. Fu, A. Fedorov, E. Damaraju, J. Sui, J. Chen, A. Mayer, S. Rosse, D. H. Mathalon, J. M. Ford, T. V. Erp and V. D. Calhoun (2019). "Group ICA for Identifying Biomarkers in Schizophrenia: 'Adaptive' Networks via Spatially Constrained ICA Show More Sensitivity to Group Differences than Spatio-temporal Regression." *Neuroimage Clin* **22**: 101747.

Sasson, N. J., A. E. Pinkham, K. L. Carpenter and A. Belger (2011). "The benefit of directly comparing autism and schizophrenia for revealing mechanisms of social cognitive impairment." *J Neurodev Disord* **3**(2): 87-100.

Sergi, M. J., Y. Rasseovsky, K. H. Nuechterlein and M. F. Green (2006). "Social perception as a mediator of the influence of early visual processing on functional status in schizophrenia." *Am J Psychiatry* **163**(3): 448-454.

Stefanik, L., L. Erdman, S. H. Ameis, G. Foussias, B. H. Mulsant, T. Behdinan, A. Goldenberg, L. J. O'Donnell and A. N. Voineskos (2018). "Brain-Behavior Participant Similarity Networks Among Youth and Emerging Adults with

- Schizophrenia Spectrum, Autism Spectrum, or Bipolar Disorder and Matched Controls." Neuropsychopharmacology **43**(5): 1180-1188.
- Trevisan, D. A., J. H. Foss-Feig, A. J. Naples, V. Srihari, A. Anticevic and J. C. McPartland (2020). "Autism Spectrum Disorder and Schizophrenia Are Better Differentiated by Positive Symptoms Than Negative Symptoms." Frontiers in Psychiatry **11**.
- Tzourio-Mazoyer, N., B. Landeau, D. Papathanassiou, F. Crivello, O. Etard, N. Delcroix, B. Mazoyer and M. Joliot (2002). "Automated anatomical labeling of activations in SPM using a macroscopic anatomical parcellation of the MNI MRI single-subject brain." Neuroimage **15**(1): 273-289.
- Yoshihara, Y., G. Lisi, N. Yahata, J. Fujino, Y. Matsumoto, J. Miyata, G.-i. Sugihara, S.-i. Urayama, M. Kubota, M. Yamashita, R. Hashimoto, N. Ichikawa, W. Cahn, N. E. M. van Haren, S. Mori, Y. Okamoto, K. Kasai, N. Kato, H. Imamizu, R. S. Kahn, A. Sawa, M. Kawato, T. Murai, J. Morimoto and H. Takahashi (2020). "Overlapping but Asymmetrical Relationships Between Schizophrenia and Autism Revealed by Brain Connectivity." Schizophrenia Bulletin **46**(5): 1210-1218.
- Zheng, Z., P. Zheng and X. Zou (2018). "Association Between Schizophrenia and Autism Spectrum Disorder: A Systematic Review and Meta-Analysis." Autism Res.
- Zhu, J. J., S. J. Zhang, H. H. Cai, C. L. Wang and Y. Q. Yu (2020). "Common and distinct functional stability abnormalities across three major psychiatric disorders." Neuroimage-Clinical **27**.
- Zuo, X. N., C. Kelly, J. S. Adelstein, D. F. Klein, F. X. Castellanos and M. P. Milham (2010). "Reliable intrinsic connectivity networks: test-retest evaluation using ICA and dual regression approach." Neuroimage **49**(3): 2163-2177.

Reviewers' comments:

Reviewer #1 (Remarks to the Author):

I feel the authors have adequately addressed my concerns in the revised manuscript. The changes made improved clarity of the methods dramatically. In addition, I feel the added classification results make the paper a lot stronger. There was, however, one point of confusion regarding the classification procedure. The authors state that "...disorder-unique, and ASD-weaker common FNC changes were used as the classification features." However the results section states "The most frequently used features in all classification runs included the unique changes of connectivity between SC (e.g. putamen, caudate, and thalamus) and SM (e.g. superior parietal lobule, paracentral lobule) domains, and also included the ASD-weaker common changes of connectivity between SC and VI/CB domains and between SM and VI domains." Ultimately, it is not clear what procedure was used to include or exclude features used for the the SVM models.

A few minor issues:

Several places the authors use the word 'degrees' in a way that is not clear. For example, in the section Unique changes in brain functional networks "Both SZ-unique and ASD-unique hypo-connectivity (i.e. decrease) degrees were 3.8%." and in Unique changes in functional network connectivity "The degrees of disorder-unique impairments (totaling <25%), with slightly more SZ-unique decreases, were much lower than the overlap degrees" Typo in this heading: SZ and ASD show shard and distinct changes in functional network connectivity Fig. S3 typo "FNCs that past" should read "FNCs that passed"

Reviewer #2 (Remarks to the Author):

I reviewed this paper previously and appreciate all the work that the authors put in to address concerns. The additions to the text to summarize and clarify the findings have greatly helped my understanding. I have a few minor suggestions.

First, the relationships between symptoms and neuroimaging measures are small (all $r < .2$) and are likely only significant due to the large sample size (and the large number of correlations conducted, although the authors did adjust their critical alpha value). I have some concern over how meaningful these relationships really are. In a similar vein that small sample sizes can lead to spurious results, very large sample sizes can highlight 'effects' or 'relationships' that are not meaningful when considering what the results mean for understanding symptomology. Including some caution when describing these relationships would be helpful.

In addition, symptom data are often not normally distributed (for example the positive PANSS scores) and Spearman's correlations are typically conducted to correct for this. Finally, in the main document, the authors write 'ADOS (or SRS)' and make it sound as though these measures are interchangeable. The authors treat them as different measures in their analyses and so this should be clarified in the main text.

Second, there are a number of very dense figures in this paper. For instance, Figure 2 does not seem to add to the explanation of the analyses, especially with text like 'ASD-weaker change in common

decrease'. Decrease compared to what? Whereas other figures such as Figure 4B are helpful but small (had to zoom in 250% to see the FNCs properly). If possible, improving the readability of the figures would be helpful. Please be consistent in either using 'decrease' or 'hypo' etc. as these seem to be interchangeable(?)

Regarding the supplementary materials, the order of the information added into the subsections in the supplementary materials does not match the order of the main paper. Keeping these two documents as similar as possible will assist reading.

Fig S3 is not mentioned in the main document.

There are some typos scattered. For example, "S4. Shard and distinct changes of SZ and ASD in whole-brain functional connectivity using Brainnetome atlas template. Using the Brainnectom template,..."

Reviewer #3 (Remarks to the Author):

The authors have performed a series of new analyses and have added some text to the main body of the manuscript. However, while they have added some information, I'm not convinced that they have answered all of my concerns. The answer to most of the data quality and sampling issues seem to be regressions and leaning back on some thresholds. Specifically:

Comment 1: Is it possible to do a sensitivity analysis that demonstrates what the impact of the lower quality (as per motion QC) is on the reported results). Why is age matching a help here? ASD severity decreases as a function of age in most cases (for example), so this further biases the results). The statement "It is seen that there was no significant group difference in head motion for each dataset, measured by two-sample t-tests." is a bit tough to believe as it goes against almost all of the functional connectivity literature related to ASD. The table provided also disagrees with this statement as it shows clear group differences between cases and controls in certain datasets. This confuses the issue further. It is also unclear to me as to what comment the classifications are meant to address.

Figure 2 provided seems to have several errors, including greater than and less than signs pointing in the wrong direction.

Response letter for manuscript COMMSBIO-20-0724-B

Evidence of shared and distinct functional and structural brain signatures in schizophrenia and autism spectrum disorder

Yuhui Du, Zening Fu, Ying Xing, Dongdong Lin, Godfrey Pearlson, Peter Kochunov, L Elliot Hong, Shile Qi, Mustafa Salman, Anees Abrol, Vince D. Calhoun

Reviewer #1:

I feel the authors have adequately addressed my concerns in the revised manuscript. The changes made improved clarity of the methods dramatically. In addition, I feel the added classification results make the paper a lot stronger. There was, however, one point of confusion regarding the classification procedure. The authors state that "...disorder-unique, and ASD-weaker common FNC changes were used as the classification features." However the results section states "The most frequently used features in all classification runs included the unique changes of connectivity between SC (e.g. putamen, caudate, and thalamus) and SM (e.g. superior parietal lobule, paracentral lobule) domains, and also included the ASD-weaker common changes of connectivity between SC and VI/CB domains and between SM and VI domains." Ultimately, it is not clear what procedure was used to include or exclude features used for the the SVM models.

Response:

We appreciate the favorable evaluation and helpful suggestion. Regarding the SVM-based classification, we have revised the descriptions to clarify.

In the "Methods" section, the revised texts are "Regarding feature selection, we first implemented the above-mentioned statistical analyses on the training data, and then took the FNCs with disorder-unique changes (i.e. SZ-unique decrease and ASD-unique decrease) and the ASD-weaker changes within the disorder-common changes (i.e. ASD-weaker decrease within the common decrease and ASD-weaker increase within the common increase) as the features."

In the "Results" section, the updated descriptions are "We were also interested in whether the identified brain changes can represent promising biomarkers to distinguish the two disorders. As described in the method section, we used the disorder-unique (SZ-unique and ASD-unique) measures and the ASD-weaker measures (within the disorder-common changes) as features for classification.

Our results support that FNC measures performed well in classifying SZ and ASD patients. Table 2 includes the classification results from 12 classification experiments that took different datasets as training and testing data for a comprehensive evaluation. The mean accuracy, sensitivity and specificity across all classifications were 75%, 83%, and 63%, respectively. The best results reached up to 80.0% accuracy, 90.0% sensitivity, and 68.0% specificity.”

A few minor issues:

Several places the authors use the word 'degrees' in a way that is not clear. For example, in the section Unique changes in brain functional networks “Both SZ-unique and ASD-unique hypo-connectivity (i.e. decrease) degrees were 3.8%.” and in Unique changes in functional network connectivity “The degrees of disorder-unique impairments (totaling <25%), with slightly more SZ-unique decreases, were much lower than the overlap degrees”

Response:

Thank you for your comment. The ‘degree’ meant the percentage of each type of change (e.g. SZ-unique decrease) that was calculated as the number of neuroimaging measures relating to the change divided by the number of measures passing ANOVA. We have revised the texts to keep consistency.

Typo in this heading: SZ and ASD show shard and distinct changes in functional network connectivity.

Fig. S3 typo “FNCs that past” should read “FNCs that passed”

Response:

The typos have been corrected. Thank you very much.

Reviewer #2:

I reviewed this paper previously and appreciate all the work that the authors put in to address concerns. The additions to the text to summarize and clarify the findings have greatly helped my understanding. I have a few minor suggestions.

Response:

Thanks for the careful review and good suggestions. We have improved our paper according to the comments.

First, the relationships between symptoms and neuroimaging measures are small (all $r < .2$) and are likely only significant due to the large sample size (and the large number of correlations conducted, although the authors did adjust their critical alpha value). I have some concern over how meaningful these relationships really are. In a similar vein that small sample sizes can lead to spurious results, very large sample sizes can highlight ‘effects’ or ‘relationships’ that are not meaningful when considering what the results mean for understanding symptomology. Including some caution when describing these relationships would be helpful.

Response:

We agree with the comment and appreciate the suggestion. Considering the suggestion, we have added some texts into “Limitations and future directions” section to point out the limitation.

The descriptions are “Another point that needs more concerns is that the biomarker-symptom associations in this work were not very significant (correlations were about 0.2). In future, validation using independent data may be needed to further validate the associations.”

In addition, symptom data are often not normally distributed (for example the positive PANSS scores) and Spearman’s correlations are typically conducted to correct for this. Finally, in the main document, the authors write ‘ADOS (or SRS)’ and make it sound as though these measures are interchangeable. The authors treat them as different measures in their analyses and so this should be clarified in the main text.

Response:

Thanks.

According to the comment, we have computed Spearman rank correlation between each

neuroimaging measure and symptom score. Considering many previous studies used Pearson correlation to explore biomarker-symptom association (Yerys, Herrington et al. 2017, Liu and Huang 2020), we include both Pearson correlation and Spearman rank correlation in the updated Fig. S6 (previous Fig. S5). To maximize the reliability, we only show the neuroimaging measures with p -value < 0.01 for both Pearson and Spearman rank correlations in Fig. S6. The conclusion is consistent to that in our previous manuscript. In addition, we have revised the descriptions to clarify the associations in more detail. The following includes the updated texts and Fig. S6.

“Some important neuroimaging measures were linked to the symptom scores. In order to evaluate the relation between each neuroimaging measure and symptom score, we computed both Pearson correlation and Spearman rank correlation between them for the SZ and ASD groups, separately. As shown in Fig. S6, two FNCs showing disorder-common decreases were negatively correlated with the symptom scores (p -value < 0.01 in both Pearson and Spearman correlations) such as ADOS and SRS in ASD. Interestingly, they were all from the within-domain connectivity (SM and CC). In sum, our results suggest that decreased connectivity strengths between brain regions within the SM and CC domains may relate to worse clinical presentations in disorders.

Five disorder-common increased FNCs, including those between SM and CB, SC and VI, SC and DM, and CC and DM, were positively correlated with the symptom scores in ASD and SZ. That means that increased strengths in those connections (such as the connectivity between CC and DM) could result in worse clinical presentations for both SZ and ASD.

Two disorder-unique FNCs showing decreased strengths in ASD but increased strengths in SZ were also found to be correlated with the SRS in ASD and PANSS positive scores in SZ. Notably, each was connectivity between the VI and CC domains, again indicating the unique property of visual impairment. Moreover, the correlation trends were consistent to the group difference results.”

(A) Associations between FNC with disorder-common decrease and symptom score

(B) Associations between FNC with disorder-common increase and symptom score

(C) Associations between FNC with disorder-unique change and symptom score

Fig. S6 Associations between the FNC strengths and symptom scores of SZ and ASD, measured by Pearson correlation and Spearman rank correlation. (A), (B), and (C) include the correlations for the FNCs with disorder-common decrease, the FNCs with disorder-common increase, and the FNCs with disorder-unique changes, respectively. In the title of each subfigure, we show Pearson correlation (r_1 and p_1) and Spearman rank correlation (r_2 and p_2) for reflecting the association and we also show T-values obtained from two-sample t-tests on any two groups for reflecting the group difference.

Second, there are a number of very dense figures in this paper. For instance, Figure 2 does not seem to add to the explanation of the analyses, especially with text like ‘ASD-weaker change in common decrease’. Decrease compared to what? Whereas other figures such as Figure 4B are helpful but small (had to zoom in 250% to see the FNCs properly). If possible, improving the readability of the figures would be helpful.

Response:

Thanks for the helpful suggestion.

Regarding Fig. 2, we have revised the texts of the caption in Fig. 2 and some descriptions in the main text to clarify the analyses. In addition, we have improved Fig. 2 accordingly.

Fig. 2 Statistical analysis outline for identifying and summarizing the common and unique brain abnormalities of SZ and ASD. The statistical analysis procedure is consistent across all neuroimaging measures mentioned in Fig. 1.

Regarding each measure, analysis of variance (ANOVA) and two-tailed two-sample t-tests were first performed, resulting in group differences for HC vs. SZ, HC vs. ASD, and SZ vs. ASD. For the measures passing ANOVA, different types of changes were then summarized, including the disorder-common decrease compared to HC (the voxels with T-values > 0 in both HC vs. SZ and HC vs. ASD), the disorder-common increase compared to HC (the voxels with T-values < 0 in both HC vs. SZ and HC vs. ASD), the SZ-unique decrease (the voxels with both T-values < 0 in HC vs. ASD and T-values > 0 in HC vs. SZ), and the ASD-unique decrease (the voxels with both T-values < 0 in HC vs. SZ and T-values > 0 in HC vs. ASD). For each of the four types of changes, the percentage was calculated as the number of neuroimaging measures relating to the change divided by the number of measures passing ANOVA. Among the measures with the disorder-common decrease (or increase), the percentage of ASD-weaker decrease (or increase) than SZ, that showed T-values < 0 in SZ vs. ASD in the disorder-common decrease (or showed T-values > 0 in SZ vs. ASD in the disorder-common increase), were further summarized.

The revised texts in the Method section include “Next, based on the results from two-sample t-tests, we summarized the disorder-common and disorder-unique changes. In terms of the voxels passing ANOVA in each network, the voxels with T-values > 0 in both HC vs. SZ and HC vs. ASD reflected the disorder-common decreases compared to HC, and the voxels with T-values < 0 in both HC vs. SZ and HC vs. ASD reflected the disorder-common increases compared to HC. The disorder-unique changes included SZ-unique decrease (i.e. ASD-unique increase) and ASD-unique decrease (i.e. SZ-unique increase). The SZ-unique decrease involved voxels with both T-values > 0 in HC vs. SZ and T-values < 0 in HC vs. ASD. That means for these voxels, network Z-score showed decrease in SZ compared to HC, but showed increase in ASD compared to HC. Similarly, the ASD-unique decrease corresponded to the voxels with both T-values < 0 in HC vs. SZ and T-values > 0 in HC vs. ASD. After that, we computed the voxel percentage for each of the four types of changes within all voxels showing group differences in ANOVA. Furthermore, among the voxels showing disorder-common decrease, we summarized the percentage of the voxels that had weaker decreases in ASD than SZ (i.e. T-values < 0 in SZ vs. ASD), and we called the type of change as ASD-weaker decrease within the common decrease. Similarly, we obtained the associated results for ASD-weaker increase within the common increase. The procedure is outlined in Fig. 2.”

For Fig. 4, we have separated the original figure into two figures (Fig. 4 and Fig. 5) to show the results more clearly. The other figures in the paper have been relabeled

Please be consistent in either using ‘decrease’ or ‘hypo’ etc. as these seem to be interchangeable(?)

Response:

Many thanks for the suggestion. In the whole manuscript, we have replaced the ‘hypo-connectivity’ by ‘decreased connectivity’ and replaced the ‘hyper-connectivity’ by ‘increased connectivity’ to keep consistency. The related figures have also been updated.

Regarding the supplementary materials, the order of the information added into the subsections in the supplementary materials does not match the order of the main paper. Keeping these two documents as similar as possible will assist reading.

Response:

Thanks. To keep the two documents as similar as possible, we have revised the titles of subsections in the supplementary materials to facilitate the understanding of them.

Fig S3 is not mentioned in the main document.

Response:

Sorry for our mistake. In the updated manuscript, we have added some descriptions in terms of Fig. S4 (previous Fig. S3) in the result section. The texts are “As shown in Fig. S4, the overall results were replicated using separate datasets and details are provided in supplementary section S4.”

There are some typos scattered. For example, “S4. Shard and distinct changes of SZ and ASD in whole-brain functional connectivity using Brainnetome atlas template. Using the Brainnectom template,…”

Response:

The typos have been corrected. Thank you very much.

Reviewer #3:

The authors have performed a series of new analyses and have added some text to the main body of the manuscript. However, while they have added some information, I'm not convinced that they have answered all of my concerns. The answer to most of the data quality and sampling issues seem to be regressions and leaning back on some thresholds. Specifically:

Comment 1: Is it possible to do a sensitivity analysis that demonstrates what the impact of the lower quality (as per motion QC) is on the reported results). Why is age matching a help here? ASD severity decreases as a function of age in most cases (for example), so this further biases the results). The statement "It is seen that there was no significant group difference in head motion for each dataset, measured by two-sample t-tests." is a bit tough to believe as it goes against almost all of the functional connectivity literature related to ASD. The table provided also disagrees with this statement as it shows clear group differences between cases and controls in certain datasets. This confuses the issue further. It is also unclear to me as to what comment the classifications are meant to address.

Response:

Thanks for the comments. We would respond to the comments in three points.

(1) In our previous paper, we handled the head motion issue from three aspects. First, we performed the rigid body motion correction to correct the head motion and regressed out nuisance covariates including six head motion parameters from fMRI data in the preprocessing step. Second, we only selected the subjects with head motions less than 3° rotations and 3 mm translations along the whole scanning period (i.e. all time points) in our analyses, in order to further decrease the effect of head motion. From Table S1, it is seen that the mean of the motion translation measure (across subjects) is smaller than 0.3 mm for all datasets, and the mean of the motion rotation measure (across subjects) is smaller than 0.3° for all datasets. Third, the motion effect was minimized for the neuroimaging measures before the statistical analyses. As mentioned in the "Data and preprocessing" of the "Methods" section, fMRI data were preprocessed by regressing out six head motion parameters before the ROI-based functional connectivity estimation. Regarding functional networks estimated from ICA, motion-related noises were removed out by decomposing the data into different components including motion-related components. For the functional network connectivity from

ICA, we also regressed out the motion effects from the time series of functional networks before computing the functional network connectivity.

Considering the comments, we have also revised the descriptions (relating to Table S1) to clarify. The updated texts are “There were no significant group differences (p -value < 0.01) in the head motion measures for each dataset, excepting the head motion transition measure in the ABIDEI (p -value = 0.0084), measured by two-sample t -tests.” We want to point out that although there was kind of motion difference (the mean values of the motion transitions are 0.2018 mm in ASD group and 0.1798 mm in HC group, respectively) in the ABIDEI dataset, the difference should have been decreased since we regressed out the head motion effect from the neuroimaging measures before the statistical analyses.

In order to validate this point, we have evaluated group difference in neuroimaging measures by only using subjects with no differences in head motion measures (i.e. both motion transition and motion rotation). Table S8 includes the information of the selected subjects. Fig. 8(C) shows the FNC group difference estimated using the selected subjects that had no difference in the motion measures. It is observed that the group difference (Fig. 8(C)) presented a consistent pattern with that using all available subjects (Fig. 4). In summary, our experiments support that nuisance effects (including the motion) had been carefully removed out and did not affect the final statistical analyses results. We have revised our paper to include the relevant results.

The revised “Results” section includes:

“Brain changes show consistency using subjects with matched age and subjects with no motion difference

As more data help to generate reliable findings, we used all available data in the above mentioned statistical analyses. In our work, we also investigated group differences in FNC using two additional sample sets. The two sets had different age ranges of subjects, and each set only included age-matched three groups. In addition, we selected some subjects with no motion difference in fMRI data to test the group differences. Our results (shown in Fig. 8) suggest that regardless of the sample sets, the group differences using age-matched subjects or no-motion-difference subjects tended to show similar patterns with the results using all available subjects (shown in Fig. 4), supporting that the nuisance effects (such as age and motion) had been carefully removed out. Our results support that our findings are relatively robust.”

Fig. 8 FNC group differences using subjects with matched age and subjects with no motion difference. (A) and (B) show group differences obtained using the sample set 1 and 2 (see Table S7, each of them includes age-matched subjects but they had different age ranges), respectively. (C) shows group differences obtained using subjects (see Table S8) with no motion differences. T-values obtained using two-sample t-tests on any pair of groups are displayed in order to compare with the results (Fig. 4) estimated using all subjects.

The added texts in the “Exploring brain changes using subjects with matched age and subjects with no motion difference” section of “Methods” includes “In the work, there were some group differences in head motion while combining subjects from all datasets, although in general there were no significant group differences in motion for each dataset. Therefore, from the large-size sample, we selected some subjects with no motion differences and verified the group differences

using the same analysis method. The details can be found in supplementary section S7.”

The added texts in the “S7. Exploring brain changes using subjects with matched age and subjects with no motion difference: information of subjects” section of “Supplementary materials” includes “We also identified group difference by only using subjects with no group difference in head motion to validate the results. Table S8 includes the information of subjects that we selected and the related motion measures. There was no significant group difference in motion across HC, SZ, and ASD, tested by ANOVA.”

Table S8. Information of the selected subjects with no motion differences across the HC, SZ, and ASD groups.

	HC	SZ	ASD
Subject number	838	212	513
Motion transitions: mean (std)	0.2431 (0.1327)	0.2578 (0.1395)	0.2485 (0.1443)
Motion rotations: mean (std)	0.2522 (0.1317)	0.2464 (0.1497)	0.2624 (0.1482)
p-value of motion transitions, tested by ANOVA	0.3616		
p-value of motion rotations, tested by ANOVA	0.2761		

(2) Considering the comment suggested a sensitivity analysis, we have added permutation tests to evaluate group differences. Our results support that the group differences that we identified are robust.

In the supplementary “S3. Investigating functional network connectivity revealed by ICA: group differences evaluated by a permutation test” section, we added the following texts.

“To assess the validity of the group differences, we conducted a permutation test with 1000 permutations to identify the group difference between different groups (e.g. HC vs. SZ), and then compared the results with the previous results obtained using the direct two-sample t-test. Taking HC vs. SZ for an example, we introduce how the permutation test was applied. In each of 1000 permutations, we randomly rearranged the subjects (all HC and SZ subjects) into two dummy groups each of which had the same number of subjects with the original group, and then applied two-sample t-tests on the dummy groups to evaluate the group differences in FNCs. After that, for each FNC, we calculated the occurring frequency of the case where the p-value obtained from two-sample t-test using the dummy groups was smaller than the corresponding p-value obtained from two-sample

t-test using the original groups, and then took the frequency as the final p-value for the FNC. Smaller frequency represents lower possibility of false positives of the identified group differences. The final p-values of FNCs were corrected by Bonferroni (BFN) correction. It is seen that the group differences identified using the permutation test (Fig. S3) were quite consistent to that estimated using the direct two-sample t-test (Fig. 4), supporting that the group differences were reliable.”

Fig. S3 Group differences that were obtained from a permutation test after Bonferroni (BFN) correction. Here, we show the T-values (HC vs. SZ, HC vs. ASD, and SZ vs. ASD using original groups) of FNCs that passed BFN correction according to the final p-values of the permutation test.

In the “Results” section, we added “In order to examine if group differences are sensitive to the statistical analysis method, we also conducted a permutation test (with 1000 permutations) instead of the above mentioned direct two-sample t-test to investigate group differences. Please find the details in the supplementary section S3.”

(3) Regarding using the age-matched subjects to investigate the group differences, we aimed to show that age effect has been well removed out before the statistical analyses, because that the group differences estimated using the age-matched subjects (Fig. 8(A)-(B), see below) tended to show similar patterns with the results using all available subjects (Fig. 4, see below). More interestingly, we selected two sample sets with varied age ranges (Table S7, see below), each sample set including age-matched subjects, to test group differences in our paper. Our results show that although the age ranges were different between the two sample sets, the group differences were quite consistent with each other (Fig. 8(A) and (B), see below), and were also similar to the results from using all subjects (Fig. 4, see below).

Table S7. Two sample sets with slightly different age ranges. Each sample set included age-matched HC, SZ, and ASD subjects.

		HC	SZ	ASD	
Sample set 1	Subject number	442	222	130	
	Age	Range	[21, 35]	[21, 35]	[21, 36]
		Mean	26.61	27.09	26.91
		Std	3.88	4.17	4.32
	p-value of age among the three groups, tested by ANOVA		0.0797		
Sample set 2	Subject number	461	248	104	
	Age	Range	[23, 42]	[23, 42]	[23, 42]
		Mean	30.50	31.09	29.60
		Std	5.67	5.74	5.49
	p-value of age among the three groups, tested by ANOVA		0.0731		

Note: Std denotes standard deviation.

Fig. 8 FNC group differences using subjects with matched age and subjects with no motion difference. (A) and (B) show group differences obtained using the sample set 1 and 2 (see Table S7, each of them includes age-matched subjects but they had different age ranges), respectively. (C) shows group differences obtained using subjects (see Table S8) with no motion differences. T-values obtained using two-sample t-tests on any pair of groups are displayed in order to compare with the results (Fig. 4) estimated using all subjects.

Fig. 4 Results from functional network connectivity (FNC) analysis. Upper subfigures: T-value maps showing the group differences in FNCs, obtained by two-sample t-tests for HC vs. SZ, HC vs. ASD, and SZ vs. ASD. Taking HC vs. SZ for example, positive T-values represented that HCs showed higher connectivity strengths than SZs. Lower subfigures: T-value maps of FNCs after Bonferroni (BFN) corrections.

(4) We have also revised the descriptions to explain more about the classification. Since we were interested in whether the identified brain changes can represent promising biomarkers to distinguish the two disorders well, we tested the classification ability using those biomarkers. To avoid bias, we conducted two-class (SZ and ASD) classification by taking different datasets for training and the remaining datasets for testing based on the FNC measures. Since there were four datasets relating to the SZ group and two datasets relating to the ASD group, totally we performed 12 classification experiments using different data assignments. Feature extraction and model training were implemented only using the training data, and then the testing data were classified and compared with their true class labels. Regarding feature selection, we first implemented the statistical analyses on the training data, and then took the FNCs with disorder-unique changes (i.e. SZ-unique decrease and ASD-unique decrease) and the ASD-weaker changes within the disorder-common changes (i.e. ASD-weaker decrease within the common decrease and ASD-weaker

increase within the common increase) as the features. Linear support vector machine with a Bayesian optimization technology (A Snoek, A Larochelle et al. 2012) to optimize the parameter was applied for the model building. Finally, the classification results were evaluated using accuracy, sensitivity, and specificity. Table 2 shows the classification results from 12 classification experiments that took different datasets as training and testing data for a comprehensive evaluation. Our results support that FNC measures performed well in classifying SZ and ASD patients.

Comment 2: Figure 2 provided seems to have several errors, including greater than and less than signs pointing in the wrong direction.

Response:

The signs in Fig. 2 are correct. Considering your comments, we have revised the caption in Fig. 2 and some descriptions in the main text to clarify the analyses. In addition, we have improved Fig. 2 accordingly (see below).

Fig. 2 Statistical analysis outline for identifying and summarizing the common and unique brain abnormalities of SZ and ASD. The statistical analysis procedure is consistent across all neuroimaging measures mentioned in Fig. 1. Regarding each measure, analysis of variance (ANOVA) and two-tailed two-sample t-tests were first performed, resulting in group differences for HC vs. SZ, HC vs. ASD, and SZ vs. ASD. For the measures passing ANOVA, different types of changes were then summarized, including the disorder-common decrease compared to HC (the voxels with T-values > 0 in both HC vs. SZ and HC vs. ASD), the disorder-common increase compared to HC (the voxels with T-values < 0 in both HC vs. SZ and HC vs. ASD), the SZ-unique decrease (the voxels with both T-values < 0 in HC vs. ASD and T-values > 0 in HC vs. SZ), and the ASD-unique decrease (the voxels with both T-values < 0 in HC vs. SZ and T-values > 0 in HC vs. ASD). For each of the four types of changes, the percentage was calculated as the number of neuroimaging measures relating to the change divided by the number of measures passing ANOVA. Among the measures with the disorder-common decrease (or increase), the percentage of ASD-weaker decrease (or increase) than SZ, that showed T-values < 0 in SZ vs. ASD in the disorder-common

decrease (or showed T-values > 0 in SZ vs. ASD in the disorder-common increase), were further summarized.

The revised texts in the “Methods” section are: “Next, based on the results from two-sample t-tests, we summarized the disorder-common and disorder-unique changes. In terms of the voxels passing ANOVA in each network, the voxels with T-values > 0 in both HC vs. SZ and HC vs. ASD reflected the disorder-common decreases compared to HC, and the voxels with T-values < 0 in both HC vs. SZ and HC vs. ASD reflected the disorder-common increases compared to HC. The disorder-unique changes included SZ-unique decrease (i.e. ASD-unique increase) and ASD-unique decrease (i.e. SZ-unique increase). The SZ-unique decrease involved voxels with both T-values > 0 in HC vs. SZ and T-values < 0 in HC vs. ASD. That means for these voxels, network Z-score showed decrease in SZ compared to HC, but showed increase in ASD compared to HC. Similarly, the ASD-unique decrease corresponded to the voxles with both T-values < 0 in HC vs. SZ and T-values > 0 in HC vs. ASD. After that, we computed the voxel percentage for each of the four types of changes within all voxels showing group differences in ANOVA. Furthermore, among the voxles showing disorder-common decrease, we summarized the percentage of the voxles that had weaker decreases in ASD than SZ (i.e. T-values < 0 in SZ vs. ASD), and we called the type of change as ASD-weaker decrease within the common decrease. Similarly, we obtained the associated results for ASD-weaker increase within the common increase. The procedure is outlined in Fig. 2.”

Reference:

- A Snoek, J., H. A Larochelle and R. P. A Adams (2012). "Practical Bayesian Optimization of Machine Learning Algorithms." NIPS: 2960-2968.
- Liu, X. and H. Huang (2020). "Alterations of functional connectivities associated with autism spectrum disorder symptom severity: a multi-site study using multivariate pattern analysis." Sci Rep **10**(1): 4330.
- Yerys, B. E., J. D. Herrington, T. D. Satterthwaite, L. Guy, R. T. Schultz and D. S. Bassett (2017). "Globally weaker and topologically different: resting-state connectivity in youth with autism." Mol Autism **8**: 39.

REVIEWERS' COMMENTS:

Reviewer #2 (Remarks to the Author):

I have reviewed this paper twice previously and find that the authors continue to respond to my comments and improve the clarity of their paper – thank you.

I have a few (very) minor suggestions that I hope will add further clarity. In the results, the subheadings clearly delineate when the analyses are switching comparisons (following Fig 2). However, adding some extra clarity as to what group is being compared to what will avoid the reader having to flick through pages to check Fig 2 or to wait until the discussion for the summary. For example, for the subheading “Common changes in brain functional networks (line 180)” is not immediately obvious that this is assessing what is common across SZ and ASD compared to HC. Similarly, the sentence (line 189), “Notably, for the voxels with common decrease changes, 85.3% voxels showed weaker (smaller) changes in ASD than SZ; for the voxels with common increase changes, 94.4% voxels had weaker (smaller) changes in ASD than SZ, supporting that in general ASD presented weaker changes than SZ for the shared abnormalities” is confusing. Does this mean that SZ is more different than ASD compared to HC? This seems to be the message from the discussion, but it would be helpful to be clearer early. Again, for the sentence (line 201), “Regions with ASD-unique decreases were primarily located at the superior frontal gyrus (IC 96, CC), hippocampus (IC 83, CC), and middle cingulate cortex (IC 37, SM).” Is this the analysis where SZ are increased compared to HC (as described in Fig 2)? Or are these just ASD-unique decreases alone?

Please be specific about what symptom scores are being compared (starting line 318). At present, the symptoms being compared in SZ are missing from this section.

In the introduction (line 108), Haigh et al. (2019) used diffusion MRI (not structural) and the fMRI findings were from the Haigh et al. (2016) paper.

There are still lots of typos. For example (line 853), “There were no significant group differences (p-value < 0.01) in head motion(s) for each dataset, except(ing) the head motion transition measure in the ABIDE()I (p-value = 0.0084), measured by two-sample t-tests. If combing (combining?)...”

Response letter for manuscript COMMSBIO-20-0724-C

**Evidence of shared and distinct functional and structural brain signatures in
schizophrenia and autism spectrum disorder**

*Yuhui Du, Zening Fu, Ying Xing, Dongdong Lin, Godfrey Pearlson, Peter Kochunov, L
Elliot Hong, Shile Qi, Mustafa Salman, Anees Abrol, Vince D. Calhoun*

Reviewer #2:

I have reviewed this paper twice previously and find that the authors continue to respond to my comments and improve the clarity of their paper – thank you.

I have a few (very) minor suggestions that I hope will add further clarity. In the results, the subheadings clearly delineate when the analyses are switching comparisons (following Fig 2). However, adding some extra clarity as to what group is being compared to what will avoid the reader having to flick through pages to check Fig 2 or to wait until the discussion for the summary. For example, for the subheading “Common changes in brain functional networks (line 180)” is not immediately obvious that this is assessing what is common across SZ and ASD compared to HC. Similarly, the sentence (line 189), “Notably, for the voxels with common decrease changes, 85.3% voxels showed weaker (smaller) changes in ASD than SZ; for the voxels with common increase changes, 94.4% voxels had weaker (smaller) changes in ASD than SZ, supporting that in general ASD presented weaker changes than SZ for the shared abnormalities” is confusing. Does this mean that SZ is more different than ASD compared to HC? This seems to be the message from the discussion, but it would be helpful to be clearer early. Again, for the sentence (line 201), “Regions with ASD-unique decreases were primarily located at the superior frontal gyrus (IC 96, CC), hippocampus (IC 83, CC), and middle cingulate cortex (IC 37, SM).” Is this the analysis where SZ are increased compared to HC (as described in Fig 2)? Or are these just ASD-unique decreases alone?

Response:

(1) We have revised the subheadings of some subsections to delineate when the analyses are switching comparisons.

(2) We have also revised some texts to describe the results in more detail. The updated texts include “Notably, for the voxels with common decreases in SZ and ASD (relative to HC), 85.3% voxels showed weaker (smaller) changes in ASD than SZ; for the voxels with common increases in SZ and ASD (relative to HC), 94.4% voxels had weaker (smaller) changes in ASD than SZ, supporting that in general ASD presented weaker changes than SZ for the shared abnormalities.” “Regions with ASD-unique decreases (in which ASD decreased but SZ increased, relative to HC) were primarily located at the superior frontal gyrus (IC 96, CC), hippocampus (IC 83, CC), and middle cingulate cortex (IC 37, SM).”

Please be specific about what symptom scores are being compared (starting line 318). At present, the symptoms being compared in SZ are missing from this section.

Response:

We have added descriptions to clarify the evaluation using symptom scores. The updated texts are “To evaluate the relationship between neuroimaging measure and symptom score (i.e., the positive and negative syndrome scale (PANSS) positive score and PANSS negative score for SZ; autism diagnostic observation schedule (ADOS) total score and social responsiveness scale (SRS) for ASD), we computed both Pearson correlation and Spearman rank correlation between them for the SZ and ASD groups, separately.”

In the introduction (line 108), Haigh et al. (2019) used diffusion MRI (not structural) and the fMRI findings were from the Haigh et al. (2016) paper.

Response:

Regarding the reference (Haigh, Eack et, 2019), the previous sentence is “In addition to using fMRI and sMRI data, Haigh et al. used diffusion data to study the two disorders, revealing SZ-specific changes in its greater mean diffusivity than both ASD and HC.” We meant others used fMRI and sMRI data but Haigh used diffusion data. Considering the comment, we have removed “In addition to using fMRI and sMRI data” to avoid confusion.

There are still lots of typos. For example (line 853), “There were no significant group differences (p-value < 0.01) in head motion(s) for each dataset, except(ing) the

head motion transition measure in the ABIDE()I (p-value = 0.0084), measured by two-sample t-tests. If combing (combining?)..."

Response:

The typos have been corrected. In addition, we have checked the paper again.